# Dimension-free convergence of diffusion models for approximate Gaussian mixtures

Gen Li [* 1]   Changxiao Cai [* 2]   Yuting Wei [3]

## Abstract

Diffusion models are distinguished by their exceptional generative performance, particularly in producing high-quality samples through iterative denoising. While current theory suggests that the number of denoising steps required for accurate sample generation should scale linearly with data dimension, this does not reflect the practical efficiency of widely used algorithms like Denoising Diffusion Probabilistic Models (DDPMs). This paper investigates the effectiveness of diffusion models in sampling complex high-dimensional distributions that can be well-approximated by Gaussian Mixture Models (GMMs). For these distributions, our main result shows that DDPM takes at most $\widetilde{O}(1/\varepsilon)$ iterations to attain an $\varepsilon$-accurate distribution in total variation (TV) distance, independent of both the ambient dimension $d$ and the number of components $K$, up to logarithmic factors. Furthermore, this result remains robust to score estimation errors. These findings highlight the remarkable effectiveness of diffusion models in high-dimensional settings given the universal approximation capability of GMMs, and provide theoretical insights into their practical success.

## 1. Introduction

Diffusion models have garnered significant attention for their remarkable generative capabilities, producing high-quality samples with enhanced stability (Diakonikolas et al., 2018; Dhariwal & Nichol, 2021; Song et al., 2020c; Ramesh

*Equal contribution [1]Department of Statistics and Data Science, Chinese University of Hong Kong, Shatin, New Territories, Hong Kong [2]Department of Industrial and Operations Engineering, University of Michigan, Ann Arbor, MI, USA [3]Department of Statistics and Data Science, University of Pennsylvania, Philadelphia, PA, USA. Correspondence to: Yuting Wei <ytwei@wharton.upenn.edu>.

*Proceedings of the 43$^{rd}$ International Conference on Machine Learning*, Seoul, South Korea. PMLR 306, 2026. Copyright 2026 by the author(s).

et al., 2022). Compared to methods like Generative Adversarial Networks (GANs) and Variational Autoencoders (VAEs), which generate samples in a single forward pass, diffusion models are designed to iteratively denoise samples over hundreds or thousands of steps. A prominent example is the widely used Denoising Diffusion Probabilistic Models (DDPM) sampler (Ho et al., 2020). The current theory suggests the number of denoising steps required for accurate sample generation should scale at least linearly with the data dimension (Chen et al., 2022; Benton et al., 2023) in order to learn the distribution accurately. While various acceleration schemes have been proposed in literature (see, e.g. (Li & Cai, 2024; Li et al., 2024a; Li & Jiao, 2024; Wu et al., 2024b; Huang et al., 2024b; Li et al., 2025; Huang et al., 2024a; Taheri & Lederer, 2025; Jiao et al., 2025)), in practical applications such as high-resolution image synthesis, where the dimensionality of the data can be extremely large, DDPM often requires far fewer steps than predicted by theory while maintaining excellent sample quality.

This gap between theoretical complexity bounds and empirical performance has inspired a strand of recent research, investigating whether diffusion models have implicitly exploited structural properties of real-world data to circumvent worst-case complexity bounds. A growing line of works have shed light on this question by showing that popular samplers, such as DDPM, in their original form, can automatically adapt to the intrinsic dimension of the target distribution without explicitly modeling its low-dimensional structure. Notably, prior work has examined cases where the data lies in low-dimensional linear spaces, low-dimensional manifolds, or distributions whose support have small covering number (Li & Yan, 2024a; Tang & Yang, 2024; Huang et al., 2024c; Potaptchik et al., 2024; Liang et al., 2025). In this work, we take a different perspective, and explore this question by focusing on a fundamental and well-studied statistical model: Gaussian Mixture Models (GMMs). GMMs serve as a cornerstone of statistical modeling and have been widely used to approximate complex distributions. Formally, we consider the setting where the target distribution is or can be well-approximated by a mixture of isotropic Gaussians:

$$\sum_{k=1}^{K} \pi_k \mathcal{N}(\mu_k, \sigma^2 I_d). \qquad (1)$$

Here, $\{\pi_k\}$ are mixture weights satisfying $\pi_k \in (0,1)$ and $\sum_{k=1}^{K} \pi_k = 1$. The study of Gaussian Mixture Models (GMMs) dates back to (Pearson, 1894), and a vast body of literature has since explored various aspects of GMMs, including parameter estimation, distribution learning, information-theoretic limits, computational efficiency and etc. This paper studies the performance of diffusion models in their original form when they are used to generate samples from a distribution that is close to GMMs. We refer readers to a more detailed exposition of related work in Section 1.3.

### 1.1. Diffusion models and sampling efficiency

In a nutshell, diffusion models consist of two processes: a forward process and a backward process. In the forward process, noise is gradually added to the data, transforming it into a noise-like distribution chosen *a priori* (e.g., a Gaussian distribution). Mathematically, given an initial sample $X_0 \in \mathbb{R}^d$ from the target distribution $p_{\mathsf{data}}$, this transformation follows

$$X_t = \sqrt{\alpha_t} X_{t-1} + \sqrt{1 - \alpha_t} W_t, \quad t = 1, 2, \ldots, T, \quad (2)$$

where $\{\alpha_t \in (0,1)\}_{t \geq 1}$ denote the learning rates and $W_t \overset{\text{i.i.d.}}{\sim} \mathcal{N}(0, I_d), t \geq 1$ are i.i.d. standard $d$-dimensional Gaussian vectors independent of $(X_t)_{t=0}^{T}$. In the backward process, starting from $Y_T \sim \mathcal{N}(0, I_d)$, diffusion models iteratively denoise $Y_T$ to approximate $p_{\mathsf{data}}$. Classical results from stochastic differential equations (SDE) theory (e.g. (Anderson, 1982; Haussmann & Pardoux, 1986)) show that under mild conditions, recovering $p_{\mathsf{data}}$ is possible provided access to the (Stein) score function $s_t^\star(\cdot) : \mathbb{R}^d \to \mathbb{R}^d$ for all $1 \leq t \leq T$, defined as

$$s_t^\star(x) := \nabla \log p_{X_t}(x), \quad \forall x \in \mathbb{R}^d. \quad (3)$$

Given the complexity of developing a comprehensive end-to-end theory, a divide-and-conquer approach — pioneered by (Chen et al., 2022) — has become standard, separating the score learning phase (i.e., estimating score functions reliably from training data) from the generative sampling phase (i.e., generating new data instances based on the estimated scores). The quality of the sampler in terms of its discrepancy to the target distribution depends on the errors from both phases. Over the past several years, the theoretical community has made significant process in understanding both phases. Notably, for the sampling phase, convergence theory has been established for various samplers (Liu et al., 2022; Lee et al., 2023; Chen et al., 2023a; Li et al., 2023; Chen et al., 2023c; Tang & Zhao, 2024; Liang et al., 2024a; Huang et al., 2024a; Gao & Zhu, 2024), especially DDPM and Denoising Diffusion Implicit Models (DDIM) which are widely adopted in practice (Ho et al., 2020; Song et al., 2020a). For DDPM, Benton et al. (2023) establishes an

iteration complexity of $\widetilde{O}(d/\varepsilon^2)$ [1] in Kullback Leibler (KL) divergence, and (Li & Yan, 2024b) shows a complexity of $\widetilde{O}(d/\varepsilon)$ in total variation (TV) distance. When it comes to the DDIM sampler or the probability flow ODE, notably, an $\widetilde{O}(d/\varepsilon)$ iteration complexity has been established in (Li et al., 2024b).

### 1.2. Learning GMMs using diffusion models

In the context of GMMs, several recent works have contributed towards unraveling the capabilities of diffusion models. In particular, inspired by diffusion models, (Shah et al., 2023) introduced an algorithm designed for GMMs that achieves polynomial time complexity in $d$, provided the component centers are well-separated. (Liang et al., 2024b) established an iteration complexity of $\widetilde{O}(d/\varepsilon^2)$ for obtaining an $\varepsilon$-accurate distribution measured in TV distance by analyzing the Lipschitz and second moments of GMMs. Additionally, (Wu et al., 2024a; Chidambaram et al., 2024) investigated the role of guidance in diffusion models. Two exciting recent works (Chen et al., 2024; Gatmiry et al., 2024) proposed using piecewise polynomial regression to estimate the score functions, and they combined this with existing convergence result for DDPM to develop an end-to-end theory for DDPM. Notably, in these works, the number of diffusion steps scales also linearly with $d$. Further, Wang et al. (2024) explored diffusion models for mixtures of low-rank Gaussians. Despite these advancements, a fundamental question remains open:

*Can diffusion models achieve efficient sampling when the target distribution is close to a GMM?*

Numerical evidence in Figure 1 suggests that the answer is affirmative. When the number of DDPM iterations is fixed, the sampling accuracy remains stable as the ambient dimension increases and exhibits only a mild (logarithmic) dependence on the number of mixture components. In this work, we aim to provide a theoretical justification for this phenomenon.

**A glimpse of our main contributions.** This paper investigates sampling from target distributions which admit faithful approximations by isotropic GMMs, without imposing any constraint on the component separation or mixture weights. Our main result provides a non-asymptotic characterization of DDPM's iteration complexity for learning an $\varepsilon$-accurate distribution in TV distance. We prove that, given perfect score estimates and small GMM approximation error, it takes DDPM at most

$$\widetilde{O}\left(\frac{1}{\varepsilon}\right),$$

---

[1] The definition for $O(\cdot)$ and $\widetilde{O}(\cdot)$ notation can be found in Section 1.4.

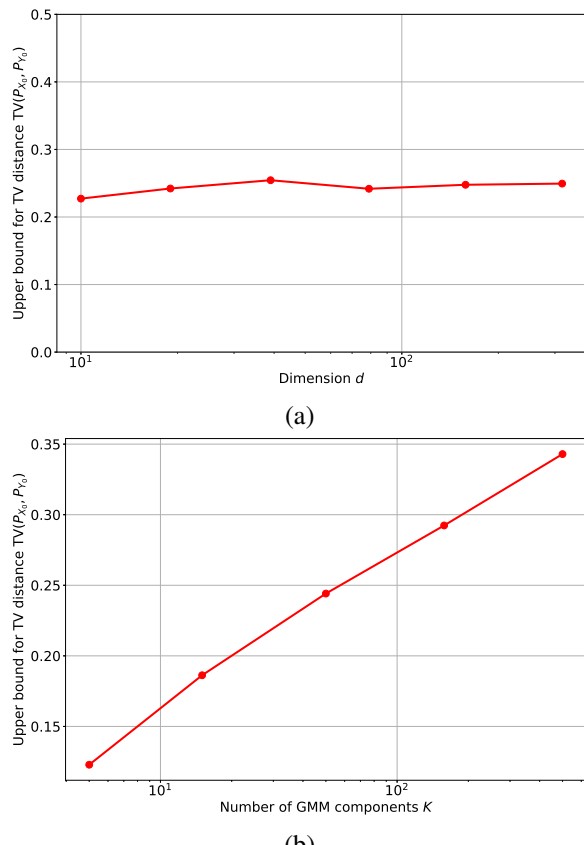

(a)

(b)

*Figure 1.* Consider a GMM with $K$ components in $\mathbb{R}^d$, where the mixture weights are set to be uniform, i.e., $\pi_k = 1/K$ for all $k \in [K]$ and the component means $\mu_k$ are sampled from $\mathcal{N}(0, 10I_d)$. We fix the number of DDPM sampling iterations to $T = 200$ and plot the total variation (TV) distance between the DDPM-generated samples and the target distribution as a function of (a) the ambient dimension $d$ and (b) the number of mixture components $K$. In plot (a), we fix $K = 50$; in plot (b), we fix $d = 50$. The implementation details can be found in Appendix C.

number of iterations. Remarkably, this iteration complexity is independent of both the ambient dimension $d$ and the number of components $K$, up to some logarithmic factors. Moreover, our result is robust to score estimation errors: the TV distance between the learned distribution and the target distribution scales proportionally to the score estimation error, modulo logarithmic factor. This leads to a surprising insight:

*Even in ultra-high-dimensional settings, diffusion models remain highly effective in sampling distributions that are close to GMMs.*

### 1.3. Other related works

**Learning GMMs.** GMMs are fundamental statistical models that bear a well-established body of research from both statistics and computer science communities. One ma-

jor line of research focuses on parameter estimation with some separation conditions. Partial examples include (Dasgupta, 1999; Vempala & Wang, 2004; Arora & Kannan, 2005; Kalai et al., 2010; Hsu & Kakade, 2013; Diakonikolas et al., 2018; Hopkins & Li, 2018; Kothari et al., 2018; Liu & Li, 2022).

Our work is more closely related to the density estimation perspective, where no separation conditions are imposed (e.g. (Diakonikolas & Kane, 2020; Moitra & Valiant, 2010; Dwivedi et al., 2020; Bakshi et al., 2022; Ho & Nguyen, 2016)). In this setting, parameter estimation is information-theoretically infeasbile, yet accurate density estimation is still possible. The information theoretical limit of this problem is first characterized in (Ashtiani et al., 2018) up to logarithmic factors, with a brute-forth algorithm that scales exponentially in both $d$ and $K$. For one-dimensional Gaussian mixtures, (Chen, 1995; Heinrich & Kahn, 2018; Wu & Yang, 2020) obtained optimal estimation rates and practical algorithms, which were generalized to the high-dimensional case for mixtures of spherical Gaussians with a computationally efficient algorithm in (Doss et al., 2023). Beyond finite mixtures, when mixing distribution is an arbitrary probability measure (e.g. (Genovese & Wasserman, 2000; Ghosal & Van Der Vaart, 2001)), (Saha & Guntuboyina, 2020; Polyanskiy & Wu, 2020; Kim & Guntuboyina, 2022) established convergence rates and adaptivity, regarding the non-parametric maximum likelihood estimatior, generalizing the one-dimension results in (Zhang, 2009).

**Score estimation.** As mentioned earlier, score estimation plays a crucial role in diffusion models. Hyvärinen (2005) introduced an integration-by-parts-based approach to simplify score estimation. More recently, Song et al. (2020b) proposed training neural networks to learn score functions by minimizing the score matching objective. The theoretical guarantees for score estimation using neural networks have been analyzed across various distributional settings, including sub-Gaussian distributions (Cole & Lu, 2024), graphical models (Mei & Wu, 2023), low-dimensional structured distributions (Chen et al., 2023b; Kwon et al., 2025; De Bortoli, 2022), and Besov function space (Oko et al., 2023). These guarantees are often achieved by designing neural architectures that well approximate the true score function. Other than neural networks, classical methods such as kernel-based approaches and empirical Bayes smoothing have also been studied for score estimation (Cai & Li, 2025; Wibisono et al., 2024; Zhang et al., 2024; Dou et al., 2024; Wu & Cai, 2026). These methods have been shown to achieve minimax-optimal rates under some smoothness assumptions. Furthermore, Feng et al. (2024) demonstrated that statistical procedures based on score matching can achieve minimal asymptotic covariance for convex M-estimation.

## 1.4. Notation

For any $a$, the Dirac delta function $\delta_a(x)$ is defined as $\delta_a(x) = \infty$ if $x = a$ and $\delta_a(x) = 0$ otherwise. For positive integer $N > 0$, let $[N] := \{1, \cdots, N\}$. In addition, given any matrix $A$, we use $\|A\|$, $\mathsf{tr}(A)$, and $\det(A)$ to denote the spectral norm, trace, and determinant of the matrix, respectively. Next, we recall the definitions of the KL divergence and TV distance to measure the discrepancies between two distributions. Specifically, for random vectors $X$ and $Y$ with probability density functions $p_X$ and $p_Y$, let

$$\mathsf{KL}(X \,\|\, Y) \equiv \mathsf{KL}(p_X \,\|\, p_Y) = \int p_X(x) \log \frac{p_X(x)}{p_Y(x)} \, \mathrm{d}x,$$

$$\mathsf{TV}(X, Y) \equiv \mathsf{TV}(p_X, p_Y) = \frac{1}{2} \int |p_X(x) - p_Y(x)| \, \mathrm{d}x.$$

For any two functions $f(T)$, $g(T) > 0$, we write $f(T) \lesssim g(T)$ or $f(T) = O\big(g(T)\big)$ to indicate $f(T) \leq Cg(T)$ for some absolute constant $C > 0$. We say $f(T) \asymp g(T)$ when $Cf(T) \leq g(T) \leq C'f(T)$ for some absolute constants $C' > C > 0$. The notation $\widetilde{O}(\cdot)$ and $\widetilde{\Omega}(\cdot)$ represent the respective bounds up to logarithmic factors. Finally, we write $f(T) = o(g(T))$ to denote $\limsup_{T \to \infty} f(T)/g(T) = 0$.

## 2. Preliminaries for diffusion models

Given training samples from a target distribution $p_{\mathsf{data}}$ on $\mathbb{R}^d$, diffusion models aim to generate new samples from $p_{\mathsf{data}}$. Recall the forward process (2). If we define

$$\overline{\alpha}_t := \prod_{k=1}^{t} \alpha_k, \quad t = 1, 2, \ldots, T, \tag{4}$$

the forward process can be expressed as a linear combination of the initial distribution and a Gaussian noise

$$X_t = \sqrt{\overline{\alpha}_t} X_0 + \sqrt{1 - \overline{\alpha}_t} \, \overline{W}_t, \quad t = 1, 2, \ldots, T, \tag{5}$$

where $\overline{W}_t \sim \mathcal{N}(0, I_d)$ denotes a $d$-dimensional standard Gaussian random vector independent of $X_0$. When $\overline{\alpha}_T$ is sufficiently small, $X_T$ is well-approximated by a standard Gaussian distribution. Taking the continuum limit of (2), the process satisfies the stochastic differential equation (SDE):

$$X_0 \sim p_{\mathsf{data}};$$
$$\mathrm{d}X_t = -\frac{1}{2}\beta_t X_t \, \mathrm{d}t + \sqrt{\beta_t} \, \mathrm{d}B_t, \quad t \in [0, T] \tag{6}$$

for some function $\beta_t : [0, T] \to \mathbb{R}$, where $(B_t)_{t \in [0,T]}$ is a standard Brownian motion in $\mathbb{R}^d$.

Diffusion models seek to reverse the above process by iteratively denoising noisy samples geneerated from $\mathcal{N}(0, I_d)$, reconstructing data samples from $p_{\mathsf{data}}$. From a continuous perspective, given a solution $(X_t)_{t \in [0,T]}$ to (6), classical

SDE theory (Anderson, 1982; Haussmann & Pardoux, 1986) ensures that its time reversal $Y_t^{\mathsf{SDE}} := X_{T-t}$ satisfies:

$$Y_0^{\mathsf{SDE}} \sim p_{X_T};$$
$$\mathrm{d}Y_t^{\mathsf{SDE}} = \frac{1}{2}\beta_{T-t}\Big(Y_t^{\mathsf{SDE}} + 2\nabla \log p_{X_{T-t}}\big(Y_t^{\mathsf{SDE}}\big)\Big) \, \mathrm{d}t$$
$$+ \sqrt{\beta_{T-t}} \, \mathrm{d}B_t, \quad t \in [0, T]. \tag{7}$$

Here, $p_{X_t}$ denotes the marginal distribution of $X_t$ in the forward SDE (6).

**Score learning/matching.** It is clear from the continuous perspective, that the score function $s_t^\star(x) := \nabla \log p_{X_t}(x)$ plays an important role in characterizing the reverse process. In fact, if $s_t^\star(x)$ were known exactly, the reverse process would be uniquely identified. In practice, however, score functions must be learned from training samples. A natural approach is to estimate $s_t^\star(x)$ within a pre-selected function class $\mathcal{F}$ by minimizing the expected squared error:

$$\min_{\widehat{s}_t \in \mathcal{F}} \mathbb{E}_{X \sim p_{X_t}} \left[ \big\| \widehat{s}_t(X) - \nabla \log p_{X_t}(X) \big\|^2 \right].$$

For Gaussian distributions, integration by parts allows reformulating this objective as (e.g., Hyvärinen (2005); Vincent (2011); Chen et al. (2022))

$$\min_{\widehat{s}_t : \mathbb{R}^d \to \mathbb{R}^d} \mathbb{E}_{W \sim \mathcal{N}(0, I_d), X_0 \sim p_{\mathsf{data}}} \left[ \left\| \widehat{s}_t(X_t) + \frac{1}{\sqrt{1 - \overline{\alpha}_t}} W \right\|_2^2 \right]. \tag{8}$$

Here, given the observed $X_t = \sqrt{\overline{\alpha}_t} X_0 + \sqrt{1 - \overline{\alpha}_t} \, W$, one seeks to predict the independent noise $W$, a strategy known as score matching. This formulation is particularly useful for practical training since it does not require explicit knowledge of the score function $\nabla \log p_{X_t}$. Instead, it can be approximated using finite samples, making it more feasible for learning the score function from data.

**The DDPM sampling procedure.** To implement the sampling process, we must discretize the continuous dynamics and obtain score estimates at discrete time steps. Suppose we have obtained score estimates $\{\widehat{s}_t(\cdot)\}$ at $t = 1, \ldots, T$. With these score estimates in hand, the renown DDPM algorithm (Ho et al., 2020) serves as a stochastic sampler that recursively generates samples using the following update rule. Starting from $Y_T \sim \mathcal{N}(0, I_d)$, DDPM computes $Y_{t-1}$ via

$$Y_{t-1} = \frac{1}{\sqrt{\alpha_t}}\big(Y_t + (1 - \alpha_t)\widehat{s}_t(Y_t)\big) + \sqrt{1 - \alpha_t} \, Z_t, \tag{9}$$

for $t = T, \ldots, 2$. Here, $Z_2, \ldots, Z_T \overset{\text{i.i.d.}}{\sim} \mathcal{N}(0, I_d)$ is a sequence of i.i.d. standard Gaussian random vectors in $\mathbb{R}^d$ that is independent of $(Y_t)_{t=2}^T$. In words, at each step, $Y_{t-1}$ is obtained as a weighted combination of $Y_t$ and its estimated score, with an addition of an independent Gaussian noise. Finally, the DDPM algorithm returns $Y_1$ as the final sample.

# 3. Main results

In this section, we state our main results on the performances of DDPM when applied to distributions that can be well-approximated by GMMs (1) and discuss their consequences. Without loss of generality, we focus on the case where $\sigma = 1$ and therefore the covariance of each component is the identity matrix. Otherwise, our algorithm and analysis framework are readily extended to general $\sigma$ by either rescaling the data or adjusting the learning rates accordingly.

We start by introducing some assumptions on the target distribution and the quality of our score estimates.

**Assumption 1.** There exists a Gaussian mixture model with $K$ components such that that target distribution $p_{X_0}$ is close to a GMM in TV distance, namely,

$$\mathsf{TV}(p_{X_0}, p_{X_0^{\mathsf{GMM}}}) \leq \varepsilon_{\mathsf{apprx}}$$

$$\text{with} \ \ X_0^{\mathsf{GMM}} \sim \sum_{k=1}^{K} \pi_k \mathcal{N}(\mu_k, I_d). \tag{10}$$

In addition, the components of the GMM satisfies

$$\max_{k \in [K]} \|\mu_k\|_2 \leq T^{c_R}, \tag{11}$$

for some absolute constant $c_R > 0$.

Here, the $\ell_2$ norm of the mean of each component is required to grow at most polynomially with the iteration number $T$. Given that the constant $c_R$ can be chosen arbitrarily large, this assumption allows each component to have exceedingly large mean value. Therefore, it holds true for most distributions that are encountered in practice.[2] Importantly, we do not impose any assumptions on the component separation $\min_{i \neq j} \|\mu_i - \mu_j\|_2$ or mixture weights $\{\pi_k\}$.

**Remark 1.** It is important to emphasize that our results stated below rely solely on the existence of such a GMM approximation $X_0^{\mathsf{GMM}}$ to the target distribution $X_0$. Crucially, there is no requirement to explicitly construct or estimate $X_0^{\mathsf{GMM}}$ in practice. Given the universal approximation capabilities of GMMs, this assumption is mild and widely applicable.

Next, we evaluate the quality of score estimates by their averaged $\ell_2$ accuracy. This form of estimation error matches naturally with training procedures such as the score matching mentioned above.

**Assumption 2.** Suppose the score estimates $\{s_t\}_{t \in [T]}$ satisfy

$$\frac{1}{T} \sum_{t=1}^{T} \varepsilon_{\mathsf{score},t}^2 \leq \varepsilon_{\mathsf{score}}^2, \tag{12a}$$

---

[2]We adopt this assumption mainly for analysis convenience, and it can be further relaxed.

where we define

$$\varepsilon_{\mathsf{score},t}^2 := \mathbb{E}_{X_t \sim p_{X_t}} \left[ \left\| s_t(X_t) - s_t^{\star}(X_t) \right\|_2^2 \right] \tag{12b}$$

for each $t \in [T]$.

Notably, this assumption requires the mean squared estimation error averaged over time steps is bounded, rather than the error at any individual step. This is commonly assumed in the literature of diffusion models (e.g., (Chen et al., 2022; Benton et al., 2023; Li & Yan, 2024b)).

In implementing DDPM as defined in (9), with the access to a sequence of score estimates $\{s_t\}_{t=1}^T$, we use the truncated score $\{\widehat{s}_t\}_{t=1}^T$ given by $\widehat{s}_t := \mathsf{clip}\{s_t\}$ for every $t \in [T]$. Here, the truncation function $\mathsf{clip}\{x\} : \mathbb{R}^d \to \mathbb{R}^d$ is defined as

$$\mathsf{clip}\{x\} := \begin{cases} x, & \text{if } \|x\|_2 \leq C_{\mathsf{clip}} \sqrt{\frac{d \log(dT)}{1 - \overline{\alpha}_t}}, \\ 0, & \text{otherwise,} \end{cases} \tag{13}$$

with $C_{\mathsf{clip}} > 0$ being a sufficiently large absolute constant. We remark that this truncation step is only introduced for technical convenience of our analysis, and can be removed with a similar but more refined argument.

**Convergence theory for DDPM.** Finally, let us specify the learning rate schedule $\{\alpha_t\}_{t \in [T]}$. As adopted in previous works on diffusion models (e.g. (Li & Cai, 2024)), the learning rate sequence is defined iteratively using the cumulative products $\overline{\alpha}_t = \prod_{k=1}^t \alpha_k$ in (4). More specifically, define

$$\overline{\alpha}_T = \frac{1}{T^{c_0}}, \tag{14a}$$

$$\overline{\alpha}_{t-1} = \overline{\alpha}_t + c_1 \frac{\log T}{T} \overline{\alpha}_t (1 - \overline{\alpha}_t) \tag{14b}$$

for $t = T, \ldots, 2$, where $c_0, c_1 > 0$ are absolute constants such that $c_0, c_1$ are sufficiently large and $c_1/c_0 > 4$. As shown in Lemma 3 in Appendix, this choice of the learning rates yields the property that for $t \geq 2$,

$$1 - \alpha_t \leq \frac{1 - \alpha_t}{1 - \overline{\alpha}_t} \lesssim \frac{\log T}{T},$$

and

$$1 - \alpha_1 \leq T^{-c_1/4}.$$

With these assumptions and preparations, we are positioned to state our main result below. The proof of this result is provided in Section 4, with the proofs of auxiliary lemmas postponed to Appendix B.

**Theorem 1.** *Under Assumptions 1–2, the output $Y_1$ of the DDPM sampler (9) with the learning rate selected according to (14) satisfies*

$$\mathsf{TV}(X_0, Y_1) \lesssim \frac{\log^2(KT) \log^2 T}{T} + \varepsilon_{\mathsf{score}} \sqrt{\log T}$$

$$+ \sqrt{d\varepsilon_{\mathsf{apprx}}} \log^{3/2}(dT). \qquad (15)$$

As a special example, when the target distribution is exactly a GMM (i.e., $\varepsilon_{\mathsf{apprx}} = 0$), we have the following guarantees:

**Corollary 1.** *Suppose the target distribution is a GMM with its components satisfying* (11) *and the score estimates satisfy Assumption* 2. *The output* $Y_1$ *of the DDPM sampler* (9) *with the learning rate selected according to* (14) *obeys*

$$\mathsf{TV}(X_0, Y_1) \lesssim \frac{\log^2(KT) \log^2 T}{T} + \varepsilon_{\mathsf{score}} \sqrt{\log T}. \quad (16)$$

In a nutshell, Theorem 1 guarantees that for a target distribution that is close to an isotropic GMM, the sampling quality of DDPM, measured in TV distance, is governed by three components: the first accounts for the time discretization error arising from approximating the continuous SDE in (7) with a discrete procedure; the second component results from the score estimation error; the third component controls the approximation error of GMMs.

As a result, given access to perfect score estimates, it only takes DDPM no larger than $\widetilde{O}(1/\varepsilon)$, number of iterations to yield a sampler that is $\varepsilon$-close to the target distribution in terms of TV distance, provided that the target distribution is close to some GMM with $\varepsilon_{\mathsf{apprx}} = \widetilde{O}(\varepsilon^2/d)$. Notably, this iteration complexity is independent of both the ambient dimension $d$ and the number of components $K$ up to some logarithmic factors. In addition, our result is robust to score estimation error: the TV distance between our output distribution and the target distribution scales proportionally to the $\varepsilon_{\mathsf{score}}$, modulo logarithmic factor. This finding stands in sharp contrast to the common belief that diffusion models inherently require iteration complexity that scales with the dimension $d$.

Theorem 1 is established with respect to the TV distance between $X_0$ and $Y_1$, whereas, most theoretical results in diffusion models fail to directly handle TV distance due to technical reasons. More specifically, most prior works consider KL divergence which is a natural choice if Girsanov's theorem is invoked to handle the discrepancy between the forward process and the process when imperfect score functions are concerned. Noteworthily, a recent line of literature (e.g. (Li et al., 2023; Li & Yan, 2024b; Li et al., 2024b)) enriches the toolbox of analyzing diffusion models by providing a framework of directly working with TV distance.

**Remark 2.** We view the GMM approximation assumption as a structural condition, rather than a literal claim that real-world data distributions are exactly isotropic Gaussian mixtures. GMMs are a classical and flexible model for heterogeneous and multi-modal data, and thus capture an important practically relevant regime. Our isotropic setting is intended as a first step that isolates the core mechanism behind the dimension-free sampling phenomenon in the cleanest form.

The result is most meaningful for structured distributions, such as multi-cluster or effectively low-dimensional data, where a moderate-size GMM approximation is plausible. In contrast, for a general high-dimensional distribution, an isotropic GMM approximation may require exponentially many components, in which case our theorem does not imply a practical dimension-free guarantee.

**Intuition for efficient sampling of GMMs.** Let us provide some intuition on why one should expect an iteration number independent of both $d$ and $K$ up to logarithmic factors for GMMs. Consider an auxiliary forward process $(X_t^{\mathsf{GMM}})_{t=0}^T$ starting with the GMM $X_0^{\mathsf{GMM}} \sim \sum_{k=1}^K \pi_k \mathcal{N}(\mu_k, I_d)$ and

$$X_t^{\mathsf{GMM}} = \sqrt{\alpha_t} X_{t-1}^{\mathsf{GMM}} + \sqrt{1 - \alpha_t} W_t, \quad t = 1, 2, \ldots, T, \qquad (17)$$

where the learning rates $(\alpha_t)_{t=1}^T$ and Gaussian noise $(W_t)_{t=1}^T$ are defined the same as those in the original forward process (2). At time $t$, the distribution of this auxiliary forward process obeys

$$X_t^{\mathsf{GMM}} \sim \sum_{k=1}^K \pi_k \mathcal{N}(\sqrt{\overline{\alpha}_t} \mu_k, I_d). \qquad (18)$$

Regarding this process, define the Jacobian matrix $J_t(x)$ : $\mathbb{R}^d \rightarrow \mathbb{R}^{d \times d}$ of the score function $s_t^{\star\mathsf{GMM}}(x) := \nabla \log p_{X_t^{\mathsf{GMM}}}(x)$ with $J_t(x) := \frac{\partial}{\partial x} s_t^{\star\mathsf{GMM}}(x)$. By direct computation (as in (36) in Appendix), it satisfies

$$J_t(x) = -\overline{\alpha}_t \left\{ \left( \sum_{k=1}^K \pi_k^{(t)}(x) \mu_k \right) \left( \sum_{k=1}^K \pi_k^{(t)}(x) \mu_k \right)^\top - \sum_{k=1}^K \pi_k^{(t)}(x) \mu_k \mu_k^\top \right\} - I_d. \qquad (19)$$

Here, $\pi_k^{(t)}(x)$ denotes the probability of $x$ drawn from the component $k$ at time $t$ of the forward process:

$$\pi_k^{(t)}(x) := \frac{\pi_k \exp\left(-\frac{1}{2} \|x - \sqrt{\overline{\alpha}_t} \mu_k\|_2^2\right)}{\sum_{i=1}^K \pi_i \exp\left(-\frac{1}{2} \|x - \sqrt{\overline{\alpha}_t} \mu_i\|_2^2\right)}. \qquad (20)$$

We prove in Lemma 8 in Appendix that with high probability

$$\mathsf{tr}\left( I_d + J_t\left( X_t^{\mathsf{GMM}} \right) \right) \leq C_1 \log(KT), \qquad (21)$$

for some absolute constant $C_1$. This relation, which does not generally hold, is the key to our dimension-free iteration complexity for GMMs. Since the target distribution is well-approximated by the Gaussian mixture satisfying (21), it is reasonable to expect a similar dimension-free iteration complexity for the target distribution as well.

However, generalizing this result to general distributions that are close to GMMs without imposing stronger assumptions on the score estimation quality is non-trivial. The challenge lies in the fact that Assumption 2 is concerned the forward process starting from the target distribution, not the auxiliary GMMs. In our analysis, we carefully control the score estimation error regarding GMMs in terms of the error $\varepsilon_{\text{score}}^2$ in our Lemma 6, which relies heavily on the statistical properties of GMMs, and may be of independent interest.

**Remark 3.** Extending our analysis from isotropic GMMs to mixtures with general well-conditioned covariances is an important direction. While we believe that such an extension may be possible, it is not immediate. The isotropic setting plays a central role in our current analysis, where the score and its Jacobian have a particularly clean structure, with each mixture component contributing the same curvature in every direction. This symmetry allows us to control the Jacobian trace through the posterior mixture weights, which is the key mechanism underlying the dimension-free bound. For mixtures with general covariances, the Jacobian becomes direction-dependent, and the isotropic cancellation no longer applies directly. Even under uniform well-conditioning assumptions, one must still control how anisotropy interacts with the posterior component weights across the mixture. Thus, the main technical obstacle is the loss of isotropic structure that makes the trace bound tractable. We expect that a related structure-adaptive result may hold under additional assumptions and with new techniques, but establishing a dimension-free bound for general covariance mixtures remains an interesting open problem.

**Comparisons to prior literature.** Alongside the seminar work (Chen et al., 2022) and the follow-up works (e.g. (Lee et al., 2023; Chen et al., 2023a; Benton et al., 2023)), Theorem 1 investigates the algorithmic aspect of learning GMMs assuming efficient score estimation/matching. Among existing works, the most closely related to ours is (Liang et al., 2024b), which established an iteration complexity of $\widetilde{O}(d/\varepsilon^2)$ by analyzing the Lipschitz properties and second moments of GMMs. Compared to (Li & Yan, 2024b), which studied DDPM for general distributions under mild assumptions, and attained an $\widetilde{O}(d/\varepsilon)$ iteration complexity, (Liang et al., 2024b) did not demonstrate any adaptation of DDPM to GMMs. Our result, however, highlights the surprising adaptive property of diffusion models in this setting.

Beyond the algorithmic aspect, (Chen et al., 2024; Gatmiry et al., 2024) developed an end-to-end theory by leveraging piecewise polynomial regression for score estimation and integrating it with existing convergence results on DDPM. The runtime and sample complexity of the resulting algorithms scale quasi-polynomially with $K/\varepsilon$ or $\log(K/\varepsilon)$ depending on the covariance assumptions. Notably, the number of diffusion steps used in these two works still scales linearly

with $d$. Our result serves as a complementary contribution by isolating the component of the iteration complexity that is independent of both $d$ and $K$, up to a logarithmic factor.

## 4. Analysis

In this section, we highlight our proof strategies for deriving Theorem 1. The detailed proofs are deferred to Appendix.

### 4.1. Step 1: Constructing auxiliary processes

To facilitate our main analysis, we introduce several auxiliary processes below. These processes are constructed only for analysis purpose and are not used in our sampling algorithm. Due to the space limit, we only provide a high-level description here and defer the formal definitions to Appendix.

**Sequence $(Y_t^\star)_{t=0}^T$ using true scores.** We begin by constructing an auxiliary reverse process $(Y_t^\star)_{t=0}^T$ using the true score functions $\{s_t^{\star\text{GMM}}(\cdot)\}_{t=1}^T$ of the diffused GMMs $(X_t^{\text{GMM}})_{t=1}^T$: $Y_T^\star \sim \mathcal{N}(0, I_d)$ and for each $t = T, \ldots, 1$,

$$Y_{t-1}^\star := \frac{1}{\sqrt{\alpha_t}}\left(Y_t^\star + (1 - \alpha_t)s_t^{\star\text{GMM}}(Y_t^\star)\right) + \sqrt{1 - \alpha_t}Z_t,$$

where $Z_t \overset{\text{i.i.d.}}{\sim} \mathcal{N}(0, I_d)$ is a sequence of independent Gaussian random vectors.

**Sequence $(\overline{Y}_t^-, \overline{Y}_t)_{t=0}^T$.** Next, we introduce two auxiliary sequences $(\overline{Y}_t^-)_{t=0}^T$ and $(\overline{Y}_t)_{t=0}^T$ to capture the discretization error modulo some low probability event. The formal definitions of these sequences rely on an auxiliary sequence of high-probability events $\{\mathcal{E}_t\}_{t=1}^T$, whose definition can be found in (37) in Appendix. These sequences, together with $Y_T$ implemented practically, form a Markov chain with the following transition structure:

$$Y_T \to \overline{Y}_T^- \to \overline{Y}_T \to \overline{Y}_{T-1}^- \to \overline{Y}_{T-1} \to \cdots$$
$$\to \overline{Y}_1^- \to \overline{Y}_1 \to \overline{Y}_0^- \to \overline{Y}_0.$$

*Initialization.* For $t = T$, we define $\overline{Y}_T^- := Y_T$ if $Y_T \in \mathcal{E}_T$ and $\overline{Y}_T^- := \infty$ otherwise.

*Transition from $\overline{Y}_t^-$ to $\overline{Y}_t$.* For $t = T, \ldots, 0$, we define $\overline{Y}_t$ as follows: conditional on $\overline{Y}_t^- = y$,

$$\overline{Y}_t := \begin{cases} y, & \text{with prob. } p_{X_t^{\text{GMM}}}(y)/p_{\overline{Y}_t^-}(y) \wedge 1, \\ \infty, & \text{with prob. } 1 - \left\{ p_{X_t^{\text{GMM}}}(y)/p_{\overline{Y}_t^-}(y) \wedge 1 \right\}, \end{cases}$$

*Transition from $\overline{Y}_t$ to $\overline{Y}_{t-1}^-$.* For each $t = T, \ldots, 1$, we first draw a candidate sample

$$\widetilde{Y}_{t-1} := \frac{1}{\sqrt{\alpha_t}}\left(\overline{Y}_t + (1 - \alpha_t)s_t^{\star\text{GMM}}(\overline{Y}_t)\right) + \sqrt{1 - \alpha_t}W_t,$$

where $W_t \overset{\text{i.i.d.}}{\sim} \mathcal{N}(0, I_d)$ is a sequence of independent Gaussian vectors, and then define $\overline{Y}_{t-1}^- := \widetilde{Y}_{t-1}$ if $\overline{Y}_t \in \mathcal{E}_t$ and $\widetilde{Y}_{t-1} \in \mathcal{E}_t$, and otherwise $\overline{Y}_{t-1}^- := \infty$.

**Sequence $(\widehat{Y}_t^-, \widehat{Y}_t)_{t=0}^T$.** Finally, we introduce two additional auxiliary sequences $(\widehat{Y}_t^-)_{t=0}^T$ and $(\widehat{Y}_t)_{t=0}^T$, which are constructed following the same principles as $(\overline{Y}_t^-)_{t=0}^T$ and $(\overline{Y}_t)_{t=0}^T$, with one key difference: the transition from $\widehat{Y}_t$ to $\widehat{Y}_t^-$ is computed using estimated score functions $(s_t)_t$ rather than the true score functions of GMMs $(s_t^{\star\mathsf{GMM}})_t$.

**Error decomposition.** In view of triangle's inequality, we can upper bound the TV distance between $p_{Y_1}$ and $p_{X_0}$ by

$$
\begin{aligned}
\mathsf{TV}(p_{Y_1}, p_{X_0}) &\leq \mathsf{TV}(p_{Y_1}, p_{X_1^{\mathsf{GMM}}}) + \mathsf{TV}(p_{X_1^{\mathsf{GMM}}}, p_{X_0^{\mathsf{GMM}}}) \\
&\quad + \mathsf{TV}(p_{X_0^{\mathsf{GMM}}}, p_{X_0}) \\
&\leq \mathsf{TV}(p_{\overline{Y}_1}, p_{X_1^{\mathsf{GMM}}}) + \mathsf{TV}(p_{\overline{Y}_1}, p_{Y_1}) \\
&\quad + \mathsf{TV}(p_{X_1^{\mathsf{GMM}}}, p_{X_0^{\mathsf{GMM}}}) + \mathsf{TV}(p_{X_0^{\mathsf{GMM}}}, p_{X_0}) \\
&\leq \mathsf{TV}(p_{\overline{Y}_1}, p_{X_1^{\mathsf{GMM}}}) + \mathsf{TV}(p_{\overline{Y}_1}, p_{Y_1}) \\
&\quad + \mathsf{TV}(p_{X_1^{\mathsf{GMM}}}, p_{X_0^{\mathsf{GMM}}}) + \varepsilon_{\mathsf{apprx}}
\end{aligned} \tag{22}
$$

Here, the first term acts in the role of discretization error, as $\overline{Y}_t$ is defined using the true scores; the second term captures the error caused by imperfect score estimation; the last two terms arise from the GMM approximation. In the sequel, we control each term separately.

## 4.2. Step 2: Bounding discretization error

In order to control $\mathsf{TV}(p_{X_1^{\mathsf{GMM}}}, p_{\overline{Y}_1})$, let us first define function $\Delta_t(x) : \mathbb{R}^d \to \mathbb{R}$ for each $t = 1, \dots, T$:

$$
\Delta_t(x) := p_{X_t^{\mathsf{GMM}}}(x) - p_{\overline{Y}_t}(x), \quad \forall x \in \mathbb{R}^d. \tag{23}
$$

Applying the TV distance formula, we can express

$$
\mathsf{TV}(p_{X_1^{\mathsf{GMM}}}, p_{\overline{Y}_1}) = \int_{\mathbb{R}^d} \Delta_1(x) \, \mathrm{d}x. \tag{24}
$$

Thus, it is sufficient to bound $\int \Delta_1(x) \, \mathrm{d}x$, which shall be done using an inductive argument. We start with the base case, which is characterized by the Lemma 1 below. The proof can be found in Appendix B.1.

**Lemma 1.** *It satisfies that*

$$
\int_{\mathbb{R}^d} \Delta_T(x) \, \mathrm{d}x \lesssim T^{-3}. \tag{25}
$$

In addition, Lemma 2 below establishes the inductive relationship between $t$ and $t - 1$; with proof deferred to Appendix B.2.

**Lemma 2.** *For all $t = T, \dots, 1$, one has*

$$
\begin{aligned}
\int \Delta_{t-1}(x) \, \mathrm{d}x &- \int \Delta_t(x) \, \mathrm{d}x \\
&\lesssim (1 - \alpha_t)^2 \log^2(KT) + T^{-3}.
\end{aligned} \tag{26}
$$

Consequently, combining Lemmas 1–2 with (24) yields

$$
\mathsf{TV}(p_{X_1^{\mathsf{GMM}}}, p_{\overline{Y}_1}) \lesssim \frac{\log^2(KT) \log^2 T}{T}. \tag{27}
$$

## 4.3. Step 3: Relating to score estimation error

Next, we control the term $\mathsf{TV}(p_{\overline{Y}_1}, p_{Y_1})$, which can be shown to satisfy

$$
\begin{aligned}
\mathsf{TV}(p_{\overline{Y}_1}, p_{Y_1}) &\leq \sqrt{\mathsf{KL}(p_{\overline{Y}_1} \| p_{\widehat{Y}_1})} \\
&\quad + \mathsf{TV}(p_{X_1^{\mathsf{GMM}}}, p_{\overline{Y}_1}).
\end{aligned} \tag{28}
$$

The second term on the right hand side has been handled in (27), while the first term can be controlled as

$$
\mathsf{KL}(p_{\overline{Y}_1} \| p_{\widehat{Y}_1}) \leq (\varepsilon_{\mathsf{score}}^{\mathsf{GMM}})^2 \log T, \tag{29}
$$

where we define

$$
\varepsilon_{\mathsf{score}, t}^{\mathsf{GMM}} := \sqrt{\int_{\mathbb{R}^d} \mathbb{E}_{X \sim X_t^{\mathsf{GMM}}} \left[ \left\| \widehat{s}_t(X) - s_t^{\star\mathsf{GMM}}(X) \right\|_2^2 \right]};
$$
$$\tag{30a}$$

$$
\varepsilon_{\mathsf{score}}^{\mathsf{GMM}} := \sqrt{\frac{1}{T} \sum_{t=2}^T (1 - \overline{\alpha}_t) (\varepsilon_{\mathsf{score}, t}^{\mathsf{GMM}})^2}. \tag{30b}
$$

## 4.4. Step 4: Controlling GMM approximation error

It remains to control $\mathsf{TV}(p_{X_0^{\mathsf{GMM}}}, p_{X_1^{\mathsf{GMM}}})$ and $\varepsilon_{\mathsf{score}}^{\mathsf{GMM}}$. Since $\alpha_1$ is sufficiently close to 1, one can show that

$$
\mathsf{TV}(p_{X_0^{\mathsf{GMM}}}, p_{X_1^{\mathsf{GMM}}}) \leq T^{-1}. \tag{31}
$$

In addition, Lemma 6 in Appendix B.3 shows

$$
\varepsilon_{\mathsf{score}}^{\mathsf{GMM}} \lesssim \varepsilon_{\mathsf{score}} + \sqrt{d \varepsilon_{\mathsf{apprx}}} \log(dT) + T^{-1}. \tag{32}
$$

Finally, putting together (27)–(32) with (22) completes the proof of Theorem 1.

## 5. Discussion

In summary, this paper explores the effectiveness of diffusion models in learning distributions that can be well-approximated by GMMs and presents new theoretical insights on how diffusion models implicitly exploit data structure to achieve efficient sampling. While DDPM requires a

number of iterations that scale linearly with data dimension in the worst case, our main result unveils a surprising efficiency of DDPM: it only requires an iteration complexity of $\widetilde{O}(1/\varepsilon)$ to learn an $\varepsilon$-accurate distribution, independent of both the data dimension $d$ and the number of mixture components $K$, up to logarithmic factors. This result suggests that diffusion models can efficiently learn structured distributions even in ultra-high-dimensional settings.

Before concluding, we highlight several promising directions for future investigation. First, while this paper focuses on mixtures of spherical Gaussians, an important next step is to analyze the iteration complexity of DDPM when applied to more general cases, such as mixtures with well-conditioned but arbitrary covariances, as considered in (Chen et al., 2024). Additionally, in order to learn an $\varepsilon$-accurate distribution, it takes DDPM an iteration complexity of order $\widetilde{O}(1/\varepsilon)$. While we suspect that the $\varepsilon$ dependence can not be further improved, it would be interesting to derive a matching lower bound to rigorously confirm the sharpness of our result. Finally, our analysis primarily addresses the sampling phase, leaving open the question of how score estimation efficiency is affected by the structure of GMMs. It remains a crucial direction to establish an end-to-end theory that integrates both score learning and sampling and fully unleashes the potential of diffusion models adapting to low-dimensional structure.

## Ackowledgement

G. Li is supported in part by the Chinese University of Hong Kong Direct Grant for Research and the Hong Kong Research Grants Council ECS 2191363 and GRF 2131005. C. Cai is supported in part by the NSF grant DMS-2515333. Y. Wei is supported in part by the NSF grants CCF-2106778, CCF-2418156 and CAREER award DMS-2143215 and the Wharton Dean's Research Fund.

## Impact Statement

This paper presents work whose goal is to advance the field of diffusion model theory. There are many potential societal consequences of our work, none of which we feel must be specifically highlighted here.

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

## A. Analysis

In this section, we describe the detailed proof strategies for deriving Theorem 1.

### A.1. Preliminaries

Before proceeding with the main analysis, let us collect several key properties that shall be used in our later analysis.

To begin with, Lemma 3 below characterizes the behavior of the learning rates $(\alpha_t)_{t\in[T]}$ chosen in (14). The proof of this result can be found in Li & Cai (2024, Appendix B.1).

**Lemma 3.** *The learning rates $(\alpha_t)_{t\in[T]}$ specified in (14) satisfy that*

$$1 - \alpha_t \le c_1 \frac{\log T}{T}, \quad t = 2, \ldots, T, \tag{33a}$$

$$1 - \alpha_1 \le \frac{1}{T^{c_1/4}}, \tag{33b}$$

*where $c_1$ is defined in (14).*

Next, in light of Assumption 1 on the GMM and the forward process (2), it is straightforward to verify each $X_t^{\mathsf{GMM}}$ is another GMM. Concretely, we have

$$X_t^{\mathsf{GMM}} \sim \sum_{k=1}^{K} \pi_k \mathcal{N}\big(\sqrt{\overline{\alpha}_t}\mu_k, I_d\big),$$

with the density function given by

$$p_{X_t^{\mathsf{GMM}}}(x) = \sum_{k=1}^{K} \pi_k \frac{1}{(2\pi)^{d/2}} \exp\Big(-\frac{1}{2}\big\|x - \sqrt{\overline{\alpha}_t}\mu_k\big\|_2^2\Big). \tag{34}$$

Through direct computation, we can derive that its score function takes the following explicit form:

$$s_t^{\star\mathsf{GMM}}(x) := \nabla \log p_{X_t^{\mathsf{GMM}}}(x) = -\sum_{k=1}^{K} \pi_k^{(t)}(x)\big(x - \sqrt{\overline{\alpha}_t}\mu_k\big) = -x + \sqrt{\overline{\alpha}_t}\sum_{k=1}^{K} \pi_k^{(t)}(x)\mu_k, \tag{35}$$

where we recall in Eq. (20) that $\pi_k^{(t)}(x) : \mathbb{R}^d \to [0, 1]$ is defined as

$$\pi_k^{(t)}(x) := \frac{\pi_k \exp\big(-\frac{1}{2}\|x - \sqrt{\overline{\alpha}_t}\mu_k\|_2^2\big)}{\sum_{i=1}^{K} \pi_i \exp\big(-\frac{1}{2}\|x - \sqrt{\overline{\alpha}_t}\mu_i\|_2^2\big)}, \quad \forall k \in [K], \ t \in [T].$$

In addition, the Jacobian matrix $J_t(x) : \mathbb{R}^d \to \mathbb{R}^{d\times d}$ of $s_t^{\star\mathsf{GMM}}(x)$ can be computed as

$$J_t(x) := \frac{\partial s_t^{\star\mathsf{GMM}}(x)}{\partial x} = -I_d + \overline{\alpha}_t \sum_{k=1}^{K} \pi_k^{(t)}\Big(\mu_k - \sum_{i=1}^{K} \pi_i^{(t)}\mu_i\Big)\Big(\mu_k - \sum_{i=1}^{K} \pi_i^{(t)}\mu_i\Big)^{\top}$$

$$= -I_d + \overline{\alpha}_t \bigg\{ \sum_{k=1}^{K} \pi_k^{(t)}\mu_k\mu_k^{\top} - \Big(\sum_{k=1}^{K} \pi_k^{(t)}\mu_k\Big)\Big(\sum_{k=1}^{K} \pi_k^{(t)}\mu_k\Big)^{\top} \bigg\}. \tag{36}$$

As a remark, we note that $I_d + J_t(x) \succeq 0$ for any $t \in [T]$ and $x \in \mathbb{R}^d$.

Next, we introduce the event $\mathcal{E}_t$ for each $t \in [T]$ as follows:

$$\mathcal{E}_t := \bigg\{ x \in \mathbb{R}^d : \mathsf{tr}\big(I_d + J_t(x)\big) \le C_1 \log(KT) \quad \text{and}$$

$$\sum_{k=1}^{K} \pi_k^{(t)} \exp\Big(-\zeta_k^{(t)}(x)\Big) \le \exp\Big(C_2(1 - \alpha_t)^2 \log^2(KT)\Big) \bigg\}, \tag{37}$$

for some absolute constants $C_1, C_2 > 0$, where we further define $\zeta_k^{(t)}(x) : \mathbb{R}^d \to \mathbb{R}$ for each $k \in [K]$:

$$\zeta_k^{(t)}(x) := \frac{1 - \alpha_t^2}{2\alpha_t^2}\left(\left\|x - \sqrt{\overline{\alpha}_t}\mu_k\right\|_2^2 - \sum_{i=1}^K \pi_i^{(t)}\left\|x - \sqrt{\overline{\alpha}_t}\mu_i\right\|_2^2\right) + \frac{1 - \alpha_t}{\alpha_t^2}s_t^\star(x)^\top \sum_{i=1}^K \pi_i^{(t)}\sqrt{\overline{\alpha}_t}(\mu_i - \mu_k). \quad (38)$$

Finally, we extend the $d$-dimensional Euclidean space $\mathbb{R}^d$ by adding a single point $\infty$, to obtain $\mathbb{R}^d \cup \{\infty\}$. Intuitively, this set $\{\infty\}$ serves as a convenient way to capture all atypical points in the reverse process.

### A.2. Step 1: Constructing auxiliary processes

To facilitate our main analysis, we introduce several auxiliary processes below. These processes are constructed only for analysis purpose and are not used in our sampling algorithm.

**Sequence** $(Y_t^\star)_{t=0}^T$ **using true scores.** We begin by constructing an auxiliary reverse process $(Y_t^\star)_{t=0}^T$ using the true score functions $\{s_t^{\star\mathsf{GMM}}(\cdot)\}_{t=1}^T$ of the diffused GMMs $(X_t^\mathsf{GMM})_{t=1}^T$:

$$Y_T^\star \sim \mathcal{N}(0, I_d), \quad Y_{t-1}^\star := \frac{1}{\sqrt{\alpha_t}}\big(Y_t^\star + (1 - \alpha_t)s_t^{\star\mathsf{GMM}}(Y_t^\star)\big) + \sqrt{1 - \alpha_t}Z_t; \quad t = T, \ldots, 1. \quad (39)$$

where $Z_t \overset{\text{i.i.d.}}{\sim} \mathcal{N}(0, I_d)$ is a sequence of i.i.d. standard $d$-dimensional Gaussian random vectors independent of $(Y_t^\star)_{t=0}^T$.

**Sequence** $(\overline{Y}_t^-, \overline{Y}_t)_{t=0}^T$. Next, we introduce two auxiliary sequences $(\overline{Y}_t^-)_{t=0}^T$ and $(\overline{Y}_t)_{t=0}^T$ to capture the discretization error modulo some low probability event. These sequences, together with $Y_T$ implemented practically, form a Markov chain with the following transition structure:

$$Y_T \to \overline{Y}_T^- \to \overline{Y}_T \to \overline{Y}_{T-1}^- \to \overline{Y}_{T-1} \to \cdots \to \overline{Y}_1^- \to \overline{Y}_1 \to \overline{Y}_0^- \to \overline{Y}_0. \quad (40)$$

- *Initialization.* For $t = T$, we define

$$\overline{Y}_T^- := \begin{cases} Y_T, & \text{if } Y_T \in \mathcal{E}_T, \\ \infty, & \text{otherwise.} \end{cases} \quad (41a)$$

The density of $\overline{Y}_T^-$ satisfies

$$p_{\overline{Y}_T^-}(y) = p_{Y_T}(y)\mathbb{1}\{y \in \mathcal{E}_T\} + \mathbb{P}\{Y_T \notin \mathcal{E}_T\}\delta_\infty(y). \quad (41b)$$

- *Transition from* $\overline{Y}_t^-$ *to* $\overline{Y}_t$. For $t = T, \ldots, 0$, we define $\overline{Y}_t$ as follows: conditional on $\overline{Y}_t^- = y$,

$$\overline{Y}_t := \begin{cases} y, & \text{with prob. } \overline{p}_t(y)/p_{\overline{Y}_t^-}(y) \wedge 1, \\ \infty, & \text{with prob. } 1 - \{\overline{p}_t(y)/p_{\overline{Y}_t^-}(y) \wedge 1\}, \end{cases} \quad (42a)$$

where we denote

$$\overline{p}_t := p_{X_t^\mathsf{GMM}}, \quad \forall t \geq 0. \quad (42b)$$

The conditional density of $\overline{Y}_t$ given $\overline{Y}_t^- = y$ obeys

$$p_{\overline{Y}_t|\overline{Y}_t^-}(x \mid y) = \{\overline{p}_t(y)/p_{\overline{Y}_t^-}(y) \wedge 1\}\delta_y(x) + \big(1 - \{\overline{p}_t(y)/p_{\overline{Y}_t^-}(y) \wedge 1\}\big)\delta_\infty(x). \quad (42c)$$

We make a critical implication of the above construction: for any $t \geq 0$, the density of $\overline{Y}_t$ satisfies

$$p_{\overline{Y}_t}(y) = \{\overline{p}_t(y)/p_{\overline{Y}_t^-}(y) \wedge 1\}p_{\overline{Y}_t^-}(y) = p_{X_t^\mathsf{GMM}}(y) \wedge p_{\overline{Y}_t^-}(y), \quad \forall y \in \mathbb{R}^d. \quad (43)$$

- *Transition from $\overline{Y}_t$ to $\overline{Y}_{t-1}$.* For each $t = T, \ldots, 1$, we first draw a candidate sample

$$\widetilde{Y}_{t-1} := \frac{1}{\sqrt{\alpha_t}}\left(\overline{Y}_t + (1-\alpha_t)s_t^{\star\mathsf{GMM}}(\overline{Y}_t)\right) + \sqrt{1-\alpha_t}W_t, \tag{44a}$$

where $W_t \overset{\text{i.i.d.}}{\sim} \mathcal{N}(0, I_d), t \geq 1$ is a sequence of i.i.d. standard Gaussian random vectors independent of $(Z_t)_{t=1}^T$, and then define

$$\overline{Y}_{t-1}^- := \begin{cases} \widetilde{Y}_{t-1}, & \text{if } \overline{Y}_t \in \mathcal{E}_t \text{ and } \widetilde{Y}_{t-1} \in \mathcal{E}_t, \\ \infty, & \text{otherwise.} \end{cases} \tag{44b}$$

The conditional density of $\overline{Y}_{t-1}^-$ given $\overline{Y}_t = y$ satisfies: if $y \in \mathcal{E}_t$, then

$$p_{\overline{Y}_{t-1}^-|\overline{Y}_t}(x \mid y) = p_{Y_{t-1}^\star|Y_t^\star}(x \mid y)\mathbb{1}\{x \in \mathcal{E}_t\} + \mathbb{P}\{Y_{t-1}^\star \notin \mathcal{E}_t \mid Y_t^\star = y\}\delta_\infty(x); \tag{44c}$$

otherwise,

$$p_{\overline{Y}_{t-1}^-|\overline{Y}_t}(x \mid y) = \delta_\infty(x). \tag{44d}$$

**Sequence $(\widehat{Y}_t^-, \widehat{Y}_t)_{t=0}^T$.** Finally, we introduce two additional auxiliary sequences $(\widehat{Y}_t^-)_{t=0}^T$ and $(\widehat{Y}_t)_{t=0}^T$, which forms the following Markov chain together with $Y_T$:

$$Y_T \to \widehat{Y}_T^- \to \widehat{Y}_T \to \widehat{Y}_{T-1}^- \to \widehat{Y}_{T-1} \to \cdots \to \widehat{Y}_1^- \to \widehat{Y}_1 \to \widehat{Y}_0^- \to \widehat{Y}_0. \tag{45}$$

- *Initialization.* For $t = T$, we initialize $\widehat{Y}_T^- = \overline{Y}_T^-$.
- *Transition from $\widehat{Y}_t^-$ to $\widehat{Y}_t$.* For $t = T, \ldots, 0$, the conditional density of $\widehat{Y}_t$ given $\widehat{Y}_t^- = y$ obeys

$$p_{\widehat{Y}_t|\widehat{Y}_t^-}(x \mid y) = p_{\overline{Y}_t|\overline{Y}_t^-}(x \mid y). \tag{46}$$

- *Transition from $\widehat{Y}_t$ to $\widehat{Y}_{t-1}^-$.* For $t = T, \ldots, 1$, the conditional density of $\overline{Y}_{t-1}^-$ given $\overline{Y}_t = y$ satisfies: if $y \in \mathcal{E}_t$, then

$$p_{\widehat{Y}_{t-1}^-|\widehat{Y}_t}(x \mid y) = p_{Y_{t-1}|Y_t}(x \mid y)\mathbb{1}\{x \in \mathcal{E}_t\} + \mathbb{P}\{Y_{t-1} \notin \mathcal{E}_t \mid Y_t = y\}\delta_\infty(x); \tag{47a}$$

otherwise,

$$p_{\widehat{Y}_{t-1}^-|\widehat{Y}_t}(x \mid y) = \delta_\infty(x). \tag{47b}$$

The sequences $(\widehat{Y}_t^-)_{t=0}^T$ and $(\widehat{Y}_t)_{t=0}^T$ are constructed following the same principles as $(\overline{Y}_t^-)_{t=0}^T$ and $(\overline{Y}_t)_{t=0}^T$, with one key difference: the transition from $\widehat{Y}_t$ to $\widehat{Y}_t^-$ is computed using estimated score functions rather than the true score functions.

**A crucial property.** It is noteworthy that for any $t \geq 0$, the density of $\widehat{Y}_t$ satisfies

$$p_{\widehat{Y}_t}(x) \leq p_{Y_t}(x), \quad \forall x \in \mathbb{R}^d, \tag{48}$$

and consequently, $p_{\widehat{Y}_t}(x) \geq p_{Y_t}(x)$ for $x = \infty$. To see this, we first note that the base case $t = T$ holds since $\widehat{Y}_t \overset{\mathrm{d}}{=} Y_t$, which arises from $\widehat{Y}_T^- = \overline{Y}_T^-$ and $p_{\widehat{Y}_t|\widehat{Y}_t^-} = p_{\overline{Y}_t|\overline{Y}_t^-}$ by (46). Next, suppose that (48) holds for $t+1$. Then for any $x \in \mathbb{R}^d$, one has

$$p_{\widehat{Y}_t}(x) \overset{\text{(i)}}{=} \{p_{X_t^{\mathsf{GMM}}}(x)/p_{\overline{Y}_t^-}(x) \wedge 1\}p_{\widehat{Y}_t^-}(x) \leq p_{\widehat{Y}_t^-}(x) = \int_{\mathbb{R}^d} p_{\widehat{Y}_t^-|\widehat{Y}_{t+1}}(x \mid y)p_{\widehat{Y}_{t+1}}(y)\,\mathrm{d}y$$

$$\overset{\text{(ii)}}{\leq} \int_{\mathbb{R}^d} p_{Y_t|Y_{t+1}}(x \mid y)p_{Y_{t+1}}(y)\,\mathrm{d}y = p_{Y_t}(x),$$

where (i) uses (46) and (42c); (ii) is true due to the induction hypothesis and (47a).

**Error decomposition.** In view of triangle's inequality, we can upper bound the TV distance between $p_{Y_1}$ and $p_{X_0}$ by

$$
\begin{aligned}
\mathsf{TV}\big(p_{Y_1}, p_{X_0}\big) &\le \mathsf{TV}\big(p_{Y_1}, p_{X_1^{\mathsf{GMM}}}\big) + \mathsf{TV}\big(p_{X_1^{\mathsf{GMM}}}, p_{X_0^{\mathsf{GMM}}}\big) + \mathsf{TV}\big(p_{X_0^{\mathsf{GMM}}}, p_{X_0}\big) \\
&\le \mathsf{TV}\big(p_{\overline{Y}_1}, p_{X_1^{\mathsf{GMM}}}\big) + \mathsf{TV}\big(p_{\overline{Y}_1}, p_{Y_1}\big) + \mathsf{TV}\big(p_{X_1^{\mathsf{GMM}}}, p_{X_0^{\mathsf{GMM}}}\big) + \mathsf{TV}\big(p_{X_0^{\mathsf{GMM}}}, p_{X_0}\big) \\
&= \mathsf{TV}\big(p_{\overline{Y}_1}, p_{X_1^{\mathsf{GMM}}}\big) + \mathsf{TV}\big(p_{\overline{Y}_1}, p_{Y_1}\big) + \mathsf{TV}\big(p_{X_1^{\mathsf{GMM}}}, p_{X_0^{\mathsf{GMM}}}\big) + \varepsilon_{\mathsf{apprx}}
\end{aligned}
\tag{49}
$$

where the last line leverages Assumption 1. Here, the first term acts in the role of discretization error modulo some low probability event, as $\overline{Y}_t$ is defined using the true scores; the second term captures the error caused by imperfect score estimation; the last two terms arise from the GMM approximation. In the sequel, we control each term separately.

### A.3. Step 2: Bounding discretization error

In this section, we proceed to bound $\mathsf{TV}(p_{X_1^{\mathsf{GMM}}}, p_{\overline{Y}_1})$. Let us first define function $\Delta_t(x) : \mathbb{R}^d \to \mathbb{R}$, where for each $t = 1, \ldots, T$:

$$
\Delta_t(x) \coloneqq p_{X_t^{\mathsf{GMM}}}(x) - p_{\overline{Y}_t}(x), \quad \forall x \in \mathbb{R}^d.
\tag{50}
$$

In view of relation (43), one has $\Delta_t(x) \ge 0$ for all $t \ge 0$ and $x \in \mathbb{R}^d$. Applying the formula for the total variation $\mathsf{TV}(p, q) = \int_{x: \, p(x) > q(x)} \big(p(x) - q(x)\big) \, \mathrm{d}x$, we find that

$$
\mathsf{TV}\big(p_{X_1^{\mathsf{GMM}}}, p_{\overline{Y}_1}\big) = \int_{\mathbb{R}^d \cup \{\infty\}} \big(p_{X_1^{\mathsf{GMM}}}(x) - p_{\overline{Y}_1}(x)\big) \mathbb{1}\big\{p_{X_1^{\mathsf{GMM}}}(x) > p_{\overline{Y}_1}(x)\big\} \, \mathrm{d}x = \int_{\mathbb{R}^d} \Delta_1(x) \, \mathrm{d}x.
\tag{51}
$$

Thus, it is sufficient to bound $\int \Delta_1(x) \, \mathrm{d}x$, which shall be done using an inductive argument. We start with the base case, which is characterized by the Lemma 4 below.

**Lemma 4.** *It satisfies that*

$$
\int_{\mathbb{R}^d} \Delta_T(x) \, \mathrm{d}x \lesssim T^{-3}.
\tag{52}
$$

*Proof.* See Appendix B.1. $\qquad\square$

In addition, Lemma 5 below establishes the inductive relationship between $t$ and $t - 1$.

**Lemma 5.** *For all $t = T, \ldots, 1$, one has*

$$
\int_{\mathbb{R}^d} \Delta_{t-1}(x) \, \mathrm{d}x - \int_{\mathbb{R}^d} \Delta_t(x) \, \mathrm{d}x \lesssim (1 - \alpha_t)^2 \log^2(KT) + T^{-3},
\tag{53}
$$

*Proof.* See Appendix B.2. $\qquad\square$

Consequently, combining (52)–(53) with (51) leads to

$$
\begin{aligned}
\mathsf{TV}\big(p_{X_1^{\mathsf{GMM}}}, p_{\overline{Y}_1}\big) = \int_{\mathbb{R}^d} \Delta_1(x) \, \mathrm{d}x &\le \int_{\mathbb{R}^d} \Delta_T(x) \, \mathrm{d}x + T \cdot O\big((1 - \alpha_t)^2 \log^2(KT)\big) + T \cdot O\big(T^{-3}\big) \\
&\lesssim \frac{1}{T^3} + \frac{\log^2(KT) \log^2 T}{T} + \frac{1}{T^2} \\
&\asymp \frac{\log^2(KT) \log^2 T}{T},
\end{aligned}
\tag{54}
$$

where the penultimate step uses $1 - \alpha_t \lesssim \log T / T$ by (33).

**A.4. Step 3: Relating to score estimation error**

Next, we control the term $\mathsf{TV}\big(p_{\overline{Y}_1}, p_{Y_1}\big)$. First, in view of basic calculations, we can write

$$
\begin{aligned}
\mathsf{TV}\big(p_{\overline{Y}_1}, p_{Y_1}\big) &= \int_{\mathbb{R}^d} \big(p_{\overline{Y}_1}(x) - p_{Y_1}(x)\big) \mathbb{1}\big\{p_{\overline{Y}_1}(x) > p_{Y_1}(x)\big\}\, \mathrm{d}x + \mathbb{P}\{\overline{Y}_1 = \infty\} \\
&\overset{(i)}{\leq} \int_{\mathbb{R}^d} \big(p_{\overline{Y}_1}(x) - p_{\widehat{Y}_1}(x)\big) \mathbb{1}\big\{p_{\overline{Y}_1}(x) > p_{\widehat{Y}_1}(x)\big\}\, \mathrm{d}x + \mathbb{P}\{\overline{Y}_1 = \infty\} \\
&\overset{(ii)}{\leq} \mathsf{TV}\big(p_{\overline{Y}_1}, p_{\widehat{Y}_1}\big) + \mathsf{TV}\big(p_{X_1^{\mathsf{GMM}}}, p_{\overline{Y}_1}\big) \\
&\overset{(iii)}{\leq} \sqrt{\mathsf{KL}\big(p_{\overline{Y}_1} \,\|\, p_{\widehat{Y}_1}\big)} + O\bigg(\frac{\log^2(KT)\log^2 T}{T}\bigg).
\end{aligned}
\tag{55}
$$

where (i) arises from (48) that $p_{Y_1}(x) \geq p_{\widehat{Y}_1}(x)$ for any $x \in \mathbb{R}^d$; (ii) uses $\mathbb{P}\{\overline{Y}_1 = \infty\} \leq \mathsf{TV}(p_{X_1^{\mathsf{GMM}}}, p_{\overline{Y}_1})$ since $X_1^{\mathsf{GMM}} \in \mathbb{R}^d$; (iii) applies Pinsker's inequality and (54).

To further control the right hand side of (55), it suffices to bound $\mathsf{KL}\big(p_{\overline{Y}_1} \,\|\, p_{\widehat{Y}_1}\big)$ from above. Towards this, notice that

$$
\begin{aligned}
\mathsf{KL}\big(p_{\overline{Y}_1} \,\|\, p_{\widehat{Y}_1}\big) &\overset{(i)}{\leq} \mathsf{KL}\big(p_{\overline{Y}_T^-, \overline{Y}_T, \ldots, \overline{Y}_1^-, \overline{Y}_1} \,\|\, p_{\widehat{Y}_T^-, \widehat{Y}_T, \ldots, \widehat{Y}_1^-, \widehat{Y}_1}\big) \\
&\overset{(ii)}{=} \mathsf{KL}\big(p_{\overline{Y}_T^-} \,\|\, p_{\widehat{Y}_T^-}\big) + \sum_{t=1}^{T} \mathbb{E}_{x_t \sim p_{\overline{Y}_t^-}}\bigg[\mathsf{KL}\big(p_{\overline{Y}_t | \overline{Y}_t^- = x_t} \,\|\, p_{\widehat{Y}_t | \widehat{Y}_t^- = x_t}\big)\bigg] \\
&\quad + \sum_{t=2}^{T} \mathbb{E}_{x_t \sim p_{\overline{Y}_t}}\bigg[\mathsf{KL}\big(p_{\overline{Y}_{t-1}^- | \overline{Y}_t = x_t} \,\|\, p_{\widehat{Y}_{t-1}^- | \widehat{Y}_t = x_t}\big)\bigg] \\
&\overset{(iii)}{=} \sum_{t=2}^{T} \mathbb{E}_{x_t \sim p_{\overline{Y}_t}}\bigg[\mathsf{KL}\big(p_{\overline{Y}_{t-1}^- | \overline{Y}_t = x_t} \,\|\, p_{\widehat{Y}_{t-1}^- | \widehat{Y}_t = x_t}\big)\bigg].
\end{aligned}
\tag{56}
$$

Here, (i) applies the data-processing inequality; (ii) uses the chain rule of KL divergence and the Markov property; (iii) is true since we initialize $\widehat{Y}_T^- = \overline{Y}_T^-$ and the transition kernels from $\widehat{Y}_t^-$ to $\widehat{Y}_t$ are the same as those from $\overline{Y}_t^-$ to $\overline{Y}_t$ for all $t \geq 1$.

Note that $Y_{t-1}^\star \mid Y_t^\star = x_t \sim \mathcal{N}\big(\frac{1}{\sqrt{\alpha_t}}(x_t + (1-\alpha_t)s_t^{\star\mathsf{GMM}}(x_t)), (1-\alpha_t)I_d\big)$ and $Y_{t-1} \mid Y_t = x_t \sim \mathcal{N}\big(\frac{1}{\sqrt{\alpha_t}}(x_t + (1-\alpha_t)\widehat{s}_t(x_t)), (1-\alpha_t)I_d\big)$. For any $x_t \in \mathcal{E}_t$, write $p(\cdot) = p_{Y_{t-1}^\star | Y_t^\star = x_t}(\cdot)$ and $q(\cdot) = p_{Y_{t-1} | Y_t = x_t}(\cdot)$. In view of the definitions (47a) and (42c), one has

$$
\mathsf{KL}\big(p_{\overline{Y}_{t-1}^- | \overline{Y}_t = x_t} \,\|\, p_{\widehat{Y}_{t-1}^- | \widehat{Y}_t = x_t}\big) = \int_{\mathcal{E}_t} p(x) \log \frac{p(x)}{q(x)}\, \mathrm{d}x + \log \frac{\int_{\mathcal{E}_t^c} p(x)\, \mathrm{d}x}{\int_{\mathcal{E}_t^c} q(x)\, \mathrm{d}x} \int_{\mathcal{E}_t^c} p(x)\, \mathrm{d}x.
\tag{57}
$$

Now invoke Li & Yan (2024b, Lemma 6) to derive

$$
\begin{aligned}
\mathsf{KL}\big(p_{\overline{Y}_{t-1}^- | \overline{Y}_t = x_t} \,\|\, p_{\widehat{Y}_{t-1}^- | \widehat{Y}_t = x_t}\big) &\leq \int_{\mathbb{R}^d} p(x) \log \frac{p(x)}{q(x)}\, \mathrm{d}x \\
&= \mathsf{KL}\big(p_{Y_{t-1}^\star | Y_t^\star = x_t} \,\|\, p_{Y_{t-1} | Y_t = x_t}\big) \\
&\overset{(i)}{=} \frac{1-\alpha_t}{2\alpha_t}\big\|\widehat{s}_t(x_t) - s_t^{\star\mathsf{GMM}}(x_t)\big\|_2^2 \\
&\overset{(ii)}{\lesssim} \frac{\log T}{T}(1-\overline{\alpha}_t)\big\|\widehat{s}_t(x_t) - s_t^{\star\mathsf{GMM}}(x_t)\big\|_2^2.
\end{aligned}
\tag{58}
$$

Here, (i) uses the formula of the KL divergence for two normal distributions; (ii) uses $1-\alpha_t \leq (1-\alpha_t)/(1-\overline{\alpha}_t) \lesssim \log T/T$ by (33). Meanwhile, for any $x_t \in \mathcal{E}_t^c$, we know from (47b) that

$$
\mathsf{KL}\big(p_{\overline{Y}_{t-1}^- | \overline{Y}_t = x_t} \,\|\, p_{\widehat{Y}_{t-1}^- | \widehat{Y}_t = x_t}\big) = 0.
\tag{59}
$$

Combined with (56), the above bounds give

$$
\begin{aligned}
\mathsf{KL}\big(p_{\overline{Y}_1} \,\|\, p_{\widehat{Y}_1}\big) &\overset{(i)}{\leq} \sum_{t=2}^{T} \mathbb{E}_{x_t \sim \overline{p}_t}\Big[\mathsf{KL}\big(p_{\overline{Y}_{t-1}^- | \overline{Y}_t = x_t} \,\|\, p_{\widehat{Y}_{t-1}^- | \widehat{Y}_t = x_t}\big)\Big] \\
&\overset{(ii)}{\lesssim} \frac{\log T}{T} \sum_{t=2}^{T} \int_{\mathbb{R}^d} (1 - \overline{\alpha}_t)\|\widehat{s}_t(x_t) - s_t^{\star\mathsf{GMM}}(x_t)\|_2^2\, p_{X_t^{\mathsf{GMM}}}(x_t)\,\mathrm{d}x_t \\
&\overset{(iii)}{=} \big(\varepsilon_{\mathsf{score}}^{\mathsf{GMM}}\big)^2 \log T,
\end{aligned}
\tag{60}
$$

where (i) arises from (59) and (43) that $p_{\overline{Y}_t}(x) \leq \overline{p}_t(x)$ for all $x \in \mathbb{R}^d$; (ii) uses (58); in (iii) we define

$$
\varepsilon_{\mathsf{score},t}^{\mathsf{GMM}} := \sqrt{\int_{\mathbb{R}^d} \big\|\widehat{s}_t(x_t) - s_t^{\star\mathsf{GMM}}(x_t)\big\|_2^2\, p_{X_t^{\mathsf{GMM}}}(x_t)\,\mathrm{d}x_t};
\tag{61a}
$$

$$
\varepsilon_{\mathsf{score}}^{\mathsf{GMM}} := \sqrt{\frac{1}{T} \sum_{t=2}^{T} (1 - \overline{\alpha}_t)\big(\varepsilon_{\mathsf{score},t}^{\mathsf{GMM}}\big)^2}.
\tag{61b}
$$

Substituting (60) into (55) leads to

$$
\mathsf{TV}\big(p_{Y_1}, p_{\overline{Y}_1}\big) \lesssim \frac{\log^2(KT)\log^2 T}{T} + \varepsilon_{\mathsf{score}}^{\mathsf{GMM}}\sqrt{\log T}.
\tag{62}
$$

### A.5. Step 4: Controlling GMM approximation error

Putting relations (54) and (62) together with (49) yields

$$
\mathsf{TV}\big(p_{X_0}, p_{Y_1}\big) \lesssim \frac{\log^2(KT)\log^2 T}{T} + \varepsilon_{\mathsf{score}}^{\mathsf{GMM}}\sqrt{\log T} + \mathsf{TV}\big(p_{X_0^{\mathsf{GMM}}}, p_{X_1^{\mathsf{GMM}}}\big) + \varepsilon_{\mathsf{apprx}}.
\tag{63}
$$

Hence, it remains to control $\mathsf{TV}\big(p_{X_0^{\mathsf{GMM}}}, p_{X_1^{\mathsf{GMM}}}\big)$ and the term $\varepsilon_{\mathsf{score}}^{\mathsf{GMM}}$ defined in (61).

We begin with $\mathsf{TV}\big(p_{X_0^{\mathsf{GMM}}}, p_{X_1^{\mathsf{GMM}}}\big)$. Since $X_1^{\mathsf{GMM}} = \sqrt{\alpha_1}X_0^{\mathsf{GMM}} + \sqrt{1-\alpha_1}\,W_1$ with $W_1 \sim \mathcal{N}(0, I_d)$ independent of $X_0^{\mathsf{GMM}}$, one knows

$$
X_1^{\mathsf{GMM}} \sim \sum_{k=1}^{K} \pi_k \mathcal{N}(\sqrt{\alpha_1}\mu_k, I_d).
$$

For each component, we invoke Pinsker's inequality and the KL divergence between Gaussian distributions to arrive

$$
\mathsf{TV}^2\big(\mathcal{N}(\mu_k, I_d), \mathcal{N}(\sqrt{\alpha_1}\mu_k, I_d)\big) \leq \frac{1}{2}\mathsf{KL}\big(\mathcal{N}(\mu_k, I_d)\,\|\,\mathcal{N}(\sqrt{\alpha_1}\mu_k, I_d)\big) \leq \frac{1}{2}(1 - \sqrt{\alpha_1})^2\|\mu_k\|_2^2 \leq \frac{1}{2}(1 - \alpha_1)^2\|\mu_k\|_2^2,
$$

where the last step holds as $\alpha_1 \in (0,1)$. It follow from the convexity of the TV distance and Jensen's inequality that

$$
\begin{aligned}
\mathsf{TV}\big(p_{X_0^{\mathsf{GMM}}}, p_{X_1^{\mathsf{GMM}}}\big) &\leq \sum_{k=1}^{K} \pi_k \mathsf{TV}\big(\mathcal{N}(\mu_k, I_d), \mathcal{N}(\sqrt{\alpha_1}\mu_k, I_d)\big) \\
&\lesssim (1 - \alpha_1)\sum_{k=1}^{K} \pi_k\|\mu_k\|_2 \lesssim T^{-c_1/4}T^{c_R} \leq T^{-1},
\end{aligned}
\tag{64}
$$

where the penultimate step arises from (33) in Lemma 3 and Assumption 1, and the last step holds as long as $c_1$ is large enough.

Now, it is only left for us to control the term $\varepsilon_{\mathsf{score}}^{\mathsf{GMM}}$. Towards this goal, it is important to notice that $\varepsilon_{\mathsf{score},t}^{\mathsf{GMM}}$ for $t \in [T]$ is defined with respect to the truncated score estimator $\widehat{s}_t = \mathsf{clip}\{s_t\}$ and $p_{X_t^{\mathsf{GMM}}}$, whereas Assumption 2 only considers the estimation error of $s_t$ with respect to $p_{X_t}$. To handle this discrepancy, we shall apply similar analyses as in Li & Cai (2024, Lemma 4) to bound $\varepsilon_{\mathsf{score}}^{\mathsf{GMM}}$ in terms of the score estimation error $\varepsilon_{\mathsf{score},t}$ defined in Assumption 2. The following lemma characterizes the relation between these two score estimation error and its proof is deferred to Appendix B.3.

**Lemma 6.** *The score error term $\varepsilon_{\text{score}}^{\text{GMM}}$ defined in* (61) *satisfies:*

$$\varepsilon_{\text{score}}^{\text{GMM}} \lesssim \varepsilon_{\text{score}} + \sqrt{d\varepsilon_{\text{apprx}}} \log(dT) + T^{-1}. \tag{65}$$

Substituting (64) and (65) into (63), we conclude that

$$\mathsf{TV}(p_{X_0}, p_{Y_1}) \lesssim \frac{\log^2(KT)\log^2 T}{T} + \sqrt{d\varepsilon_{\text{apprx}}} \log^{3/2}(dT) + \varepsilon_{\text{score}}\sqrt{\log T}.$$

This completes the proof of Theorem 1.

## B. Proof of technical lemmas

### B.1. Proof of Lemma 4

Recalling that $\Delta_T(x) := p_{X_T^{\text{GMM}}}(x) - p_{\overline{Y}_T}(x) \geq 0$ for any $x \in \mathbb{R}^d$, we can derive

$$
\begin{aligned}
\int_{\mathbb{R}^d} \Delta_T(x)\,\mathrm{d}x &= \int_{\mathbb{R}^d} \big(p_{X_T^{\text{GMM}}}(x) - p_{\overline{Y}_T}(x)\big)\mathbb{1}\big\{p_{X_T^{\text{GMM}}}(x) > p_{\overline{Y}_T}(x)\big\}\,\mathrm{d}x \\
&\overset{(i)}{=} \int_{\mathbb{R}^d} \big(p_{X_T^{\text{GMM}}}(x) - p_{\overline{Y}_T^-}(x)\big)\mathbb{1}\big\{p_{X_T^{\text{GMM}}}(x) > p_{\overline{Y}_T^-}(x)\big\}\,\mathrm{d}x \\
&\overset{(ii)}{=} \mathsf{TV}\big(p_{X_T^{\text{GMM}}}, p_{\overline{Y}_T^-}\big) \\
&\leq \mathsf{TV}\big(p_{X_T^{\text{GMM}}}, p_{Y_T}\big) + \mathsf{TV}\big(p_{Y_T}, p_{\overline{Y}_T^-}\big),
\end{aligned}
\tag{66}
$$

where (i) arises from (43) that $p_{\overline{Y}_T}(x) = p_{X_T^{\text{GMM}}}(x) \wedge p_{\overline{Y}_T^-}(x)$ for any $x \in \mathbb{R}^d$, (ii) uses the formula of the total variation $\mathsf{TV}(p, q) = \int_{x:\, p(x) > q(x)} \big(p(x) - q(x)\big)\,\mathrm{d}x$ and $X_T^{\text{GMM}} \in \mathbb{R}^d$.

Consequently, it suffices to control the two quantities in (66) respectively.

- For the first term $\mathsf{TV}\big(p_{X_T^{\text{GMM}}}, p_{Y_T}\big)$ corresponding to the initialization error, we can derive

$$
\begin{aligned}
\mathsf{KL}\big(p_{X_T^{\text{GMM}}} \parallel p_{Y_T}\big) &\overset{(i)}{\leq} \mathbb{E}\Big[\mathsf{KL}\big(p_{X_T^{\text{GMM}}}(\cdot \mid X_0^{\text{GMM}}) \parallel p_{Y_T}(\cdot)\big)\Big] \\
&\overset{(ii)}{=} \frac{1}{2}\mathbb{E}\Big[d(1 - \overline{\alpha}_T) - d + \big\|\sqrt{\overline{\alpha}_T}X_0^{\text{GMM}}\big\|_2^2 - d\log(1 - \overline{\alpha}_T)\Big] \\
&\overset{(iii)}{\lesssim} T^{-c_0}\mathbb{E}\Big[\big\|X_0^{\text{GMM}}\big\|_2^2\Big] \overset{(iv)}{\lesssim} T^{-c_0}(T^{c_R} + d) \overset{(v)}{\lesssim} T^{-8},
\end{aligned}
\tag{67}
$$

where (i) arises from the convexity of the KL divergence; (ii) applies the KL divergence formula for two normal distributions; (iii) is due to the choice of the learning rate $\overline{\alpha}_T = T^{-c_0} = o(1)$ in (14) and $\log(1 - x) \geq -x$ for any $x \in [0, 1/2]$; (iv) holds due to Assumption 1 that

$$\mathbb{E}\Big[\big\|X_0^{\text{GMM}}\big\|_2^2\Big] = \sum_{k=1}^K \pi_k\big(\|\mu_k\|_2^2 + d\big) \leq T^{c_R} + d;$$

and (v) holds as long as $T$ and $c_0$ are sufficiently large. It then follows from Pinsker's inequality that

$$\mathsf{TV}\big(p_{X_T^{\text{GMM}}} \parallel p_{Y_T}\big) \leq \sqrt{\mathsf{KL}\big(p_{X_T^{\text{GMM}}} \parallel p_{Y_T}\big)} \lesssim T^{-4}. \tag{68}$$

- We proceed to consider the second term $\mathsf{TV}\big(p_{Y_T}, p_{\overline{Y}_T^-}\big)$. By the construction of $\overline{Y}_T^-$ (see (41)), one can write

$$\mathsf{TV}\big(p_{Y_T}, p_{\overline{Y}_T^-}\big) \overset{(i)}{=} \int_{\mathbb{R}^d} \big(p_{Y_T}(x) - p_{\overline{Y}_T^-}(x)\big)\mathbb{1}\big\{p_{Y_T}(x) > p_{\overline{Y}_T^-}(x)\big\}\,\mathrm{d}x$$

$$\overset{\text{(ii)}}{=} \int_{\mathcal{E}_t^{\mathrm{c}}} p_{Y_T}(x)\,\mathrm{d}x$$

$$\overset{\text{(iii)}}{\leq} \int_{\mathcal{E}_t^{\mathrm{c}}} p_{X_T^{\mathsf{GMM}}}(x)\,\mathrm{d}x + \mathsf{TV}\big(p_{X_T^{\mathsf{GMM}}}, p_{Y_T}\big).$$

Here, (i) uses the formula of the total variation $\mathsf{TV}(p, q) = \int_{x:\, p(x)>q(x)}\big(p(x) - q(x)\big)\,\mathrm{d}x$; (ii) follows from $Y_T \in \mathbb{R}^d$ and (41b) that $p_{\overline{Y}_T^-}(x) = p_{Y_T}(x)$ if $x \in \mathcal{E}_T$ and $p_{\overline{Y}_T^-}(x) = 0$ if $x \in \mathbb{R}^d \setminus \mathcal{E}_T$; (iii) arises from the definition of total variation distance that $\mathsf{TV}(p, q) = \sup_B |p(B) - q(B)|$. As we shall see momentarily in Lemma 8 in Appendix B.5, one has

$$\int_{\mathcal{E}_T^{\mathrm{c}}} p_{X_T^{\mathsf{GMM}}}(x) \lesssim T^{-3}.$$

Combined with (68), this leads to

$$\mathsf{TV}\big(p_{Y_T}, p_{\overline{Y}_T^-}\big) \lesssim T^{-3} + T^{-4} \asymp T^{-3}. \tag{69}$$

**In conclusion,** substituting (68) and (69) into (66) completes the proof of Lemma 4.

### B.2. Proof of Lemma 5

Fix an arbitrary $t \in [T]$. To analyze $\int_{\mathbb{R}^d} \Delta_t(x_t)\,\mathrm{d}x_t$, let us first introduce a function $\Delta_{t \to t-1}(x) : \mathbb{R}^d \to \mathbb{R}$ where

$$\Delta_{t \to t-1}(x) := \int_{x_t \in \mathcal{E}_t} p_{Y_{t-1}^\star | Y_t^\star}(x \mid x_t)\Delta_t(x_t)\,\mathrm{d}x_t, \quad \forall x \in \mathbb{R}^d. \tag{70}$$

Note that in view of relation (43), $\Delta_t(x) \geq 0$ for all $x \in \mathbb{R}^d$ and therefore $\Delta_{t \to t-1}(x) \geq 0$ for all $x \in \mathbb{R}^d$. It is easily seen that

$$\int_{\mathbb{R}^d} \Delta_{t \to t-1}(x_{t-1})\,\mathrm{d}x_{t-1} = \int_{x_t \in \mathcal{E}_t} \int_{x_{t-1} \in \mathbb{R}^d} p_{Y_{t-1}^\star | Y_t^\star}(x_{t-1} \mid x_t)\,\mathrm{d}x_{t-1}\Delta_t(x_t)\,\mathrm{d}x_t \leq \int_{\mathbb{R}^d} \Delta_t(x_t)\,\mathrm{d}x_t. \tag{71}$$

As a result, to upper bound $\int_{\mathbb{R}^d} \Delta_{t-1}(x)\,\mathrm{d}x - \int_{\mathbb{R}^d} \Delta_t(x)\,\mathrm{d}x$, it is sufficient to consider $\int_{\mathbb{R}^d} \Delta_{t-1}(x)\,\mathrm{d}x - \int_{\mathbb{R}^d} \Delta_{t \to t-1}(x_{t-1})\,\mathrm{d}x_{t-1}$.

Towards this, we find it helpful to first make the following observation. For any $x_{t-1} \in \mathbb{R}$ such that $\Delta_{t-1}(x_{t-1}) > 0$, or equivalently, $p_{X_{t-1}^{\mathsf{GMM}}}(x_{t-1}) > p_{\overline{Y}_{t-1}}(x_{t-1})$, we have

$$p_{X_{t-1}}(x_{t-1}) - \Delta_{t-1}(x_{t-1}) + \Delta_{t \to t-1}(x_{t-1}) \geq p_{\overline{Y}_{t-1}^-}(x_{t-1}) + \Delta_{t \to t-1}(x_{t-1}). \tag{72}$$

Here, we use the fact that $p_{\overline{Y}_{t-1}^-}(x_{t-1}) = p_{X_{t-1}^{\mathsf{GMM}}}(x_{t-1}) \wedge p_{\overline{Y}_{t-1}^-}(x_{t-1})$ since $p_{X_{t-1}^{\mathsf{GMM}}}(x_{t-1}) > p_{\overline{Y}_{t-1}}(x_{t-1})$. To further control the right hand side, recall the definition of $\Delta_t(x)$ in (50) and the constructed transition kernel of $p_{\overline{Y}_{t-1}^- | \overline{Y}_t}$ in (44c). For any $x_{t-1} \in \mathbb{R}^d$, we arrive at

$$p_{\overline{Y}_{t-1}^-}(x_{t-1}) \geq \int_{x_t \in \mathcal{E}_t} p_{Y_{t-1}^\star | Y_t^\star}(x_{t-1} \mid x_t)p_{\overline{Y}_t}(x_t)\,\mathrm{d}x_t$$

$$= \int_{x_t \in \mathcal{E}_t} p_{Y_{t-1}^\star | Y_t^\star}(x_{t-1} \mid x_t)p_{X_t^{\mathsf{GMM}}}(x_t)\,\mathrm{d}x_t - \Delta_{t \to t-1}(x_{t-1}). \tag{73}$$

As a result, we obtain

$$p_{X_{t-1}^{\mathsf{GMM}}}(x_{t-1}) - \Delta_{t-1}(x_{t-1}) + \Delta_{t \to t-1}(x_{t-1})$$

$$\geq \int_{x_t \in \mathcal{E}_t} p_{Y_{t-1}^\star | Y_t^\star}(x_{t-1} \mid x_t)p_{X_t^{\mathsf{GMM}}}(x_t)\,\mathrm{d}x_t$$

$$\overset{\text{(i)}}{=} \int_{\mathcal{E}_t} \left(\frac{1}{2\pi(1-\alpha_t)}\right)^{d/2} \exp\left(-\frac{\|\sqrt{\alpha_t}x_{t-1} - x_t - (1-\alpha_t)s_t^\star(x_t)\|_2^2}{2\alpha_t(1-\alpha_t)}\right)p_{X_t^{\mathsf{GMM}}}(x_t)\,\mathrm{d}x_t$$

$$\overset{\text{(ii)}}{=} \int_{\mathcal{E}_t} \left(\frac{1}{2\pi(1-\alpha_t)}\right)^{d/2} \exp\left(-\frac{\|\sqrt{\alpha_t}x_{t-1}-u_t\|_2^2}{2\alpha_t(1-\alpha_t)}\right) \det\left(I_d + (1-\alpha_t)J_t(x_t)\right)^{-1} p_{X_t^{\mathsf{GMM}}}(x_t)\, \mathrm{d}u_t. \tag{74}$$

where (i) uses (73); (ii) arises from (39) that $Y_{t-1}^\star \mid Y_t^\star \sim \mathcal{N}\left(\alpha_t^{-1/2}[Y_t^\star + (1-\alpha_t)s_t^{\star\mathsf{GMM}}(Y_t^\star)], (1-\alpha_t)I_d\right)$; (ii) applies the change of variable

$$u_t := x_t + (1-\alpha_t)s_t^{\star\mathsf{GMM}}(x_t).$$

To bound the integral in (74), we present Lemma 7 below.

**Lemma 7.** *For any $t \in [T]$, the following holds for any $x_t \in \mathcal{E}_t$:*

$$\det\left(I_d + (1-\alpha_t)J_t(x_t)\right)^{-1} p_{X_t^{\mathsf{GMM}}}(x_t)$$
$$= \left(\frac{1}{2\pi\alpha_t^2}\right)^{d/2} \exp\left(O\left((1-\alpha_t)^2 \log^2(KT)\right)\right) \sum_{k=1}^{K} \pi_k \exp\left(-\frac{\|u_t - \sqrt{\overline{\alpha}_t}\mu_k\|^2}{2\alpha_t^2}\right). \tag{75}$$

*Proof.* See Appendix B.4. □

Plugging (75) into (74) and invoking the inequality $\exp(x) \geq 1 + x$ leads to

$$p_{X_{t-1}^{\mathsf{GMM}}}(x_{t-1}^{\mathsf{GMM}}) - \Delta_{t-1}(x_{t-1}) + \Delta_{t\to t-1}(x_{t-1})$$
$$\geq \exp\left(O\left((1-\alpha_t)^2 \log^2(KT)\right)\right)$$
$$\cdot \int_{\mathcal{E}_t} \left(\frac{1}{4\pi^2\alpha_t^2(1-\alpha_t)}\right)^{d/2} \exp\left(-\frac{\|\sqrt{\alpha_t}x_{t-1}-u_t\|_2^2}{2\alpha_t(1-\alpha_t)}\right) \sum_{k=1}^{K} \pi_k \exp\left(-\frac{\|u_t - \sqrt{\overline{\alpha}_t}\mu_k\|_2^2}{2\alpha_t^2}\right) \mathrm{d}u_t. \tag{76}$$

To further control the right hand side, direct computations give that

$$\int_{\mathbb{R}^d} \left(\frac{1}{4\pi^2\alpha_t^2(1-\alpha_t)}\right)^{d/2} \exp\left(-\frac{\|\sqrt{\alpha_t}x_{t-1}-u_t\|_2^2}{2\alpha_t(1-\alpha_t)}\right) \sum_{k=1}^{K} \pi_k \exp\left(-\frac{\|u_t - \sqrt{\overline{\alpha}_t}\mu_k\|_2^2}{2\alpha_t^2}\right) \mathrm{d}u_t$$
$$= \int_{\mathbb{R}^d} \left(\frac{1}{4\pi^2\alpha_t^2(1-\alpha_t)}\right)^{d/2} \sum_{k=1}^{K} \pi_k \exp\left(-\frac{\|u_t - \sqrt{\overline{\alpha}_t}(1-\alpha_t)\mu_k - \alpha_t^{3/2}x_{t-1}\|_2^2}{2\alpha_t^2(1-\alpha_t)} - \frac{\|x_{t-1} - \sqrt{\overline{\alpha}_t/\alpha_t}\mu_k\|_2^2}{2}\right) \mathrm{d}u_t$$
$$\overset{\text{(i)}}{=} \int_{\mathbb{R}^d} \left(\frac{1}{2\pi}\right)^{d/2} \sum_{k=1}^{K} \pi_k \exp\left(-\frac{1}{2}\|x_{t-1} - \sqrt{\overline{\alpha}_{t-1}}\mu_k\|_2^2\right) \mathrm{d}u_t$$
$$\overset{\text{(ii)}}{=} p_{X_{t-1}^{\mathsf{GMM}}}(x_{t-1}). \tag{77}$$

Here (i) is true as $u_t \mapsto \left(2\pi\alpha_t^2(1-\alpha_t)\right)^{-d/2} \exp\left(-\left(2\alpha_t^2(1-\alpha_t)\right)^{-1}\|u_t - \sqrt{\overline{\alpha}_t}(1-\alpha_t)\mu_k - \alpha_t^{3/2}x_{t-1}\|_2^2\right)$ is a density function and $\overline{\alpha}_t := \prod_{i=1}^{t} \alpha_t$, and (ii) arises from (34). Hence, if we define function $\delta_{t-1}(x) : \mathbb{R}^d \to \mathbb{R}$ to capture the integral on set $\mathcal{E}_t^c$ where

$$\delta_{t-1}(x) := \int_{\mathcal{E}_t^c} \left(\frac{1}{4\pi^2\alpha_t^2(1-\alpha_t)}\right)^{d/2} \exp\left(-\frac{\|\sqrt{\alpha_t}x-u_t\|_2^2}{2\alpha_t(1-\alpha_t)}\right) \sum_{k=1}^{K} \pi_k \exp\left(-\frac{\|u_t - \sqrt{\overline{\alpha}_t}\mu_k\|^2}{2\alpha_t^2}\right) \mathrm{d}u_t,$$

then it obeys

$$\delta_{t-1}(x) = p_{X_{t-1}^{\mathsf{GMM}}}(x) - \int_{\mathcal{E}_t} \left(\frac{1}{4\pi^2\alpha_t^2(1-\alpha_t)}\right)^{d/2} \exp\left(-\frac{\|\sqrt{\alpha_t}x-u_t\|_2^2}{2\alpha_t(1-\alpha_t)}\right) \sum_{k=1}^{K} \pi_k \exp\left(-\frac{\|u_t - \sqrt{\overline{\alpha}_t}\mu_k\|^2}{2\alpha_t^2}\right) \mathrm{d}u_t.$$

Combining this definition with relation (76), we obtain

$$p_{X_{t-1}^{\mathsf{GMM}}}(x_{t-1}) - \Delta_{t-1}(x_{t-1}) + \Delta_{t\to t-1}(x_{t-1})$$

$$\geq \exp\Big(O\big((1-\alpha_t)^2 \log^2(KT)\big)\Big)\big(p_{X_{t-1}^{\mathsf{GMM}}}(x_{t-1}) - \delta_{t-1}(x_{t-1})\big)$$

$$\geq p_{X_{t-1}^{\mathsf{GMM}}}(x_{t-1}) + O\big((1-\alpha_t)^2 \log^2(KT)\big)p_{X_{t-1}^{\mathsf{GMM}}}(x_{t-1}) - O(1)\delta_{t-1}(x_{t-1}),$$

or equivalently,

$$\Delta_{t-1}(x_{t-1}) \leq \Delta_{t \to t-1}(x_{t-1}) + O\big((1-\alpha_t)^2 \log^2(KT)\big)p_{X_{t-1}^{\mathsf{GMM}}}(x_{t-1}) + O(1)\delta_{t-1}(x_{t-1}). \tag{78}$$

Here, we use (33) that $(1-\alpha_t)^2 \log^2(KT) \lesssim \log^2(KT)\log^2 T/T^2 = o(1)$ as long as $T$ is large enough.

We claim that $\int_{\mathbb{R}^d} \delta_{t-1}(x)\,\mathrm{d}x$ satisfies

$$\int_{\mathbb{R}^d} \delta_{t-1}(x)\,\mathrm{d}x \lesssim T^{-3} + (1-\alpha_t)^2 \log^2(KT). \tag{79}$$

Therefore, substituting (79) and (76) into (78) and integrating over $x_{t-1}$ yields

$$\int_{\mathbb{R}^d} \Delta_{t-1}(x_{t-1})\,\mathrm{d}x_{t-1} \leq \int_{\mathbb{R}^d} \Delta_{t \to t-1}(x_{t-1})\,\mathrm{d}x_{t-1} + O\big((1-\alpha_t)^2 \log^2(KT)\big) \int_{\mathbb{R}^d} p_{X_{t-1}^{\mathsf{GMM}}}(x_{t-1})\,\mathrm{d}x_{t-1}$$

$$+ O(1) \int_{\mathbb{R}^d} \delta_{t-1}(x_{t-1})\,\mathrm{d}x_{t-1}$$

$$\leq \int_{\mathbb{R}^d} \Delta_t(x_{t-1})\,\mathrm{d}x_{t-1} + O\big((1-\alpha_t)^2 \log^2(KT)\big) + O\big(T^{-3}\big),$$

where the penultimate line uses (71)

This completes the proof of Lemma 5.

**Proof of Claim** (79). It remains to control $\int_{\mathbb{R}^d} \delta_{t-1}(x)\,\mathrm{d}x$. To this end, the expression above allows us to derive

$$\int_{\mathbb{R}^d} \delta_{t-1}(x_{t-1})\,\mathrm{d}x_{t-1}$$

$$= 1 - \int_{\mathbb{R}^d} \int_{\mathcal{E}_t} \left(\frac{1}{4\pi^2\alpha_t^2(1-\alpha_t)}\right)^{d/2} \sum_{k=1}^{K} \pi_k \exp\left(-\frac{\big\|u_t - \sqrt{\overline{\alpha}_t}\mu_k\big\|_2^2}{2\alpha_t^2}\right) \exp\left(-\frac{\|\sqrt{\alpha_t}x_{t-1} - u_t\|_2^2}{2\alpha_t(1-\alpha_t)}\right) \mathrm{d}u_t\,\mathrm{d}x_{t-1}$$

$$\overset{(i)}{=} 1 - \int_{\mathbb{R}^d} \int_{\mathcal{E}_t} \exp\Big(O\big((1-\alpha_t)^2\log^2(KT)\big)\Big)p_{X_t^{\mathsf{GMM}}}(x_t)\left(\frac{1}{2\pi(1-\alpha_t)}\right)^{d/2} \exp\left(-\frac{\|\sqrt{\alpha_t}x_{t-1} - u_t\|_2^2}{2\alpha_t(1-\alpha_t)}\right) \mathrm{d}x_t\,\mathrm{d}x_{t-1}$$

$$= 1 - \int_{\mathcal{E}_t} \exp\Big(O\big((1-\alpha_t)^2\log^2(KT)\big)\Big)p_{X_t^{\mathsf{GMM}}}(x_t) \int_{\mathbb{R}^d} \left(\frac{1}{2\pi(1-\alpha_t)}\right)^{d/2} \exp\left(-\frac{\|x_{t-1} - u_t/\sqrt{\alpha_t}\|_2^2}{2(1-\alpha_t)}\right) \mathrm{d}x_{t-1}\,\mathrm{d}x_t$$

$$\overset{(ii)}{=} 1 - \int_{\mathcal{E}_t} \exp\Big(O\big((1-\alpha_t)^2\log^2(KT)\big)\Big)p_{X_t^{\mathsf{GMM}}}(x_t)\,\mathrm{d}x_t$$

$$\overset{(iii)}{\leq} 1 - \int_{\mathcal{E}_t} p_{X_t^{\mathsf{GMM}}}(x_t)\,\mathrm{d}x_t + O\big((1-\alpha_t)^2\log^2(KT)\big) \int_{\mathcal{E}_t} p_{X_t^{\mathsf{GMM}}}(x_t)\,\mathrm{d}x_t$$

$$\leq \int_{\mathcal{E}_t^{\mathsf{c}}} p_{X_t^{\mathsf{GMM}}}(x_t)\,\mathrm{d}x_t + O\big((1-\alpha_t)^2\log^2(KT)\big). \tag{80}$$

Here, (i) invokes Lemma 7, (ii) is true as $x_{t-1} \mapsto \big(2\pi(1-\alpha_t)\big)^{-d/2} \exp\big(-\big(2(1-\alpha_t)\big)^{-1}\|x_{t-1} - u_t/\sqrt{\alpha_t}\|_2^2\big)$ is a density function, and (iii) holds as $\exp(x) \geq 1 + x$ for all $x \in \mathbb{R}^d$.

Finally, the right-hand-side of the above bound is controlled by Lemma 8 below.

**Lemma 8.** *Recall the definition of $\mathcal{E}_t$ in (37). For any $t \in [T]$, one has*

$$\int_{\mathcal{E}_t^{\mathsf{c}}} p_{X_t^{\mathsf{GMM}}}(x_t)\,\mathrm{d}x_t \lesssim T^{-3}. \tag{81}$$

*Proof.* See Appendix B.5. □

Putting everything together completes the proof of Claim (79).

### B.3. Proof of Lemma 6

Recall the definition of the score estimation error with respect to the auxiliar GMM sequence $\varepsilon_{\mathsf{score}}^{\mathsf{GMM}}$ and $\varepsilon_{\mathsf{score},t}^{\mathsf{GMM}}$ in (61). For some sufficiently large absolute constants $C_7 > 0$, we define sets

$$\mathcal{A}_t := \left\{ x \in \mathbb{R}^d \colon \log p_{X_t^{\mathsf{GMM}}}(x) \geq -C_7 d \log(dT), \ \|x\|_2 \leq \sqrt{2T^{2c_R} + 16 d \log T} \right\}, \tag{82}$$

$$\mathcal{B}_t := \left\{ x \in \mathbb{R}^d \colon p_{X_t^{\mathsf{GMM}}}(x) \leq 2\, p_{X_t}(x) \right\}, \tag{83}$$

for each $t \in [T]$. We collect several important properties of the set $\mathcal{A}_t$ in the following lemma. Its proof can be found in Appendix B.6.

**Lemma 9.** *It satisfies that*

$$\mathbb{P}\{X_t^{\mathsf{GMM}} \notin \mathcal{A}_t\} \lesssim \exp(-2d \log T), \tag{84}$$

*and on the set $\mathcal{A}_t$, the score function of $X_t^{\mathsf{GMM}}$ obeys*

$$\left\| s_t^{\star\mathsf{GMM}}(x) \right\|_2 \leq C_{\mathsf{clip}} \sqrt{\frac{d \log(dT)}{1 - \overline{\alpha}_t}}, \tag{85}$$

*where $C_{\mathsf{clip}}$ is defined in (13). In addition, one has*

$$\mathbb{E}\left[ \left\| s_t^{\star\mathsf{GMM}}(X_t^{\mathsf{GMM}}) \right\|_2^4 \right] \lesssim \left( \frac{d}{1 - \alpha_t} \right)^2. \tag{86}$$

With this lemma in place, let us proceed to control $\varepsilon_{\mathsf{score}}^{\mathsf{GMM}}$. Recall that $\widehat{s}_t = \mathsf{clip}\{s_t\}$ defined in (13). Fixing an arbitrary $t \in [T]$, we first decompose

$$\left( \varepsilon_{\mathsf{score},t}^{\mathsf{GMM}} \right)^2 = \int_{\mathbb{R}^d} \left\| \mathsf{clip}\{s_t(x_t)\} - s_t^{\star\mathsf{GMM}}(x_t) \right\|_2^2 p_{X_t^{\mathsf{GMM}}}(x_t)\, \mathrm{d}x_t$$

$$= \underbrace{\int_{\mathcal{A}_t \cap \mathcal{B}_t} \left\| \mathsf{clip}\{s_t(x_t)\} - s_t^{\star\mathsf{GMM}}(x_t) \right\|_2^2 p_{X_t^{\mathsf{GMM}}}(x_t)\, \mathrm{d}x_t}_{=:(\mathrm{I})} + \underbrace{\int_{\mathcal{A}_t \cap \mathcal{B}_t^c} \left\| \mathsf{clip}\{s_t(x_t)\} - s_t^{\star\mathsf{GMM}}(x_t) \right\|_2^2 p_{X_t^{\mathsf{GMM}}}(x_t)\, \mathrm{d}x_t}_{=:(\mathrm{II})}$$

$$+ \underbrace{\int_{\mathcal{A}_t^c} \left\| \mathsf{clip}\{s_t(x_t)\} - s_t^{\star\mathsf{GMM}}(x_t) \right\|_2^2 p_{X_t^{\mathsf{GMM}}}(x_t)\, \mathrm{d}x_t}_{=:(\mathrm{III})}.$$

In what follows, we will bound these three quantities separately.

**Controlling (I).** Notice that on the set $\mathcal{A}_t$, the score function of $X_t^{\mathsf{GMM}}$ satisfies $\mathsf{clip}\{s_t^{\star\mathsf{GMM}}(x)\} = s_t^{\star\mathsf{GMM}}(x)$. Thus, we can use the Cauchy-Schwartz inequality to derive

$$(\mathrm{I}) \leq \int_{\mathcal{A}_t \cap \mathcal{B}_t} \left\| s_t(x_t) - s_t^{\star\mathsf{GMM}}(x_t) \right\|_2^2 p_{X_t^{\mathsf{GMM}}}(x_t)\, \mathrm{d}x_t$$

$$\leq 2 \int_{\mathcal{A}_t \cap \mathcal{B}_t} \left\| s_t(x_t) - s_t^\star(x_t) \right\|_2^2 p_{X_t^{\mathsf{GMM}}}(x_t)\, \mathrm{d}x_t + 2 \int_{\mathcal{A}_t \cap \mathcal{B}_t} \left\| s_t^\star(x_t) - s_t^{\star\mathsf{GMM}}(x_t) \right\|_2^2 p_{X_t^{\mathsf{GMM}}}(x_t)\, \mathrm{d}x_t$$

$$\leq 4 \int_{\mathbb{R}^d} \left\| s_t(x_t) - s_t^\star(x_t) \right\|_2^2 p_{X_t}(x_t)\, \mathrm{d}x_t + 4 \int_{\mathbb{R}^d} \left\| s_t^\star(x_t) - s_t^{\star\mathsf{GMM}}(x_t) \right\|_2^2 \left\{ p_{X_t}(x_t) \wedge p_{X_t^{\mathsf{GMM}}}(x_t) \right\}\, \mathrm{d}x_t$$

$$\asymp \varepsilon_{\mathsf{score},t}^2 + \int_{\mathbb{R}^d} \left\| s_t^\star(x_t) - s_t^{\star\mathsf{GMM}}(x_t) \right\|_2^2 \left\{ p_{X_t}(x_t) \wedge p_{X_t^{\mathsf{GMM}}}(x_t) \right\}\, \mathrm{d}x_t,$$

where the last step uses Assumption 2. Next, we claim that

$$\int_{\mathbb{R}^d} \left\| s_t^\star(x_t) - s_t^{\star\mathsf{GMM}}(x_t) \right\|_2^2 \left\{ p_{X_t}(x_t) \wedge p_{X_t^{\mathsf{GMM}}}(x_t) \right\}\, \mathrm{d}x_t \lesssim \frac{\varepsilon_{\mathsf{apprx}} d \log T}{1 - \overline{\alpha}_t} + \frac{1}{(1 - \overline{\alpha}_t) T^d}, \tag{87}$$

with the proof deferred to the end of this section. As a result, one finds

$$(\mathrm{I}) \lesssim \varepsilon_{\mathsf{score},t}^2 + \frac{\varepsilon_{\mathsf{apprx}} d \log T}{1 - \overline{\alpha}_t} + \frac{1}{(1 - \overline{\alpha}_t) T^d}. \tag{88}$$

**Controlling (II).** By our construction of the truncation function $\mathrm{clip}\{\cdot\}$ in (13) and the set $\mathcal{A}_t$, one has

$$\big\|\mathrm{clip}\{s_t(x_t)\} - s_t^{\star\mathsf{GMM}}(x_t)\big\|_2^2 \lesssim \big\|\mathrm{clip}\{s_t(x_t)\}\big\|_2^2 + \big\|s_t^{\star\mathsf{GMM}}(x_t)\big\|_2^2 \lesssim \frac{d\log(dT)}{1-\overline{\alpha}_t}, \tag{89}$$

where the last step follows from (85) in Lemma 9. In addition, on the set $\mathcal{B}_t^c = \{2\,p_{X_t}(x) < p_{X_t^{\mathsf{GMM}}}(x)\}$, one has

$$p_{X_t^{\mathsf{GMM}}}(x_t) \le 2\left|p_{X_t^{\mathsf{GMM}}}(x_t) - p_{X_t}(x_t)\right|.$$

It follows that

$$\mathbb{P}\{X_t^{\mathsf{GMM}} \notin \mathcal{B}_t\} = \int_{\mathbb{R}^d} p_{X_t^{\mathsf{GMM}}}(x_t)\mathbb{1}\big\{2\,p_{X_t}(x_t) < p_{X_t^{\mathsf{GMM}}}(x_t)\big\}\,\mathrm{d}x_t$$

$$\le 2\int_{\mathbb{R}^d}\left|p_{X_t^{\mathsf{GMM}}}(x_t) - p_{X_t}(x_t)\right|\,\mathrm{d}x_t$$

$$= 4\,\mathsf{TV}(p_{X_t^{\mathsf{GMM}}}, p_{X_t}) \overset{(i)}{\lesssim} \mathsf{TV}(p_{X_0^{\mathsf{GMM}}}, p_{X_0}) \overset{(ii)}{=} \varepsilon_{\mathsf{apprx}}, \tag{90}$$

where (i) holds due to the data processing inequality, and (ii) arises from Assumption 1.

Collecting (89) and (90) together, we arrive at

$$(\mathrm{II}) \lesssim \frac{\varepsilon_{\mathsf{apprx}}d\log(dT)}{1-\overline{\alpha}_t}. \tag{91}$$

**Controlling (III).** Using the definition of $\mathrm{clip}\{\cdot\}$ again, we apply the Cauchy-Schwartz inequality to obtain

$$(\mathrm{III}) \lesssim \frac{d\log(dT)}{1-\overline{\alpha}_t}\mathbb{P}\{X_t^{\mathsf{GMM}} \notin \mathcal{A}_t\} + \sqrt{\mathbb{E}\Big[\big\|s_t^{\star\mathsf{GMM}}(X_t^{\mathsf{GMM}})\big\|_2^4\Big]\mathbb{P}\{X_t^{\mathsf{GMM}} \notin \mathcal{A}_t\}}$$

$$\overset{(i)}{\lesssim} \frac{d\log(dT)}{1-\overline{\alpha}_t}\mathbb{P}\{X_t^{\mathsf{GMM}} \notin \mathcal{A}_t\} + \frac{d}{1-\overline{\alpha}_t}\sqrt{\mathbb{P}\{X_t^{\mathsf{GMM}} \notin \mathcal{A}_t\}}$$

$$\overset{(ii)}{\lesssim} \frac{d\log(dT)}{(1-\overline{\alpha}_t)}\exp(-2d\log T) + \frac{d}{(1-\overline{\alpha}_t)}\exp(-d\log T)$$

$$\lesssim \frac{1}{(1-\overline{\alpha}_t)T}. \tag{92}$$

where (i) invokes (86) from Lemma 9 and (ii) arises from (84) in Lemma 9.

**Putting (I)–(III) together.** Putting (88), (91), and (92) together, we obtain

$$(1-\overline{\alpha}_t)\big(\varepsilon_{\mathsf{score},t}^{\mathsf{GMM}}\big)^2 \lesssim (1-\overline{\alpha}_t)\varepsilon_{\mathsf{score},t}^2 + \varepsilon_{\mathsf{apprx}}d\log(dT) + T^{-1}.$$

Then the claim (65) in Lemma 6 immediately follows from the definition of $\varepsilon_{\mathsf{score}}^{\mathsf{GMM}}$ in (61).

**Proof of Claim** (87). Define the set

$$\mathcal{H}_x := \Big\{x_0 \in \mathbb{R}^d\colon \big\|x - \sqrt{\overline{\alpha}_t}x_0\big\|_2 \le 3\sqrt{d(1-\overline{\alpha}_t)\log T}\Big\}.$$

Using Tweedie's formula, we can bound the discrepancy between the score functions of the target distribution and its GMM approximation as follows:

$$\big\|s_t^\star(x_t) - s_t^{\star\mathsf{GMM}}(x_t)\big\|_2 = \frac{1}{1-\overline{\alpha}_t}\Big\|\mathbb{E}\big[\sqrt{\overline{\alpha}_t}X_0 - X_t \mid X_t = x_t\big] - \mathbb{E}\big[\sqrt{\overline{\alpha}_t}X_0^{\mathsf{GMM}} - X_t^{\mathsf{GMM}} \mid X_t^{\mathsf{GMM}} = x_t\big]\Big\|_2$$

$$\le \frac{1}{1-\overline{\alpha}_t}\int_{\mathbb{R}^d}\big|p_{X_0^{\mathsf{GMM}}|X_t^{\mathsf{GMM}}}(x_0 \mid x_t) - p_{X_0|X_t}(x_0 \mid x_t)\big|\,\big\|x_t - \sqrt{\overline{\alpha}_t}x_0\big\|_2\,\mathrm{d}x_0$$

$$\leq 2\,\mathsf{TV}(p_{X_0^{\mathsf{GMM}}|X_t^{\mathsf{GMM}}=x_t}, p_{X_0|X_t=x_t})\sqrt{\frac{d\log T}{1-\overline{\alpha}_t}}$$

$$+ \frac{1}{1-\overline{\alpha}_t}\int_{x_0\in\mathcal{H}_{x_t}^{\mathsf{c}}}\left(p_{X_0^{\mathsf{GMM}}|X_t^{\mathsf{GMM}}}(x_0\mid x_t) + p_{X_0|X_t}(x_0\mid x_t)\right)\left\|x_t - \sqrt{\overline{\alpha}_t}x_0\right\|_2 \mathrm{d}x_0,$$

where the last uses the definition of $\mathcal{H}_x$ and the TV distance. By the Cauchy-Schwartz inequality, we can bound

$$\left(\int_{\mathcal{H}_{x_t}^{\mathsf{c}}} p_{X_0|X_t}(x_0\mid x_t)\left\|x_t - \sqrt{\overline{\alpha}_t}x_0\right\|_2 \mathrm{d}x_0\right)^2 \leq \mathbb{P}\{X_0\in\mathcal{H}_x^{\mathsf{c}}\mid X_t=x_t\}\int_{\mathcal{H}_{x_t}^{\mathsf{c}}} p_{X_0|X_t}(x_0\mid x_t)\left\|x_t - \sqrt{\overline{\alpha}_t}x_0\right\|_2^2 \mathrm{d}x_0$$

$$\leq \int_{\mathcal{H}_{x_t}^{\mathsf{c}}} p_{X_0|X_t}(x_0\mid x_t)\left\|x_t - \sqrt{\overline{\alpha}_t}x_0\right\|_2^2 \mathrm{d}x_0.$$

Similarly, one also has

$$\left(\int_{\mathcal{H}_{x_t}^{\mathsf{c}}} p_{X_0|X_t}(x_0\mid x_t)\left\|x_t - \sqrt{\overline{\alpha}_t}x_0\right\|_2 \mathrm{d}x_0\right)^2 \leq \int_{\mathcal{H}_{x_t}^{\mathsf{c}}} p_{X_0^{\mathsf{GMM}}|X_t^{\mathsf{GMM}}}(x_0\mid x_t)\left\|x_t - \sqrt{\overline{\alpha}_t}x_0\right\|_2^2 \mathrm{d}x_0.$$

Taking the above collectivly, the quantity of interest satisfies

$$\int_{\mathbb{R}^d}\|s_t^{\star}(x_t) - s_t^{\star\mathsf{GMM}}(x_t)\|_2^2\left\{p_{X_t}(x_t)\wedge p_{X_t^{\mathsf{GMM}}}(x_t)\right\}\mathrm{d}x_t$$

$$\lesssim \underbrace{\frac{d\log T}{1-\overline{\alpha}_t}\int_{\mathbb{R}^d}\mathsf{TV}(p_{X_0^{\mathsf{GMM}}|X_t^{\mathsf{GMM}}=x_t}, p_{X_0|X_t=x_t})\left\{p_{X_t}(x_t)\wedge p_{X_t^{\mathsf{GMM}}}(x_t)\right\}\mathrm{d}x_t}_{=:(\mathrm{IV})}$$

$$+ \underbrace{\frac{1}{(1-\overline{\alpha}_t)^2}\int_{x_t\in\mathbb{R}^d}\int_{x_0\in\mathcal{H}_{x_t}^{\mathsf{c}}}\left(p_{X_0^{\mathsf{GMM}},X_t^{\mathsf{GMM}}}(x_0,x_t) + p_{X_0,X_t}(x_0,x_t)\right)\left\|x_t - \sqrt{\overline{\alpha}_t}x_0\right\|_2^2 \mathrm{d}x_0\,\mathrm{d}x_t}_{=:(\mathrm{V})}. \quad (93)$$

where we use the fact that $\mathsf{TV}(p_{X_0^{\mathsf{GMM}}|X_t^{\mathsf{GMM}}=x_t}, p_{X_0|X_t=x_t})\in[0,1]$. It then boils down to control quantities (IV) and (V) respectively, which shall we done as follows.

- To bound (IV) in (93), first notice that

$$\left|p_{X_0|X_t}(x_0\mid x_t) - p_{X_0^{\mathsf{GMM}}|X_t^{\mathsf{GMM}}}(x_0\mid x_t)\right|\left\{p_{X_t}(x_t)\wedge p_{X_t^{\mathsf{GMM}}}(x_t)\right\}$$

$$\leq p_{X_0|X_t}(x_0\mid x_t)\cdot\left|\left\{p_{X_t}(x_t)\wedge p_{X_t^{\mathsf{GMM}}}(x_t)\right\} - p_{X_t}(x_t)\right|$$

$$+ \left|p_{X_0|X_t}(x_0\mid x_t)p_{X_t}(x_t) - p_{X_0^{\mathsf{GMM}}|X_t^{\mathsf{GMM}}}(x_0\mid x_t)p_{X_t^{\mathsf{GMM}}}(x_t)\right|$$

$$+ p_{X_0^{\mathsf{GMM}}|X_t^{\mathsf{GMM}}}(x_0\mid x_t)\cdot\left|p_{X_t^{\mathsf{GMM}}}(x_t) - \left\{p_{X_t}(x_t)\wedge p_{X_t^{\mathsf{GMM}}}(x_t)\right\}\right|$$

$$\leq \left(p_{X_0|X_t}(x_0\mid x_t) + p_{X_0^{\mathsf{GMM}}|X_t^{\mathsf{GMM}}}(x_0\mid x_t)\right)\left|p_{X_t}(x_t) - p_{X_t^{\mathsf{GMM}}}(x_t)\right| + \left|p_{X_0,X_t}(x_0,x_t) - p_{X_0^{\mathsf{GMM}},X_t^{\mathsf{GMM}}}(x_0,x_t)\right|.$$

Consequently, we can bound

$$\int_{\mathbb{R}^d}\mathsf{TV}(p_{X_0^{\mathsf{GMM}}|X_t^{\mathsf{GMM}}=x_t}, p_{X_0|X_t=x_t})\left\{p_{X_t}(x_t)\wedge p_{X_t^{\mathsf{GMM}}}(x_t)\right\}\mathrm{d}x_t$$

$$= \frac{1}{2}\int_{\mathbb{R}^d}\int_{\mathbb{R}^d}\left|p_{X_0|X_t}(x_0\mid x_t) - p_{X_0^{\mathsf{GMM}}|X_t^{\mathsf{GMM}}}(x_0\mid x_t)\right|\left\{p_{X_t}(x_t)\wedge p_{X_t^{\mathsf{GMM}}}(x_t)\right\}\mathrm{d}x_0\,\mathrm{d}x_t$$

$$\leq \int_{\mathbb{R}^d}\left|p_{X_t}(x_t) - p_{X_t^{\mathsf{GMM}}}(x_t)\right|\mathrm{d}x_t + \frac{1}{2}\int_{\mathbb{R}^d}\int_{\mathbb{R}^d}\left|p_{X_0,X_t}(x_0,x_t) - p_{X_0^{\mathsf{GMM}},X_t^{\mathsf{GMM}}}(x_0,x_t)\right|\mathrm{d}x_0\,\mathrm{d}x_t$$

$$\leq 2\mathsf{TV}(p_{X_t^{\mathsf{GMM}}}, p_{X_t}) + \mathsf{TV}(p_{X_0^{\mathsf{GMM}},X_t^{\mathsf{GMM}}}, p_{X_0,X_t}) \leq 3\mathsf{TV}(p_{X_0^{\mathsf{GMM}}}, p_{X_0}) = 3\varepsilon_{\mathsf{apprx}},$$

where the last line follows from the data processing inequality and Assumption 1. Thus, we arrive

$$(\mathrm{IV}) \lesssim \frac{\varepsilon_{\mathsf{apprx}}d\log T}{1-\overline{\alpha}_t}. \quad (94)$$

- Regarding (V) in (93), since $X_t \mid X_0 = x_0 \sim \mathcal{N}(\sqrt{\overline{\alpha}_t}x_0, (1 - \overline{\alpha}_t)I_d)$, we can derive

$$\int_{x_t \in \mathbb{R}^d} \int_{x_0 \in \mathcal{H}_{x_t}^c} p_{X_0, X_t}(x_0, x_t) \big\| x_t - \sqrt{\overline{\alpha}_t}x_0 \big\|_2^2 \, \mathrm{d}x_0 \, \mathrm{d}x_t$$

$$= \int_{x_0 \in \mathbb{R}^d} p_{X_0}(x_0) \int_{x_t \in \mathbb{R}^d} p_{X_t \mid X_0}(x_t \mid x_0) \big\| x_t - \sqrt{\overline{\alpha}_t}x_0 \big\|_2^2 \mathbb{1}\big\{ \big\| x_t - \sqrt{\overline{\alpha}_t}x_0 \big\|_2 > 3\sqrt{d(1-\overline{\alpha}_t)\log T} \big\} \, \mathrm{d}x_t \, \mathrm{d}x_0$$

$$\leq (1 - \overline{\alpha}_t)\mathbb{E}\big[ \|Z\|_2^2 \mathbb{1}\big\{ \|Z\|_2 > 3\sqrt{d\log T} \big\} \big]$$

where $Z \sim \mathcal{N}(0, I_d)$ is standard Gaussian random vector in $\mathbb{R}^d$. It follows that

$$\mathbb{E}\big[ \|Z\|_2^2 \mathbb{1}\big\{ \|Z\|_2 > 3\sqrt{d\log T} \big\} \big] = \int_0^\infty \mathbb{P}\big\{ \|Z\|_2^2 \mathbb{1}\big\{ \|Z\|_2^2 > 9d\log T \big\} > x \big\} \, \mathrm{d}x$$

$$= \int_{9d\log T}^\infty \mathbb{P}\big\{ \|Z\|_2^2 > x \big\} \, \mathrm{d}x$$

$$\overset{(i)}{\leq} \int_{9d\log T}^\infty \mathbb{P}\big\{ \|Z\|_2^2 > 2d + x/2 \big\} \, \mathrm{d}x$$

$$\overset{(ii)}{\leq} \int_{9d\log T}^\infty \exp\big(-x/6\big) \, \mathrm{d}x$$

$$= 6\exp\big(-(3/2)d\log T\big) \lesssim T^{-d},$$

where (i) holds as long as $9\log T > 4$; (ii) applies the Gaussian concentration inequality Laurent & Massart (2000, Lemma 1):

$$\mathbb{P}\big\{ \|Z\|_2^2 > 2d + 3x \big\} \leq \mathbb{P}\big\{ \|Z\|_2^2 - d > 2\sqrt{dx} + 2x \big\} \leq \exp(-x), \quad \forall x > 0. \tag{95}$$

Clearly, the above bound is also valid for $\int_{x_t \in \mathbb{R}^d} \int_{x_0 \in \mathcal{H}_{x_t}^c} p_{X_0^{\mathsf{GMM}}, X_t^{\mathsf{GMM}}}(x_0, x_t)\big\| x_t - \sqrt{\overline{\alpha}_t}x_0 \big\|_2^2 \, \mathrm{d}x_0 \, \mathrm{d}x_t$. Hence, we conclude

$$(\mathrm{V}) \lesssim \frac{1}{(1 - \overline{\alpha}_t)T^d}. \tag{96}$$

Plugging (94) and (96) into (93) completes the proof of Claim (87).

### B.4. Proof of Lemma 7

Let us first derive two relations that are key for this proof. To start with, fix an arbitrary $x_t \in \mathcal{E}_t$. Recalling the definition that $u_t := x_t + (1 - \alpha_t)s_t^\star(x_t)$, direct calculations yield

$$\frac{1}{2\alpha_t^2}\big\| u_t - \sqrt{\overline{\alpha}_t}\mu_k \big\|_2^2$$

$$= \frac{1}{2}\big\| x_t - \sqrt{\overline{\alpha}_t}\mu_k \big\|_2^2 + \frac{1 - \alpha_t^2}{2\alpha_t^2}\big\| x_t - \sqrt{\overline{\alpha}_t}\mu_k \big\|_2^2 + \frac{(1 - \alpha_t)}{\alpha_t^2}s_t^\star(x_t)^\top(x_t - \sqrt{\overline{\alpha}_t}\mu_k) + \frac{(1 - \alpha_t)^2}{2\alpha_t^2}\big\| s_t^\star(x_t) \big\|_2^2$$

$$= \frac{1}{2}\big\| x_t - \sqrt{\overline{\alpha}_t}\mu_k \big\|_2^2 + \frac{1 - \alpha_t^2}{2\alpha_t^2}\sum_{i=1}^K \pi_i^{(t)}\big\| x_t - \sqrt{\overline{\alpha}_t}\mu_i \big\|_2^2 + \left( \frac{(1 - \alpha_t)^2}{2\alpha_t^2} - \frac{1 - \alpha_t}{\alpha_t^2} \right)\big\| s_t^\star(x_t) \big\|_2^2$$

$$+ \frac{1 - \alpha_t^2}{2\alpha_t^2}\left( \big\| x_t - \sqrt{\overline{\alpha}_t}\mu_k \big\|_2^2 - \sum_{i=1}^K \pi_i^{(t)}\big\| x_t - \sqrt{\overline{\alpha}_t}\mu_i \big\|_2^2 \right) + \frac{1 - \alpha_t}{\alpha_t^2}s_t^\star(x_t)^\top\big( s_t^\star(x_t) + x_t - \sqrt{\overline{\alpha}_t}\mu_k \big)$$

$$\overset{(i)}{=} \frac{1}{2}\big\| x_t - \sqrt{\overline{\alpha}_t}\mu_k \big\|_2^2 + \frac{1 - \alpha_t^2}{2\alpha_t^2}\sum_{i=1}^K \pi_i^{(t)}\big\| x_t - \sqrt{\overline{\alpha}_t}\mu_i \big\|_2^2 - \frac{1 - \alpha_t^2}{2\alpha_t^2}\big\| s_t^\star(x_t) \big\|_2^2 + \zeta_k^{(t)}(x_t)$$

$$\overset{(ii)}{=} \frac{1}{2}\big\| x_t - \sqrt{\overline{\alpha}_t}\mu_k \big\|_2^2 + \frac{1 - \alpha_t^2}{2\alpha_t^2}\left( \sum_{i=1}^K \pi_i^{(t)}\big\| x_t - \sqrt{\overline{\alpha}_t}\mu_i \big\|_2^2 - \left\| \sum_{i=1}^K \pi_i^{(t)}\big( x_t - \sqrt{\overline{\alpha}_t}\mu_i \big) \right\|_2^2 \right) + \zeta_k^{(t)}(x_t)$$

$$\overset{\text{(iii)}}{=} \frac{1}{2}\left\|x_t - \sqrt{\overline{\alpha}_t}\mu_k\right\|_2^2 + \frac{(1-\alpha_t^2)}{2\alpha_t^2}\mathsf{tr}\big(I_d + J_t(x_t)\big) + \zeta_k^{(t)}(x_t). \tag{97}$$

Here, (i) uses the definition of $\zeta_k^{(t)}(x)$ in (38); (ii) arises from the expression of $s_t^\star(x) = -\sum_{k=1}^K \pi_k^{(t)}\big(x - \sqrt{\overline{\alpha}_t}\mu_k\big)$ in (35); (iii) uses the expression of $J_t(x)$ in (19) that

$$
\begin{aligned}
I_d + J_t(x) &= \sum_{k=1}^K \pi_k^{(t)}\bigg(\sqrt{\overline{\alpha}_t}\mu_k - \sum_{i=1}^K \pi_i^{(t)}\sqrt{\overline{\alpha}_t}\mu_i\bigg)\bigg(\sqrt{\overline{\alpha}_t}\mu_k - \sum_{i=1}^K \pi_i^{(t)}\sqrt{\overline{\alpha}_t}\mu_i\bigg)^\top \\
&= \sum_{k=1}^K \pi_k^{(t)}\bigg(x - \sqrt{\overline{\alpha}_t}\mu_k - \sum_{i=1}^K \pi_i^{(t)}\big(x - \sqrt{\overline{\alpha}_t}\mu_i\big)\bigg)\bigg(x - \sqrt{\overline{\alpha}_t}\mu_k - \sum_{i=1}^K \pi_i^{(t)}\big(x - \sqrt{\overline{\alpha}_t}\mu_i\big)\bigg)^\top \\
&= \sum_{k=1}^K \pi_k^{(t)}\big(x - \sqrt{\overline{\alpha}_t}\mu_k\big)\big(x - \sqrt{\overline{\alpha}_t}\mu_k\big)^\top - \bigg(\sum_{k=1}^K \pi_k^{(t)}\big(x - \sqrt{\overline{\alpha}_t}\mu_k\big)\bigg)\bigg(\sum_{k=1}^K \pi_k^{(t)}\big(x - \sqrt{\overline{\alpha}_t}\mu_k\big)\bigg)^\top.
\end{aligned}
$$

In addition, recall the definition of $\mathcal{E}_t$ (cf. (37)). For any $x_t \in \mathcal{E}_t$, using (33) that $1 - \alpha_t \lesssim \log T/T$, we know that

$$
\begin{aligned}
\frac{1-\alpha_t}{\alpha_t}\mathsf{tr}\big(I_d + J_t(x_t)\big) &\lesssim (1-\alpha_t)\mathsf{tr}\big(I_d + J_t(x_t)\big) \lesssim \frac{\log(KT)}{T} = o(1), \\
\frac{1-\alpha_t^2}{2\alpha_t^2}\mathsf{tr}\big(I_d + J_t(x_t)\big) &\lesssim (1-\alpha_t)\mathsf{tr}\big(I_d + J_t(x_t)\big) \lesssim \frac{\log(KT)}{T} = o(1),
\end{aligned}
$$

for large enough $T$. It follows that

$$
\begin{aligned}
\det\Big(&I_d + \big(\alpha_t^{-1} - 1\big)\big(I_d + J_t(x_t)\big)\Big) \\
&\overset{\text{(i)}}{=} 1 + \frac{1-\alpha_t}{\alpha_t}\mathsf{tr}\big(I_d + J_t(x_t)\big) + O\bigg(\frac{(1-\alpha_t)^2}{\alpha_t^2}\mathsf{tr}^2\big(I_d + J_t(x_t)\big)\bigg) \\
&= 1 + \frac{1-\alpha_t^2}{2\alpha_t^2}\mathsf{tr}\big(I_d + J_t(x_t)\big) - \frac{(1-\alpha_t)^2}{2\alpha_t^2}\mathsf{tr}\big(I_d + J_t(x_t)\big) + O\bigg(\frac{(1-\alpha_t)^2}{\alpha_t^2}\mathsf{tr}^2\big(I_d + J_t(x_t)\big)\bigg) \\
&\overset{\text{(ii)}}{=} 1 + \frac{1-\alpha_t^2}{2\alpha_t^2}\mathsf{tr}\big(I_d + J_t(x_t)\big) + O\big((1-\alpha_t)^2 \log^2(KT)\big) \\
&= \exp\bigg(\frac{1-\alpha_t^2}{2\alpha_t^2}\mathsf{tr}\big(I_d + J_t(x_t)\big) + O\big((1-\alpha_t)^2 \log^2(KT)\big)\bigg).
\end{aligned}
$$

where (i) holds as $I_d + J_t(x_t) \succeq 0$ and $\det(I + \varepsilon A) = 1 + \mathsf{tr}(A)\varepsilon + O\big(\varepsilon^2(\mathsf{tr}^2(A) - \mathsf{tr}(A^2))\big)$ for any matrix $A$ and $\varepsilon > 0$; (ii) is true since $\alpha_t \gtrsim 1$ by (33) and $\mathsf{tr}\big(I_d + J_t(x_t)\big) \lesssim \log(KT)$ by the choice of $\mathcal{E}_t$ in (37). Consequently, we can derive

$$
\begin{aligned}
\det\big(I_d + (1-\alpha_t)J_t(x_t)\big) &= \det\Big(\alpha_t I_d + (1-\alpha_t)\big(I_d + J_t(x_t)\big)\Big) \\
&= \alpha_t^d \det\Big(I_d + \big(\alpha_t^{-1} - 1\big)\big(I_d + J_t(x_t)\big)\Big) \\
&= \alpha_t^d \exp\bigg(\frac{1-\alpha_t^2}{2\alpha_t^2}\mathsf{tr}\big(I_d + J_t(x_t)\big) + O\big((1-\alpha_t)^2 \log^2(KT)\big)\bigg), \tag{98}
\end{aligned}
$$

As a consequence of the above two relations, we move on to prove Lemma 7. In view of relation (97), we arrive at

$$
\begin{aligned}
\bigg(\frac{1}{2\pi\alpha_t^2}\bigg)^{d/2}&\sum_{k=1}^K \pi_k \exp\bigg(-\frac{1}{2\alpha_t^2}\big\|u_t - \sqrt{\overline{\alpha}_t}\mu_k\big\|_2^2\bigg) \\
&= \bigg(\frac{1}{2\pi\alpha_t^2}\bigg)^{d/2}\exp\bigg(-\frac{(1-\alpha_t^2)}{2\alpha_t^2}\mathsf{tr}\big(I + J_t(x_t)\big)\bigg)\sum_{k=1}^K \pi_k \exp\bigg(-\frac{1}{2}\big\|x_t - \sqrt{\overline{\alpha}_t}\mu_k\big\|_2^2\bigg)\exp\big(-\zeta_k^{(t)}(x_t)\big) \\
&= \det\big(I + (1-\alpha_t)J_t(x_t)\big)^{-1}\exp\Big(O\big((1-\alpha_t)^2 \log^2(KT)\big)\Big)p_{X_t^{\mathsf{GMM}}}(x_t)\sum_{k=1}^K \pi_k^t \exp\big(-\zeta_k^{(t)}(x_t)\big)
\end{aligned}
$$

where the last equality uses (98) and $\pi_k \exp\big(-\|x - \sqrt{\overline{\alpha}_t}\mu_k\|_2^2/2\big) = \pi_k^{(t)}(2\pi)^{d/2}p_{X_t^{\mathsf{GMM}}}(x)$ due to (34) and (20). To further control the right hand side, by the definition of $\mathcal{E}_t$ in (37), it satisfies that

$$1 \leq \sum_{k=1}^{K} \pi_k^t \exp\big(-\zeta_k^{(t)}(x_t)\big) \leq \exp\big(C_2(1 - \alpha_t)^2 \log^2(KT)\big).$$

Therefore, we can conclude that

$$\left(\frac{1}{2\pi\alpha_t^2}\right)^{d/2} \sum_{k=1}^{K} \pi_k \exp\Big(-\frac{1}{2\alpha_t^2}\|u_t - \sqrt{\overline{\alpha}_t}\mu_k\|_2^2\Big)$$
$$= \det\big(I + (1 - \alpha_t)J_t(x_t)\big)^{-1} \exp\Big(O\big((1 - \alpha_t)^2 \log^2(KT)\big)\Big)p_{X_t^{\mathsf{GMM}}}(x_t),$$

which completes the proof of Lemma 7.

## B.5. Proof of Lemma 8

Recalling the definition of $\mathcal{E}_t$ in expression (37), we have

$$\mathbb{P}\{X_t^{\mathsf{GMM}} \in \mathcal{E}_t^c\} \leq \mathbb{P}\Big\{\mathsf{tr}\big(I_d + J_t(X_t^{\mathsf{GMM}})\big) \geq C_1 \log(KT)\Big\} + \mathbb{P}\Big\{\sum_{k=1}^{K} \pi_k^{(t)} \exp\big(-\zeta_k^{(t)}(X_t^{\mathsf{GMM}})\big) < 1\Big\}$$
$$+ \mathbb{P}\Big\{\sum_{k=1}^{K} \pi_k^{(t)} \exp\big(-\zeta_k^{(t)}(X_t^{\mathsf{GMM}})\big) > \exp\big(C_2(1 - \alpha_t)^2 \log^2(KT)\big)\Big\}. \tag{99}$$

In the following, we bound the three terms on the right respectively.

Before proceeding, we make the following observation. Fix an arbitrary $t \geq 1$. For each $k \in [K]$, we define the event

$$\mathcal{T}_k := \Big\{x \in \mathbb{R}^d : \big|(x - \sqrt{\overline{\alpha}_t}\mu_k)^\top \sqrt{\overline{\alpha}_t}(\mu_i - \mu_k)\big| \leq C_5 \sqrt{\overline{\alpha}_t \log(KT)}\,\|\mu_i - \mu_k\|_2 \text{ for all } i \in [K]\Big\} \tag{100}$$

for some absolute constant $C_5 > 0$. Note that if we let $Z_k \sim \mathcal{N}(\sqrt{\overline{\alpha}_t}\mu_k, I_d)$ be a Gaussian random vector in $\mathbb{R}^d$, which implies that $(Z_k - \sqrt{\overline{\alpha}_t}\mu_k)^\top \sqrt{\overline{\alpha}_t}(\mu_i - \mu_k) \sim \mathcal{N}(0, \overline{\alpha}_t\|\mu_i - \mu_k\|_2^2)$, the standard Gaussian concentration inequality guarantees that

$$\mathbb{P}\{Z_k \notin \mathcal{T}_k\} \lesssim T^{-3}, \tag{101}$$

provided $C_5$ is large enough.

BOUNDING THE FIRST TERM IN EQ. (99)

Let us begin with the first event $\{\mathsf{tr}\big(I_d + J_t(X_t^{\mathsf{GMM}})\big) \leq C_1 \log(KT)\}$. As $X_t^{\mathsf{GMM}} \sim \sum_{k=1}^{K} \pi_k \mathcal{N}(\sqrt{\overline{\alpha}_t}\mu_k, I_d)$, it is easily seen that

$$\mathbb{P}\Big\{\mathsf{tr}\big(I_d + J_t(X_t^{\mathsf{GMM}})\big) > C_1 \log(KT)\Big\}$$
$$= \sum_{k=1}^{K} \pi_k \mathbb{P}\Big\{\mathsf{tr}\big(I_d + J_t(Z_k)\big) > C_1 \log(KT)\Big\}$$
$$\leq \sum_{k=1}^{k} \pi_k \mathbb{P}\Big\{\mathsf{tr}\big(I_d + J_t(Z_k)\big) > C_1 \log(KT)\Big\}\mathbb{1}\big\{\pi_k \geq 1/(KT^3)\big\} + \sum_{k=1}^{K} \pi_k \mathbb{1}\big\{\pi_k < 1/(KT^3)\big\}$$
$$\leq \sum_{k=1}^{k} \pi_k \mathbb{P}\Big\{\mathsf{tr}\big(I_d + J_t(Z_k)\big) > C_1 \log(KT)\Big\}\mathbb{1}\big\{\pi_k \geq 1/(KT^3)\big\} + T^{-3}.$$

We claim that for any $k \in [K]$ such that $\pi_k \geq 1/(KT^3)$, one has

$$\mathcal{T}_k \subset \Big\{x \in \mathbb{R}^d : \mathsf{tr}\big(I_d + J_t(x)\big) \leq C_1 \log(KT)\Big\}. \tag{102}$$

It then immediately follows from (101) that

$$\mathbb{P}\Big\{\mathsf{tr}\big(I_d + J_t\big(X_t^{\mathsf{GMM}}\big)\big) > C_1 \log(KT)\Big\} \leq \sum_{k=1}^{K} \pi_k \mathbb{P}\big\{Z_k \notin \mathcal{T}_k\big\} \mathbb{1}\big\{\pi_k \geq 1/(KT^3)\big\} + T^{-3}$$

$$\lesssim T^{-3} \sum_{k=1}^{K} \pi_k + T^{-3} \asymp T^{-3}. \tag{103}$$

**Proof of Claim** (102). Towards this, fix an arbitrary $k \in [K]$ such that $\pi_k \geq 1/(KT^3)$. For any $x \in \mathcal{T}_k$, we know that for all $i \in [K]$,

$$\pi_i^{(t)} \leq \frac{\pi_i}{\pi_k} \exp\Big(-\frac{1}{2}\big\|x - \sqrt{\overline{\alpha}_t}\mu_i\big\|_2^2 + \frac{1}{2}\big\|x - \sqrt{\overline{\alpha}_t}\mu_k\big\|_2^2\Big) \wedge 1$$

$$= \frac{\pi_i}{\pi_k} \exp\Big(-\frac{1}{2}\overline{\alpha}_t\|\mu_i - \mu_k\|_2^2 + (x - \sqrt{\overline{\alpha}_t}\mu_k)^\top \sqrt{\overline{\alpha}_t}(\mu_i - \mu_k)\Big) \wedge 1$$

$$\leq \exp\Big(-\frac{1}{2}\overline{\alpha}_t\|\mu_i - \mu_k\|_2^2 + C_5\sqrt{\overline{\alpha}_t \log(KT)}\,\|\mu_i - \mu_k\|_2 + 3\log(KT)\Big) \wedge 1,$$

where the last line holds due to the definition of $\mathcal{T}_k$. As a result, for any $i \in [K]$ satisfying $\sqrt{\overline{\alpha}_t}\|\mu_i - \mu_k\|_2 > 6C_5\sqrt{\log(KT)}$, one has

$$\pi_i^{(t)} \leq \exp\Big(-\frac{1}{6}\overline{\alpha}_t\|\mu_i - \mu_k\|_2^2\Big)$$

as long as $C_5 \geq \sqrt{2}/2$. This further implies that

$$\pi_i^{(t)}\overline{\alpha}_t\|\mu_i - \mu_k\|_2^2 \leq \overline{\alpha}_t\|\mu_i - \mu_k\|_2^2 \exp\Big(-\frac{1}{6}\overline{\alpha}_t\|\mu_i - \mu_k\|_2^2\Big)$$

$$\leq \exp\Big(-\frac{1}{12}\overline{\alpha}_t\|\mu_i - \mu_k\|_2^2\Big) \leq \exp\big(-3C_5^2 \log(KT)\big), \tag{104}$$

provided $T$ is large enough. Meanwhile, for any $i \in [K]$ obeying $\sqrt{\overline{\alpha}_t}\|\mu_i - \mu_k\|_2 \leq 6C_5\sqrt{\log(KT)}$, we can simply upper bound

$$\pi_i^{(t)}\overline{\alpha}_t\|\mu_i - \mu_k\|_2^2 \leq \pi_i^{(t)} \cdot 36C_5^2 \log(KT). \tag{105}$$

Denote the set $\mathcal{F}_k := \big\{i \in [K] : \sqrt{\overline{\alpha}_t}\|\mu_i - \mu_k\|_2 \leq 6C_5\sqrt{\log(KT)}\big\}$. Using the expression of $J_t$ (cf. (19)), we conclude that

$$\mathsf{tr}\big(I_d + J_t(x)\big) = \sum_{i=1}^{K} \pi_i^{(t)}\overline{\alpha}_t\Big\|\mu_i - \sum_{k=1}^{K}\pi_k^{(t)}\mu_i\Big\|_2^2$$

$$\overset{(i)}{\leq} \sum_{i=1}^{K} \pi_i^{(t)}\overline{\alpha}_t\big\|\mu_i - \mu_k\big\|_2^2$$

$$\overset{(ii)}{\leq} 36C_5^2 \log(KT) \sum_{i\in\mathcal{F}_k} \pi_i^{(t)} + \sum_{i\in\mathcal{F}_k^c} \exp\big(-3C_5^2 \log(KT)\big)$$

$$\leq 36C_5^2 \log(KT)\log(KT) + K\exp\big(-3C_5^2 \log(KT)\big)$$

$$\leq C_1 \log(KT) \tag{106}$$

provided $C_5$ and $C_1/C_5^2$ are large enough. Here, (i) is true since $\sum_{k=1}^{K} \pi_k^{(t)}\mu_k$ is the minimizer of the function $x \mapsto \sum_{i=1}^{K} \pi_i^{(t)}\|\mu_i - x\|_2^2$ and (ii) uses (104)–(105). This establishes the claim (102).

BOUNDING THE REMAINING TERMS IN EQ. (99)

Next, let analyze the second event $\{1 \leq \sum_{k=1}^{K} \pi_k^{(t)} \exp(-\zeta_k^{(t)}(X_t^{\mathsf{GMM}})) \leq \exp(C_2(1-\alpha_t)^2 \log^2(KT))\}$. We first establish the lower bound of 1. For any $x \in \mathbb{R}^d$, given $\sum_k \pi_k^{(t)} = 1$, direct calculation shows that

$$
\sum_{k=1}^{K} \pi_k^{(t)} \left( \left\| x - \sqrt{\overline{\alpha}_t}\mu_k \right\|_2^2 - \sum_{i=1}^{K} \pi_i^{(t)} \left\| x - \sqrt{\overline{\alpha}_t}\mu_i \right\|_2^2 \right) + \sum_{k=1}^{K} \pi_k^{(t)} \sum_{i=1}^{K} \pi_i^{(t)} (\mu_i - \mu_k)
$$

$$
= \sum_{k=1}^{K} \pi_k^{(t)} \left\| x - \sqrt{\overline{\alpha}_t}\mu_k \right\|_2^2 - \sum_{i=1}^{K} \pi_i^{(t)} \left\| x - \sqrt{\overline{\alpha}_t}\mu_i \right\|_2^2 + \sum_{i=1}^{K} \pi_i^{(t)}\mu_i - \sum_{k=1}^{K} \pi_k^{(t)}\mu_k = 0.
$$

Combined with the definition of $\zeta_k^{(t)}(x)$ in (38), this yields

$$
\sum_{k=1}^{K} \pi_k^{(t)} \zeta_k^{(t)}(x) = 0.
$$

We can then apply Jensen's inequality to obtain that for any $x \in \mathbb{R}^d$,

$$
\sum_{k=1}^{K} \pi_k^{(t)} \exp\left(-\zeta_k^{(t)}(x)\right) \geq \exp\left(-\sum_{k=1}^{K} \pi_k^{(t)} \zeta_k^{(t)}(x)\right) = 1. \tag{107}
$$

Recall the definition of $\mathcal{T}_k$ in expression (100). To bound the second term in Eq. (99), it suffices to prove that for any $k \in [K]$ such that $\pi_k \geq 1/(KT^3)$,

$$
\mathcal{T}_k \subset \left\{ x \in \mathbb{R}^d : \sum_{k=1}^{K} \pi_k^{(t)} \exp\left(-\zeta_k^{(t)}(x)\right) \leq \exp\left(C_2(1-\alpha_t)^2 \log^2(KT)\right) \right\}. \tag{108}
$$

Indeed, assuming (108) holds, one can apply the same reasoning as that for (103) to obtain

$$
\mathbb{P}\left\{ \sum_{k=1}^{K} \pi_k^{(t)} \exp\left(-\zeta_k^{(t)}\left(X_t^{\mathsf{GMM}}\right)\right) > \exp\left(C_2(1-\alpha_t)^2 \log^2(KT)\right) \right\}
$$

$$
\leq \sum_{k=1}^{K} \pi_k \mathbb{P}\left\{ Z_k \notin \mathcal{T}_k \right\} \mathbb{1}\left\{ \pi_k \geq 1/(KT^3) \right\} + T^{-3} \lesssim T^{-3}, \tag{109}
$$

Taking this collectively with relations (103), and (99) completes the proof of Lemma 8. Now it is only left for us to prove inequality (108).

**Proof of inequality** (108).  To this end, recall the definitions of $\zeta_k^{(t)}(x)$ and $s_t^\star(x)$ in (38) and (35), respectively. By some basic algebra, $\zeta_k^{(t)}(x)$ can be written as

$$
\zeta_k^{(t)}(x) = \frac{1-\alpha_t^2}{2\alpha_t^2} \sum_{i=1}^{K} \pi_i^{(t)} \left( \left\| x - \sqrt{\overline{\alpha}_t}\mu_k \right\|_2^2 - \left\| x - \sqrt{\overline{\alpha}_t}\mu_i \right\|_2^2 \right)
$$

$$
+ \frac{1-\alpha_t}{\alpha_t^2} \left( -x + \sum_{i=1}^{K} \pi_i^{(t)} \sqrt{\overline{\alpha}_t}\mu_i \right)^\top \left( \sum_{i=1}^{K} \pi_i^{(t)} \sqrt{\overline{\alpha}_t}(\mu_i - \mu_k) \right)
$$

$$
= \frac{1-\alpha_t^2}{2\alpha_t^2} \sum_{i=1}^{K} \pi_i^{(t)} \left( -\frac{1}{2}\overline{\alpha}_t \|\mu_i - \mu_k\|_2^2 + (x - \sqrt{\overline{\alpha}_t}\mu_k)^\top \sqrt{\overline{\alpha}_t}(\mu_i - \mu_k) \right)
$$

$$
- \frac{1-\alpha_t}{\alpha_t^2} \sum_{i=1}^{K} \pi_i^{(t)} (x - \sqrt{\overline{\alpha}_t}\mu_k)^\top \sqrt{\overline{\alpha}_t}(\mu_i - \mu_k) + \frac{1-\alpha_t}{\alpha_t^2} \left\| \sum_{i=1}^{K} \pi_i^{(t)} \sqrt{\overline{\alpha}_t}(\mu_i - \mu_k) \right\|_2^2.
$$

For any $x \in \mathcal{T}_k$, one can obtain

$$\left|\zeta_k^{(t)}(x)\right| \lesssim (1-\alpha_t) \sum_{i=1}^K \pi_i^{(t)} \left(-\frac{1}{2}\overline{\alpha}_t \|\mu_i - \mu_k\|_2^2 + C_5 \sqrt{\overline{\alpha}_t \log(KT)} \|\mu_i - \mu_k\|_2\right)$$

$$+ (1-\alpha_t)\sqrt{\log(KT)} \sum_{i=1}^K \pi_i^{(t)} \sqrt{\overline{\alpha}_t} \|\mu_i - \mu_k\|_2 + (1-\alpha_t) \sum_{i=1}^K \pi_i^{(t)} \overline{\alpha}_t \|\mu_i - \mu_k\|_2^2$$

$$\asymp (1-\alpha_t) \sum_{i=1}^K \pi_i^{(t)} \overline{\alpha}_t \|\mu_i - \mu_k\|_2^2 + (1-\alpha_t)\sqrt{\log(KT)} \sum_{i=1}^K \pi_i^{(t)} \sqrt{\overline{\alpha}_t} \|\mu_i - \mu_k\|_2. \tag{110}$$

where the first inequality holds due to $1 - \alpha_t \lesssim \log T/T$ in (33), the definition of $\mathcal{T}_k$ in (100), and Jensen's inequality. Using the same argument as that for (106), it can be easily seen that

$$\sum_{i=1}^K \pi_i^{(t)} \sqrt{\overline{\alpha}_t} \|\mu_i - \mu_k\|_2 \lesssim \sqrt{\log(KT)}. \tag{111}$$

Plugging (111) and (106) into (110) demonstrates that

$$\left|\zeta_k^{(t)}(x)\right| \lesssim (1-\alpha_t)\log(KT) = o(1), \tag{112}$$

since $1 - \alpha_t \lesssim \log T/T$ as in (33).

As a consequence, for any $x \in \mathcal{T}_k$, we find that

$$\sum_{k=1}^K \pi_k^{(t)} \exp\left(-\zeta_k^{(t)}(x)\right) = \sum_{k=1}^K \pi_k^{(t)} \left(1 - \zeta_k^{(t)}(x) + \frac{1}{2}\left(\zeta_k^{(t)}(x)\right)^2 + o\left(\left(\zeta_k^{(t)}(x)\right)^2\right)\right)$$

$$= 1 + \frac{1}{2} \sum_{k=1}^K \pi_k^{(t)} \left(\zeta_k^{(t)}(x)\right)^2 + \sum_{k=1}^K \pi_k^{(t)} o\left(\left(\zeta_k^{(t)}(x)\right)^2\right)$$

$$= 1 + O\left((1-\alpha_t)^2 \log^2(KT)\right)$$

$$\leq \exp\left(C_2(1-\alpha_t)^2 \log^2(KT)\right)$$

as long as $C_2$ is sufficiently large. This establishes the claim (108), thereby leads to (109).

### B.6. Proof of Lemma 9

**Proof of Claim (84).** In light of the definition of $\mathcal{A}_t$ in (82), one has

$$\mathbb{P}\{X_t^{\mathsf{GMM}} \notin \mathcal{A}_t\} = \underbrace{\int_{\mathbb{R}^d} p_{X_t^{\mathsf{GMM}}}(x) \mathbb{1}\left\{\log p_{X_t^{\mathsf{GMM}}}(x) < -C_7 d\log(dT), \|x\|_2 \leq \sqrt{2T^{2c_R} + 16d\log T}\right\} dx}_{:=(\mathrm{I})}$$

$$+ \underbrace{\int_{\mathbb{R}^d} p_{X_t^{\mathsf{GMM}}}(x) \mathbb{1}\left\{\|x\|_2 > \sqrt{2T^{2c_R} + 16d\log T}\right\} dx}_{:=(\mathrm{II})}.$$

Thus, it suffices to control these two terms separately.

- For term (I), some direct algebra yields

$$(\mathrm{I}) \leq \int_{\mathbb{R}^d} \exp\left(-C_7 d\log(dT)\right) \mathbb{1}\left\{\|x\|_2 \leq \sqrt{2T^{2c_R} + 16d\log T}\right\} dx$$

$$\leq \exp\left(-C_7 d\log(dT)\right) 2^d \left(2T^{2c_R} + 16d\log T\right)^{d/2}$$

$$\leq \exp\left(-C_7 d\log(dT)\right) 2^{3d/2}\left(2^{d/2}T^{c_R d} + 16^{d/2}(d\log T)^{d/2}\right)$$

$$\leq \exp\big(-2d\log(dT)\big).$$

where the penultimate line holds as $(x+y)^{d/2} \leq 2^{d/2}(x^{d/2}+y^{d/2})$ for any $d \geq 1$ and $x, y > 0$; the last step is valid as long as $C_7/c_R$ is large enough.

- Regarding term (II), since $X_t^{\mathsf{GMM}} \sim \pi_k \sum_{k=1}^{K} \mathcal{N}(\sqrt{\overline{\alpha}_t}\mu_k, I_d)$, one can derive

$$
\begin{aligned}
(\mathrm{II}) &= \sum_{k\in[K]} \pi_k \mathbb{P}\Big\{\big\|\sqrt{\overline{\alpha}_t}\mu_k + Z\big\|_2^2 > 2T^{2c_R} + 16d\log T\Big\} \\
&\overset{(\mathrm{i})}{\leq} \sum_{k\in[K]} \pi_k \mathbb{P}\Big\{\overline{\alpha}_t\|\mu_k\|_2^2 + \|Z\|_2^2 > T^{2c_R} + 8d\log T\Big\} \\
&\overset{(\mathrm{ii})}{\leq} \sum_{k\in[K]} \pi_k \mathbb{P}\big\{\|Z\|_2^2 > 2d + 6d\log T\big\} \\
&\overset{(\mathrm{iii})}{\lesssim} \sum_{k\in[K]} \pi_k \exp\big(-2d\log T\big) \\
&\leq \exp\big(-2d\log T\big).
\end{aligned}
$$

where $Z \sim \mathcal{N}(0, I_d)$ is a standard Gaussian random vector in $\mathbb{R}^d$. Here, (i) applies the Cauchy-Schwartz inequality; (ii) holds due to $\overline{\alpha}_t \in (0, 1)$ and Assumption 1 that $\max_k \|\mu_k\|_2 \leq T^{c_R}$; (iii) applies the Gaussian concentration inequality (95).

Putting the above bounds together yields Claim (84).

**Proof of Claim** (85). As $X_t^{\mathsf{GMM}} \mid X_0^{\mathsf{GMM}} = x_0 \sim \mathcal{N}(\sqrt{\overline{\alpha}_t}x_0, (1-\overline{\alpha}_t)I_d)$, we can use Tweedie's formula to derive (see also Saha & Guntuboyina (2020, Lemma 4.3) and Jiang & Zhang (2009, Theorem 3)):

$$
\begin{aligned}
\big\|s_t^{\star\mathsf{GMM}}(x)\big\|_2^2 &= \bigg\|\frac{1}{1-\overline{\alpha}_t}\mathbb{E}\Big[\sqrt{\overline{\alpha}_t}X_0^{\mathsf{GMM}} - x \mid X_t^{\mathsf{GMM}} = x\Big]\bigg\|_2^2 \\
&\leq \frac{1}{(1-\overline{\alpha}_t)^2}\mathbb{E}\Big[\big\|\sqrt{\overline{\alpha}_t}X_0^{\mathsf{GMM}} - x\big\|_2^2 \mid X_t^{\mathsf{GMM}} = x\Big] \\
&\overset{(\mathrm{i})}{\leq} \frac{2}{1-\overline{\alpha}_t}\log\mathbb{E}\bigg[\exp\Big(\frac{1}{2(1-\overline{\alpha}_t)}\big\|\sqrt{\overline{\alpha}_t}X_0^{\mathsf{GMM}} - x\big\|_2^2\Big) \mid X_t^{\mathsf{GMM}} = x\bigg] \\
&\overset{(\mathrm{ii})}{=} -\frac{2}{1-\overline{\alpha}_t}\log\Big(\big(2\pi(1-\overline{\alpha}_t)\big)^{d/2}p_{X_t^{\mathsf{GMM}}}(x)\Big).
\end{aligned}
\tag{113}
$$

Here, (i) applies Jensen's inequality; (ii) is true because

$$
p_{\sqrt{\alpha_t}X_0^{\mathsf{GMM}}|X_t^{\mathsf{GMM}}}(y \mid x) = \big(2\pi(1-\overline{\alpha})\big)^{-d/2}\exp\Big(-\frac{1}{2(1-\overline{\alpha}_t)}\|y-x\|_2^2\Big)\frac{p_{X_0^{\mathsf{GMM}}}(y)}{p_{X_t^{\mathsf{GMM}}}(x)},
$$

which further leads to

$$
\begin{aligned}
\mathbb{E}\bigg[\exp\Big(\frac{1}{2(1-\overline{\alpha}_t)}\big\|\sqrt{\overline{\alpha}_t}X_0^{\mathsf{GMM}} - x\big\|_2^2\Big) \mid X_t^{\mathsf{GMM}} = x\bigg] &= \int_{y\in\mathbb{R}^d}\exp\Big(\frac{1}{2(1-\overline{\alpha}_t)}\|y-x\|_2^2\Big)p_{\sqrt{\alpha_t}X_0^{\mathsf{GMM}}|X_t^{\mathsf{GMM}}}(y \mid x)\,\mathrm{d}y \\
&= \big(2\pi(1-\overline{\alpha}_t)\big)^{-d/2}\frac{1}{p_{X_t^{\mathsf{GMM}}}(x)}.
\end{aligned}
$$

Therefore, as $\log p_{X_t^{\mathsf{GMM}}}(x) \geq -C_7 d\log T$ on the set $\mathcal{A}_t$, we arrive at the claimed bound:

$$
\big\|s_t^{\star\mathsf{GMM}}(x)\big\|_2^2 \leq \frac{d}{1-\overline{\alpha}_t}\log\frac{1}{2\pi(1-\overline{\alpha}_t)} - \frac{2}{1-\overline{\alpha}_t}\log p_{X_t^{\mathsf{GMM}}}(x)
$$

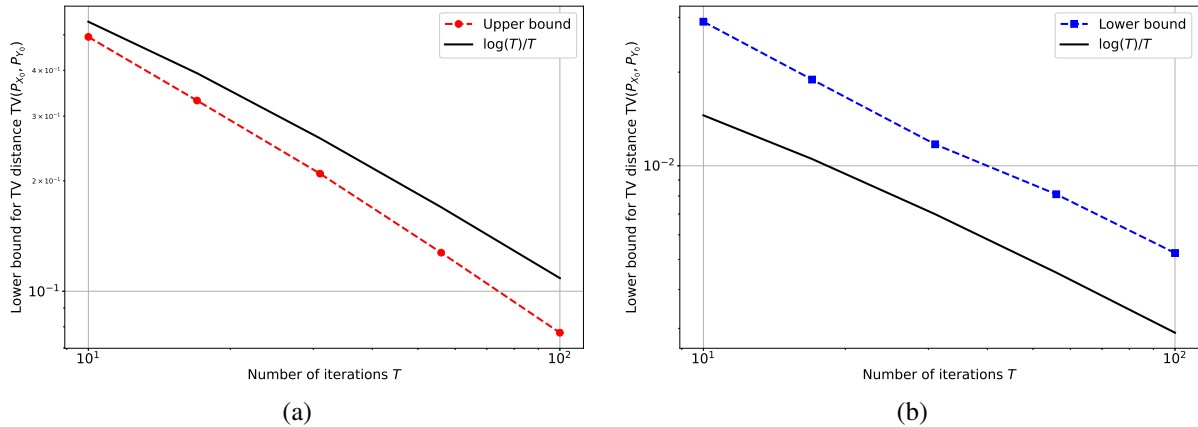

(a)                                (b)

*Figure 2.* Take the setting in Figure 1 and fix $d = 50, K = 50$. We plot the upper and lower bounds, in (a) and (b) respectively, of sampling errors measured in TV distance. Both bounds are compared with the function $T \mapsto C(\log T/T)$ for $C = 2.36$ and $0.06$.

$$\lesssim \frac{d}{1 - \overline{\alpha}_t} \log \frac{c_1 T}{\pi \log T} + \frac{2}{1 - \overline{\alpha}_t} \log \left( C_7 d \log T \right)$$

$$\leq C_{\mathsf{clip}} \frac{d \log T}{1 - \overline{\alpha}_t}, \tag{114}$$

where the penultimate step holds since $1 - \overline{\alpha}_t \geq c_1(T/\log T)(1 - \alpha_t) \geq (c_1/2)T/\log T$ due to (14) and (33) from Lemma 3; the last line holds provided $C_{\mathsf{clip}}$ is large enough.

**Proof of Claim** (86). Applying Tweedie's formula again, we can derive

$$\mathbb{E}\left[ \left\| s_t^{\star \mathsf{GMM}} (X_t^{\mathsf{GMM}}) \right\|_2^4 \right] = \frac{1}{(1 - \overline{\alpha}_t)^4} \mathbb{E}\left[ \left\| \mathbb{E}\left[ X_t^{\mathsf{GMM}} - \sqrt{\overline{\alpha}_t} X_0^{\mathsf{GMM}} \mid X_t^{\mathsf{GMM}} \right] \right\|_2^4 \right]$$

$$\overset{(i)}{\leq} \frac{1}{(1 - \overline{\alpha}_t)^4} \mathbb{E}\left[ \mathbb{E}\left[ \left\| X_t^{\mathsf{GMM}} - \sqrt{\overline{\alpha}_t} X_0^{\mathsf{GMM}} \right\|_2^4 \mid X_t^{\mathsf{GMM}} \right] \right]$$

$$\overset{(ii)}{=} \frac{1}{(1 - \overline{\alpha}_t)^4} \mathbb{E}\left[ \left\| X_t^{\mathsf{GMM}} - \sqrt{\overline{\alpha}_t} X_0^{\mathsf{GMM}} \right\|_2^4 \right]$$

$$\overset{(iii)}{\lesssim} \left( \frac{d}{1 - \overline{\alpha}_t} \right)^2 .$$

Here, (i) applies the convexity of the function $x \mapsto \|x\|_2^4$ and Jensen's inequality; (ii) uses the tower property; and (iii) leverages the properties of the standard Gaussian distribution.

## C. Implementation details

In this section, we provide some implementation ideas for our simulations in Figure 1. Evaluating the TV distance between two high-dimensional distributions in a sample-efficient manner is challenging, even for high dimensional GMMs with descent $K$. In order to numerically valid our theoretical results (as in Theorem 1), we develop an upper and a lower bounds that grow at the same rate with $T$ to estimate the TV distance.

**Upper bound.** First, we can upper bound $\mathsf{TV}(X_0, Y_0)$ by triangle's inequality and Markov's property as

$$\mathsf{TV}(X_0, Y_0) \leq \mathsf{TV}(X_T, Y_T) + \sum_{t=1}^{T} \int_{x_{t-1}} \left| p_{X_{t-1}}(x_{t-1}) - \int_{x_t \in \mathcal{E}_t} p_{Y_{t-1}^\star | Y_t^\star}(x_{t-1} \mid x_t) p_{X_t}(x_t) \mathrm{d}x_t \right| \mathrm{d}x_{t-1}$$

$$\leq \mathsf{TV}(X_T, Y_T) + \sum_{t=1}^{T} \int_{x_{t-1} \notin \mathcal{E}_{t-1}} p_{X_{t-1}}(x_{t-1}) \mathrm{d}x_{t-1}$$

$$+ \sum_{t=1}^{T} \int_{x_{t-1} \in \mathcal{E}_{t-1}} \left| p_{X_{t-1}}(x_{t-1}) - \int_{x_t \in \mathcal{E}_t} p_{Y_{t-1}^\star | Y_t^\star}(x_{t-1} \,|\, x_t) p_{X_t}(x_t) \mathrm{d}x_t \right| \mathrm{d}x_{t-1}, \tag{115}$$

where we recall the definition of set $\mathcal{E}_{t-1}$ as in (37).

The first two terms can be controlled given Lemma 8 and our initialization of DDPM. It then remains to consider the third term. Here, we make use of the following two relations:

$$p_{X_{t-1}}(x_{t-1}) = \int_{u_t} \left( \frac{1}{4\pi^2 \alpha_t (1-\alpha_t)} \right)^{d/2} \sum_{k=1}^{K} \pi_k \exp\left( -\frac{\|u_t - \sqrt{\bar{\alpha}_t} \mu_k\|^2}{2\alpha_t^2} \right) \exp\left( -\frac{\|\sqrt{\alpha_t} x_{t-1} - u_t\|^2}{2(1-\alpha_t)\alpha_t} \right) \mathrm{d}u_t$$

$$= \int_{x_t} \left( \frac{1}{4\pi^2 \alpha_t (1-\alpha_t)} \right)^{d/2} \sum_{k=1}^{K} \pi_k \exp\left( -\frac{\|u_t - \sqrt{\bar{\alpha}_t} \mu_k\|^2}{2\alpha_t^2} \right) \exp\left( -\frac{\|\sqrt{\alpha_t} x_{t-1} - u_t\|^2}{2(1-\alpha_t)\alpha_t} \right) \det\left( I + (1-\alpha_t) J_t(x_t) \right) \mathrm{d}x_t,$$

and

$$\int_{x_t \in \mathcal{E}_t} p_{Y_{t-1}^\star | Y_t^\star}(x_{t-1} \,|\, x_t) p_{X_t}(x_t) \mathrm{d}x_t = \int_{x_t \in \mathcal{E}_t} p_{X_t}(x_t) \left( \frac{\alpha_t}{2\pi(1-\alpha_t)} \right)^{d/2} \exp\left( -\frac{\|\sqrt{\alpha_t} x_{t-1} - u_t\|^2}{2(1-\alpha_t)\alpha_t} \right) \mathrm{d}x_t.$$

As a result, the third term on the right hand side of (115) can be controlled by the following expression:

$$\mathbb{E}_{X_t \sim p_{X_t}}[|\Delta_t(X_t)|] + \int_{x_t \notin \mathcal{E}_t} p_{X_t}(x_t) \mathrm{d}x_t, \tag{116}$$

where

$$\Delta_t(x_t) := \frac{1}{p_{X_t}(x_t)} \left( \frac{1}{2\pi\alpha_t^2} \right)^{d/2} \sum_{k=1}^{K} \pi_k \exp\left( -\frac{\|u_t - \sqrt{\bar{\alpha}_t} \mu_k\|^2}{2\alpha_t^2} \right) \det\left( I + (1-\alpha_t) J_t(x_t) \right) - 1$$

with $u_t = x_t + (1-\alpha_t) s_t^\star(x_t)$. To further bound the term in (116), in view of Lemma 8, the second term is small, and we use Monte Carlo method with 100 samples of $X_t$ to compute the expectation term.

**Lower bound.** In view of the data processing inequality for TV distance, we lower bound the TV distance by considering one-dimensional projections of $X_0$ and $Y_0$. We generate $10^6$ samples using DDPM and project the samples onto their first coordinates. We then construct the kernel density estimator (KDE), and calculate the TV distance between the KDE and the true density of the projected GMM via numerical integration.

**Results.** In Figure 2, we show the upper and lower bounds of the sampling error in TV distance as a function of the number of iterations $T$. Both bounds decrease with increasing $T$ and can be well approximated by $\log T/T$ up to a constant factor, consistent with our theoretical prediction. In Figure 1, we use the upper-bound estimate for the TV distance between $X_0$ and $Y_0$; this choice does not affect our conclusions, as the upper and lower bounds exhibit the same scaling behavior.

