# OpenReview forum: "Dimension-free convergence of diffusion models for approximate Gaussian mixtures"
_ICML.cc/2026/Conference — ICML 2026 regular_

### Official Review · Reviewer_4Eqr · 2026-03-11

**Soundness:** 3
**Presentation:** 3
**Significance:** 3
**Originality:** 3
**Overall Recommendation:** 5
**Confidence:** 3

**Summary:**

This paper studies the iteration complexity of the Denoising Diffusion Probabilistic Model (DDPM) when the target distribution can be well-approximated by a Gaussian Mixture Model (GMM) with isotropic covariance. The main result (Theorem 1) establishes that DDPM requires at most Õ(1/ε) iterations to achieve an ε-accurate distribution in total variation (TV) distance, given access to sufficiently accurate score estimates and a small GMM approximation error. Remarkably, this iteration complexity is independent of both the ambient dimension d and the number of mixture components K, up to logarithmic factors. The result is also shown to be robust to score estimation errors: the TV distance between the learned and target distributions scales proportionally to the score estimation error (modulo logarithmic factors). The paper provides a four-step proof strategy, that is, constructing auxiliary processes, bounding discretization error, relating score estimation error, and controlling GMM approximation error. The key technical insight is that the Jacobian trace tr(I_d + J_t(X_t^{GMM})) is bounded by O(log(KT)), which is dimension-free. Numerical experiments on synthetic GMMs corroborate the theoretical findings.

**Compliance With Llm Reviewing Policy:**

Affirmed.

**Final Justification:**

I thank the authors for their thoughtful and honest rebuttal. The clarification that the GMM assumption should be viewed as a structural condition rather than a universal approximation claim is helpful and should be made explicit in the revision. The discussion of the technical obstacle for non isotropic covariances (loss of the isotropic cancellation in the Jacobian trace bound) is clear. I maintain my score of 5.

**Key Questions For Authors:**

1. Could you provide a concrete example of a non-GMM distribution and quantify its ε_approx as a function of d? This would help assess when the dimension-free guarantee is practically meaningful versus when the √(d·ε_approx) term dominates.

2. The key insight is that tr(I_d + J_t(X_t^{GMM})) = O(log(KT)). Does an analogous dimension-free trace bound hold for GMMs with general (non-isotropic) covariances? If not, what is the obstacle?

3. How does the end-to-end complexity look when combining your sampling guarantee with the score estimation results of Chen et al. (2024) or Gatmiry et al. (2024)? Does the combined result still exhibit advantageous dimension dependence compared to existing end-to-end guarantees?

**Limitations:**

yes

**Strengths And Weaknesses:**

### Strengths

1. The dimension-free iteration complexity for GMM targets is a significant theoretical advance. Prior work (Liang et al., 2024b; Li & Yan, 2024b) established that iteration complexity scales at least linearly with d for GMMs or general distributions. Removing the d-dependence entirely (up to logs) demonstrates a qualitatively different phenomenon: diffusion models can implicitly exploit GMM structure to bypass the dimension barrier. This is precisely the kind of result that provides deep insight into why diffusion models work well in practice on high-dimensional structured data.

2. The proof strategy is elegant and clearly isolated as the source of dimension-freeness. The auxiliary process construction is technically sophisticated but well-motivated. The use of high-probability events to handle truncation is a careful technical device.

### Weaknesses

1. The paper focuses exclusively on GMMs with identity covariance (σ = 1). While the authors briefly mention that general σ can be handled by rescaling (Section 3), extending to mixtures with arbitrary but well-conditioned covariances (the practically relevant case) is non-trivial and left entirely open. The discussion (Section 5) acknowledges this limitation, but given that real-world data distributions are rarely well-approximated by isotropic GMMs, this significantly limits the practical significance of the result.

2. When ε_approx > 0, the bound in Theorem 1 includes the term √(d·ε_approx)·log^{3/2}(dT), which depends on √d. For the result to yield an ε-accurate sampler, one needs ε_approx = O(ε²/d) (ignoring logs), which reintroduces dimension dependence through the approximation quality requirement. The paper would benefit from a more explicit discussion of how well practical distributions (e.g., image distributions) can be approximated by isotropic GMMs and what the resulting ε_approx would be as a function of d.

---

> ### Author Rebuttal · Authors · 2026-03-30
>
> We thank the reviewer for the careful reading and for the positive assessment.
>
> **GMM approximation.**
> First, we would like to emphasize that the GMM assumption is not merely a mathematically convenient abstraction. GMMs are among the most classical and widely used probabilistic models in statistics and machine learning. They are valued for their ability to tractably model heterogeneous and multi-modal data, and also serve as a standard flexible approximation family for complex densities. In this sense, the GMM assumption already captures an important practically relevant regime. At the same time, we agree that the most realistic formulation would involve mixtures with general covariances; our current isotropic setting should be viewed as a first step that isolates the core mechanism behind the dimension-free sampling phenomenon in the cleanest possible way. Extending the result to mixtures with general well-conditioned covariances is an important next direction.
>
> Second, our theory is not intended to claim that all distributions can be well approximated by isotropic GMMs with moderate $K.$ In the worst case, approximating a general $d$-dimensional distribution by isotropic Gaussians may require $K$ to grow exponentially with $d$, in which case our result does not yield a practically meaningful dimension-free guarantee. Rather, the intended regime is that of **structured data distributions**, where the GMM assumption serves as a tractable model for multi-cluster or effectively low-dimensional structure. In such cases, our theorem shows that DDPM can exploit this structure automatically during sampling.
>
> Thus, we view the GMM approximation assumption as a **structural condition**, not as a literal claim that real-world distributions are exactly isotropic GMMs. We will revise the paper to make this interpretation more explicit and to better distinguish the theorem’s structural message from a universal approximation claim.
>
> **Dimension-free behavior for general GMMs.**
> We believe such an extension is plausible, but it is not immediate. The isotropic setting is special because the score and its Jacobian have a particularly clean form: each component contributes the same curvature in every direction, and this symmetry is what allows us to control the Jacobian trace through the posterior mixture weights. This is the key mechanism behind the dimension-free bound in our current proof. For GMMs with general well-conditioned covariances, the score Jacobian becomes direction-dependent, and the isotropic cancellation used in the proof no longer holds in the same way. Even if all covariance matrices are uniformly well-conditioned, one still needs to control how anisotropy interacts with posterior component weights across the mixture. So the main obstacle is the loss of the isotropic structure that makes the trace bound tractable. Our current view is that a related structure-adaptive result may still hold under additional assumptions and with new techniques, but establishing a dimension-free trace bound in that setting is an interesting open problem. We will make this limitation and the technical obstacle more explicit in the revision.
>
> **End-to-end guarantee.**
> Our work focuses specifically on the sampling stage and studies how fast DDPM converges given a sufficiently accurate score estimate. Therefore, when combined with score estimation results such as Chen et al. and Gatmiry et al. (2024), our theorem shows that if the target distribution is well approximated by a GMM, then the sampling error of DDPM converges at a nearly dimension-free rate to the score-estimation error characterized by their analysis.
>
> On the other hand, from an information-theoretic perspective, we do not expect fully dimension-free guarantees in general. Even for a simple GMM, estimating the component means from samples typically incurs dimension dependence, since these objects live in \mathbb{R}^d. Therefore, while the sampling stage can be dimension-free in our theorem, the learning stage will generally still depend on $d$ unless one imposes additional low-dimensional or sparsity structure on the parameters themselves. So a genuinely dimension-free end-to-end theory would likely require stronger structure beyond the GMM assumption alone.

---

> > ### Author Rebuttal · Reviewer_4Eqr · 2026-04-02
> >
> > I thank the authors for their thoughtful and honest rebuttal. The clarification that the GMM assumption should be viewed as a structural condition rather than a universal approximation claim is helpful and should be made explicit in the revision. The discussion of the technical obstacle for non isotropic covariances (loss of the isotropic cancellation in the Jacobian trace bound) is clear. I maintain my score of 5.

---

> > > ### Author Response · Authors · 2026-04-04
> > >
> > > Thank you again for your time and understanding. In the revision, we will make the role of the GMM approximation assumption more explicit and add further discussion on the technical difficulties of extending the analysis to GMMs with non-isotropic covariances.

---

### Official Review · Reviewer_4G4X · 2026-03-12

**Soundness:** 3
**Presentation:** 3
**Significance:** 3
**Originality:** 4
**Overall Recommendation:** 4
**Confidence:** 3

**Summary:**

This paper studies the iteration complexity of DDPM for sampling from distributions that are well-approximated by Gaussian mixture models (GMMs). The authors prove that, under certain assumptions, DDPM achieves an $\tilde{O}(1/\epsilon)$ iteration complexity in TV distance, which is independent of both the ambient dimension $d$ and the number of mixture components $K$ up to logarithmic factors. The result is robust to score estimation errors and provides theoretical justification for the empirical observation that diffusion models can efficiently sample from high-dimensional structured distributions.

**Compliance With Llm Reviewing Policy:**

Affirmed.

**Key Questions For Authors:**

1. The paper focuses on the sampling phase, assuming perfect or near-perfect score estimates. For an end-to-end theory, one would need to combine this with score estimation guarantees. Do you have thoughts on how the sample complexity of score estimation might interact with the dimension-free sampling result? Could the overall procedure still achieve dimension-free total complexity?
2. The experiments in Figure 1 are limited to $d,K \leq 50$. Have you tested higher dimensions (e.g., $d=500$) with correspondingly larger $K$ to see if the TV distance remains stable? Does the logarithmic dependence on $K$ hold when $K$ is large relative to $d$? Additional experiments would help validate the dimension-free claim empirically.
3. The bound includes a term $\sqrt{d\epsilon_{\text{approx}}}\log^{3/2}(dT)$, indicating that approximation error is amplified by dimension. For a fixed target distribution and increasing $d$, how should one expect the required number of components $K$ to scale to keep $\epsilon_{\text{approx}}$ sufficiently small? Can you provide intuition or examples (e.g., smooth densities, manifolds) where this scaling might be favorable?

**Limitations:**

The authors discuss some limitations in the conclusion—extension to general covariances, matching lower bounds, end-to-end theory—but do not address the practical challenges of Assumption 2, the implicit dimension dependence through the GMM approximation condition, or the lack of empirical validation across a wider range of $d$ and $K$. A more thorough discussion of these points would help readers assess the result's applicability and guide future work.

**Strengths And Weaknesses:**

## Strength
1. The paper presents a significant theoretical advance by demonstrating dimension-free convergence for diffusion models on distributions close to GMMs. The key insight—linking the trace of the score Jacobian to logarithmic scaling rather than linear dimension dependence—is novel and well-exploited. The proof technique is rigorous, building a sophisticated auxiliary process framework to handle the discrepancy between true and estimated scores. The result stands in contrast to prior work requiring $\tilde{O}(d/\epsilon)$ iterations, highlighting an important adaptive property of diffusion models to data structure. The authors address the key question of whether diffusion models can escape the curse of dimensionality for structured distributions, and their affirmative answer is theoretically compelling.
## Weakness
1. Assumption 1 requires the target distribution to be close to a GMM in TV distance. However, as dimension increases, the number of components $K$ needed to approximate a given distribution to a fixed accuracy $\epsilon_{\text{approx}}$ may itself grow with $d$. While the final bound depends only logarithmically on $K$, it does not account for this potential implicit dimension dependence. The term $\sqrt{d\epsilon_{\text{approx}}}\log^{3/2}(dT)$ also suggests that approximation error is amplified by dimension, which could undermine the dimension-free claim in practice.
2. The experimental support is confined to Figure 1, which shows TV distance scaling for $d,K \leq 50$. The paper does not systematically investigate how the bounds behave for larger dimensions, nor does it explore the relationship between $d$, $K$, and approximation quality.

---

> ### Author Rebuttal · Authors · 2026-03-30
>
> We thank the reviewer for the constructive feedback.
>
> **Dimension-free for score estimation.**
> This is an excellent question. From an information-theoretic perspective, we do not expect fully dimension-free guarantees in general. Even for a simple GMM, estimating the component means from samples typically incurs dimension dependence, since these objects live in $\mathbb{R}^d$. Therefore, while the sampling stage can be dimension-free in our theorem, the score learning stage will generally still depend on $d$ unless one imposes additional low-dimensional or sparsity structure on the parameters themselves.
> So a genuinely dimension-free end-to-end theory would likely require stronger structure beyond the GMM assumption alone. We agree that developing such an end-to-end theory is an important open problem, and we will emphasize this more clearly.
> We also note that score estimation for GMMs has already been studied in Chen et al. (2024) and Gatmiry et al. (2024), as discussed in the introduction. Our paper is therefore complementary to this line of work: rather than investigating the statistical aspect of score-learning guarantees, we focus on the sampling stage and study the computational aspect.
>
> **Experiments.**
> This is a good suggestion. Our current experiment is intended as a controlled proof-of-concept to validate the scaling predicted by the theorem in a regime where TV distance can still be estimated reliably, rather than as a comprehensive empirical study across all parameter ranges. The main obstacle to exploring much larger values of $d$ and $K$ is that evaluating TV distance itself becomes computationally prohibitive in those regimes.
> We will revise the paper to make this point clearer: the current experiment is designed to confirm the predicted trend in a controlled setting, while a broader empirical study over larger $d$ and $K$ is an important direction for future work.
>
> **GMM approximation.**
> First, we would like to emphasize that the GMM assumption is not merely a mathematically convenient abstraction. GMMs are among the most classical and widely used probabilistic models in statistics and machine learning. They are valued for their ability to tractably model heterogeneous and multi-modal data, and also serve as a standard flexible approximation family for complex densities. In this sense, the GMM assumption already captures an important practically relevant regime. At the same time, we agree that the most realistic formulation would involve mixtures with general covariances; our current isotropic setting should be viewed as a first step that isolates the core mechanism behind the dimension-free sampling phenomenon in the cleanest possible way. Extending the result to mixtures with general well-conditioned covariances is an important next direction.
>
> Second, our theory is not intended to claim that all distributions can be well approximated by isotropic GMMs with moderate $K.$ In the worst case, approximating a general $d$-dimensional distribution by isotropic Gaussians may require $K$ to grow exponentially with $d$, in which case our result does not yield a practically meaningful dimension-free guarantee. Rather, the intended regime is that of **structured data distributions**, where the GMM assumption serves as a tractable model for multi-cluster or effectively low-dimensional structure. In such cases, our theorem shows that DDPM can exploit this structure automatically during sampling.
>
> From this viewpoint, the $\sqrt{d\varepsilon_{\mathrm{approx}}}$ term should be read as saying that the GMM approximation must be sufficiently accurate for the theorem to be informative; it is not meant to suggest that isotropic GMMs are universally efficient approximators in high dimension. We will revise the discussion to emphasize this interpretation more clearly: the theorem identifies a regime where once the target admits a sufficiently accurate and structured GMM approximation, DDPM sampling automatically adapts to that structure and avoids the standard linear dependence on $d$.

---

> > ### Author Rebuttal · Reviewer_4G4X · 2026-04-07
> >
> > We thank the authors for their detailed rebuttal. However, like other reviewers, we remain concerned about how often real-world high-dimensional distributions can be well approximated by isotropic GMMs with moderate K. The practical significance of the dimension‑free guarantee therefore remains unclear. We maintain our score of 4.

---

### Official Review · Reviewer_5ZZb · 2026-03-12

**Soundness:** 3
**Presentation:** 3
**Significance:** 3
**Originality:** 2
**Overall Recommendation:** 4
**Confidence:** 4

**Summary:**

This paper studies the convergence rates of DDMP when applied to target distributions that can be approximated by Gaussian mixture models (GMM). They establish that given perfect score estimates and a small GMM approximation error, DDPM requires at most $\tilde{O}(1/\epsilon)$ iterations to achieve an $\epsilon$-accurate distribution in TV distance. When the GMM approximation error is sufficiently small, this iteration complexity is independent of both the data's ambient dimension $d$ and the number of GMM components $K$, up to logarithmic factors.

**Compliance With Llm Reviewing Policy:**

Affirmed.

**Final Justification:**

I thank the authors for their detailed rebuttal. However, I share Reviewer 52ht's concern that it remains unclear to see how often this structural regime actually applies to realistic high-dimensional distributions. The practical impact of the dimension-free sampling guarantee is limited if the approximation requires $K$ to scale exponentially with $d$, or when considering the inherently dimension-dependent nature of end-to-end learning. Nevertheless, the technical analysis provides a sound preliminary effort, so I will maintain my score of 4.

**Key Questions For Authors:**

1. If the authors can provide a more nuanced discussion or perhaps a corollary showing how $K$ scales with $d$ for typical structured distributions (even under the isotropic assumption), it would significantly strengthen the paper's claim of practical relevance.

2. Can the authors comment on whether the structure of GMMs could potentially allow for dimension-free (or at least highly efficient) score estimation as well, perhaps by leveraging the explicit form of the GMM score?

3. The crucial step in escaping the dimension dependence is Lemma 8, which bounds the trace of the Jacobian: $tr(I_d + J_t(x)) \le C_1 \log(KT)$ on the typical set $\mathcal{E}_t$. While this is independent of $d$, it does depend on the number of components $K$. How tight is this $\log(K)$ bound? Is it possible that for highly overlapping components, the trace could be bounded by a quantity even smaller than $\log(K)$?

**Limitations:**

Yes.

**Strengths And Weaknesses:**

## Strenghts

- The paper is technically sound.

- To my knowledge, the established $\tilde{O}(1/\epsilon)$ iteration complexity represents a state-of-the-art result for DDPMs.

- The finding that DDPM can adapt to the intrinsic structure of GMMs without explicit low-dimensional modeling is interesting. This suggests that the practical success of diffusion models is deeply tied to their ability to exploit the latent structure of real-world data, opening avenues for analyzing other structured distributions.


## Weaknesses

- The paper relies heavily on the assumption that the target distribution can be well-approximated by a mixture of **isotropic** Gaussians. While GMMs are universal approximators, using strictly isotropic components to approximate highly correlated or lower-dimensional manifold distributions would require an **exponentially** large number of components ($K$). Moreover, the requirement for a small approximation error $\epsilon_{apprx}$ (specifically $\tilde{O}(\epsilon^2/d)$) creates a tension with the isotropic assumption. Achieving such a small error in high dimensions with isotropic components might necessitate a $K$ so large that the logarithmic dependence on $K$ is no longer negligible, potentially undermining the practical applicability of the "dimension-free" claim.

- While the result is original, some of the analytical techniques (e.g., using TV distance instead of KL divergence, and the specific bounds on SDE discretization) build heavily on recent concurrent works (e.g., Li \& Yan, 2024b). The novelty lies more in the application of these tools to the GMM setting and the subsequent derivation of the bounded Jacobian trace, rather than the invention of entirely new mathematical machinery.

- The paper assumes we already have a good estimate of the score function with a bounded error $\epsilon_{score}$. However, even if the target is a simple GMM, estimating its parameters (such as the $K$ different $d$-dimensional means) may require a sample complexity that inherently scales with the dimension $d$. Since $\epsilon_{score}$ itself will practically depend on $d$ due to the difficulty of high-dimensional learning, the final end-to-end performance remains coupled to the ambient dimension. The authors acknowledge this limitation, noting that a complete end-to-end theory remains a crucial open direction.

---

> ### Author Rebuttal · Authors · 2026-03-30
>
> We thank the reviewers for their thoughtful comments and for their time.
>
> **GMM approximation.**
> First, we would like to emphasize that the GMM assumption is not merely a mathematically convenient abstraction. GMMs are among the most classical and widely used probabilistic models in statistics and machine learning. They are valued for their ability to tractably model heterogeneous and multi-modal data, and also serve as a standard flexible approximation family for complex densities. In this sense, the GMM assumption already captures an important practically relevant regime. At the same time, we agree that the most realistic formulation would involve mixtures with general covariances; our current isotropic setting should be viewed as a first step that isolates the core mechanism behind the dimension-free sampling phenomenon in the cleanest possible way. Extending the result to mixtures with general well-conditioned covariances is an important next direction.
>
> Second, our theory is not intended to claim that all distributions can be well approximated by isotropic GMMs with moderate $K.$ In the worst case, approximating a general $d$-dimensional distribution by isotropic Gaussians may require $K$ to grow exponentially with $d$, in which case our result does not yield a practically meaningful dimension-free guarantee. Rather, the intended regime is that of **structured data distributions**, where the GMM assumption serves as a tractable model for multi-cluster or effectively low-dimensional structure. In such cases, our theorem shows that DDPM can exploit this structure automatically during sampling.
>
> Thus, we view the GMM approximation assumption as a **structural condition**, not as a literal claim that real-world distributions are exactly isotropic GMMs. We will revise the paper to make this interpretation more explicit and to better distinguish the theorem’s structural message from a universal approximation claim.
>
> **Dimension-free for score estimation.**
> This is an excellent question. From an information-theoretic perspective, we do not expect fully dimension-free guarantees in general. Even for a simple GMM, estimating the component means from samples typically incurs dimension dependence, since these objects live in $\mathbb{R}^d$. Therefore, while the sampling stage can be dimension-free in our theorem, the score learning stage will generally still depend on $d$ unless one imposes additional low-dimensional or sparsity structure on the parameters themselves.
> So a genuinely dimension-free end-to-end theory would likely require stronger structure beyond the GMM assumption alone. We agree that developing such an end-to-end theory is an important open problem, and we will emphasize this more clearly.
> We also note that score estimation for GMMs has already been studied in Chen et al. (2024) and Gatmiry et al. (2024), as discussed in the introduction. Our paper is therefore complementary to this line of work: rather than investigating the statistical aspect of score-learning guarantees, we focus on the sampling stage and study the computational aspect.
>
> **$\log K$ dependence.**
> We agree that the $\log K$ dependence should be viewed as a worst-case upper bound, rather than as always tight. Intuitively, this factor arises because the score of a GMM is governed by the posterior mixture weights, and in the worst case, one may need to account for uncertainty among up to $K$ competing components. This is what leads to a logarithmic dependence on $K$.
> However, in regimes where the components are highly overlapping, the effective complexity can be smaller. In such cases, the posterior weights are often much more regular, and one may expect a sharper bound depending on the effective number of active components at a point $x$, rather than the global value of $K$. So it is plausible that for highly overlapping mixtures the trace could be bounded by something strictly smaller than $\log K$, perhaps in terms of a local notion of active mixture complexity.
> Our current lemma does not pursue this refinement, since $\log K$ is already sufficient for the main theorem, but we agree that this is a very interesting direction. We will add a remark clarifying that the $\log K$ term is likely not always tight and may be improvable in more special mixture geometries.

---

> > ### Author Rebuttal · Reviewer_5ZZb · 2026-04-04
> >
> > I thank the authors for their detailed rebuttal. However, I share Reviewer 52ht's concern that it remains unclear to see how often this structural regime actually applies to realistic high-dimensional distributions. The practical impact of the dimension-free sampling guarantee is limited if the approximation requires $K$ to scale exponentially with $d$, or when considering the inherently dimension-dependent nature of end-to-end learning. Nevertheless, the technical analysis provides a sound preliminary effort, so I will maintain my score of 4.

---

> > > ### Author Response · Authors · 2026-04-04
> > >
> > > Thank you again for your time and thoughtful feedback, and for recognizing this work as a sound preliminary effort. In the revised paper, we will expand the discussion of the GMM approximation assumption, including its scope, applicability, and limitations.

---

### Official Review · Reviewer_52ht · 2026-03-12

**Soundness:** 3
**Presentation:** 3
**Significance:** 2
**Originality:** 3
**Overall Recommendation:** 4
**Confidence:** 3

**Summary:**

This paper studies the sampling efficiency of DDPMs when the target distribution can be approximated by a Gaussian mixture model (GMM). The main claim is that the number of sampling iterations needed to obtain an $\varepsilon$-accurate distribution in total variation distance scales as $\tilde{O}(1/\varepsilon)$ and is independent of both the ambient dimension $d$ and the number of mixture components $K$ (up to logarithmic factors). The analysis focuses on isotropic GMMs.

**Compliance With Llm Reviewing Policy:**

Affirmed.

**Key Questions For Authors:**

1. Do the authors believe the dimension-free behavior would still hold for GMMs with general (well-conditioned) covariance matrices?
2. In practice, how well do realistic data distributions satisfy the GMM approximation assumption used in the theory?

**Limitations:**

Overall, the paper presents an interesting theoretical observation, but I am not fully convinced about the practical significance of the result yet. The analysis relies on fairly specific assumptions, and it is unclear how much insight it provides into the behavior of diffusion models on real data. Stronger empirical evidence or extensions of the theory to more general distributions would make the contribution more compelling.

**Strengths And Weaknesses:**

# Strength:
- The research question is interesting: why diffusion models seem to work well in high-dimensional settings despite theoretical analyses often suggesting a dimension dependence.
- The proof structure is reasonably organized, with different auxiliary processes used to separate the different sources of error.
- The result itself is potentially interesting if it holds more broadly, since dimension-free convergence would help explain some empirical observations about diffusion models.

# Weaknesses / Concerns

My main concern is that it is not entirely clear how important or meaningful the result is in practice.

First, the analysis is limited to isotropic GMMs. While this is mathematically convenient, it is a fairly restrictive class of distributions. Real data distributions are rarely close to spherical mixtures.

Additionally, the approximation assumption for the GMM also becomes more demanding as the dimension increases (the required approximation error scales like $\tilde{O}(\varepsilon^2/d)$). This seems to partially offset the dimension-free claim, since achieving such an approximation might itself become harder in high dimensions.


Also, the empirical section is quite limited. The main figure only studies scaling with dimension. It would be helpful to see experiments varying other factors such as the number of mixture components $K$, the separation between components, or the approximation quality.

---

> ### Author Rebuttal · Authors · 2026-03-30
>
> We thank the reviewer for the careful reading and the insightful comments.
>
> **Dimension-free behavior for general GMMs.**
> We believe such an extension is plausible, but it is not immediate. The isotropic setting is special because the score and its Jacobian have a particularly clean form: each component contributes the same curvature in every direction, and this symmetry is what allows us to control the Jacobian trace through the posterior mixture weights. This is the key mechanism behind the dimension-free bound in our current proof. For GMMs with general well-conditioned covariances, the score Jacobian becomes direction-dependent, and the isotropic cancellation used in the proof no longer holds in the same way. Even if all covariance matrices are uniformly well-conditioned, one still needs to control how anisotropy interacts with posterior component weights across the mixture. So the main obstacle is the loss of the isotropic structure that makes the trace bound tractable. Our current view is that a related structure-adaptive result may still hold under additional assumptions and with new techniques, but establishing a dimension-free trace bound in that setting is an interesting open problem. We will make this limitation and the technical obstacle more explicit in the revision.
>
> **GMM approximation.**
> First, we would like to emphasize that the GMM assumption is not merely a mathematically convenient abstraction. GMMs are among the most classical and widely used probabilistic models in statistics and machine learning. They are valued for their ability to tractably model heterogeneous and multi-modal data, and also serve as a standard flexible approximation family for complex densities. In this sense, the GMM assumption already captures an important practically relevant regime. At the same time, we agree that the most realistic formulation would involve mixtures with general covariances; our current isotropic setting should be viewed as a first step that isolates the core mechanism behind the dimension-free sampling phenomenon in the cleanest possible way. Extending the result to mixtures with general well-conditioned covariances is an important next direction.
>
> Second, our theory is not intended to claim that all distributions can be well approximated by isotropic GMMs with moderate $K.$ In the worst case, approximating a general $d$-dimensional distribution by isotropic Gaussians may require $K$ to grow exponentially with $d$, in which case our result does not yield a practically meaningful dimension-free guarantee. Rather, the intended regime is that of **structured data distributions**, where the GMM assumption serves as a tractable model for multi-cluster or effectively low-dimensional structure. In such cases, our theorem shows that DDPM can exploit this structure automatically during sampling.
>
> Thus, we view the GMM approximation assumption as a **structural condition**, not as a literal claim that real-world distributions are exactly isotropic GMMs. We will revise the paper to make this interpretation more explicit and to better distinguish the theorem’s structural message from a universal approximation claim.
>
> **Experiments.**
> This is a good suggestion. Our current experiment is intended as a controlled proof-of-concept to validate the scaling predicted by the theorem in a regime where TV distance can still be estimated reliably, rather than as a comprehensive empirical study across all parameter ranges. The main obstacle to exploring much larger values of $d$ and $K$ is that evaluating TV distance itself becomes computationally prohibitive in those regimes.
> We will revise the paper to make this point clearer: the current experiment is designed to confirm the predicted trend in a controlled setting, while a broader empirical study over larger $d$ and $K$ is an important direction for future work.

---

> > ### Author Rebuttal · Reviewer_52ht · 2026-04-03
> >
> > Thank you for the clear and thoughtful rebuttal. I appreciate the explanation of why isotropy is essential to the current analysis and the discussion of challenges in extending to general GMMs.
> >
> > However, my main concern about practical significance remains only partially addressed. The result relies on the assumption that the data distribution can be well-approximated by an isotropic GMM with a moderate number of components. While the rebuttal clarifies this as a structural regime, it is still unclear how often this applies in realistic high-dimensional settings. In particular, the approximation complexity may still scale unfavorably with dimension, which could limit the practical implications of the dimension-free guarantee.
> >
> > Overall, while the technical contribution is sound, addressing these concerns would likely require more substantial theoretical or empirical extensions beyond what can be covered in a short rebuttal.

---

> > > ### Author Response · Authors · 2026-04-04
> > >
> > > Thank you again for your careful reading and thoughtful comments, and for recognizing the technical contribution of this work. In the revised paper, we will make the role of the isotropic GMM approximation assumption more explicit and further discuss its scope and limitations.

---

### Decision · Program_Chairs · 2026-04-30

**Decision:**

Accept (regular)

**Comment:**

This paper addresses the key question of why diffusion models, specifically DDPMs, can efficiently sample high-dimensional structured distributions. The authors consider a pressing problem: the iteration complexity of DDPM when the target distribution can be approximated by a Gaussian mixture model (GMM). They prove that, under isotropic GMM assumptions and sufficiently accurate score estimates, DDPM achieves ε-accurate sampling in total variation distance with Õ(1/ε) iterations, independent of ambient dimension and the number of mixture components (up to logarithmic factors). The result is robust to score estimation errors and provides theoretical justification for observed empirical efficiency.

Strengths include a rigorous and elegant proof strategy, the novel use of the Jacobian trace to avoid linear dimension dependence, and the insight that DDPM implicitly exploits structure in GMMs without explicit low-dimensional modeling. All reviewers agree the technical contribution is sound and novel.

Limitations involve practical applicability: the isotropic GMM assumption restricts generalization to realistic high-dimensional data, approximation error may implicitly scale with dimension, and experimental validation is limited to small synthetic examples. End-to-end guarantees incorporating score learning remain an open direction.

Overall, this is a technically solid and original contribution that advances understanding of diffusion models. While practical impact is limited under current assumptions, it identifies a structural regime where dimension-free sampling occurs and lays groundwork for future extensions.